# A coarse-grained NADH redox model enables inference of subcellular metabolic fluxes from fluorescence lifetime imaging

**Xingbo Yang[1]\*, Gloria Ha[1], Daniel J Needleman[1,2]**

[1]Department of Molecular and Cellular Biology and John A. Paulson School of Engineering and Applied Sciences, Harvard University, Cambridge, United States; [2]Center for Computational Biology, Flatiron Institute, New York, United States

**Abstract** Mitochondrial metabolism is of central importance to diverse aspects of cell and developmental biology. Defects in mitochondria are associated with many diseases, including cancer, neuropathology, and infertility. Our understanding of mitochondrial metabolism in situ and dysfunction in diseases are limited by the lack of techniques to measure mitochondrial metabolic fluxes with sufficient spatiotemporal resolution. Herein, we developed a new method to infer mitochondrial metabolic fluxes in living cells with subcellular resolution from fluorescence lifetime imaging of NADH. This result is based on the use of a generic coarse-grained NADH redox model. We tested the model in mouse oocytes and human tissue culture cells subject to a wide variety of perturbations by comparing predicted fluxes through the electron transport chain (ETC) to direct measurements of oxygen consumption rate. Interpreting the fluorescence lifetime imaging microscopy measurements of NADH using this model, we discovered a homeostasis of ETC flux in mouse oocytes: perturbations of nutrient supply and energy demand of the cell do not change ETC flux despite significantly impacting NADH metabolic state. Furthermore, we observed a subcellular spatial gradient of ETC flux in mouse oocytes and found that this gradient is primarily a result of a spatially heterogeneous mitochondrial proton leak. We concluded from these observations that ETC flux in mouse oocytes is not controlled by energy demand or supply, but by the intrinsic rates of mitochondrial respiration.

## Editor's evaluation

This paper describes the derivation and validation of a coarse-grained model to measure mitochondrial metabolism at cellular and subcellular resolution by exploiting fluorescence lifetime imaging of NADH. This technique is applied to mouse oocytes subjected to a variety of metabolic stresses and to human tissue culture cells, revealing spatial gradients in mitochondrial NADH oxidation. This method represents an exciting new approach to quantifying mitochondrial electron transport chain rates and provides for the first time a method to study mitochondrial metabolic flux with subcellular resolution.

**\*For correspondence:**
xingbo_yang@fas.harvard.edu

**Competing interest:** The authors declare that no competing interests exist.

## Introduction

Cells transduce energy from the environment to power cellular processes. Decades of extensive research have produced a remarkable body of detailed information about the biochemistry of mitochondrial energy metabolism (*Salway, 2017*). In brief, metabolites, such as pyruvate, are transported into mitochondria, where they are broken down and their products enter the tricarboxylic acid cycle (TCA). The TCA is composed of a number of chemical reactions, which ultimately reduces $NAD^+$ to NADH. NADH and oxygen are then utilized by the electron transport chain

(ETC) to pump hydrogen ions across the mitochondrial membrane. ATP synthase uses this proton gradient to power the synthesis of ATP from ADP (*Mitchell, 1961*). The activities of mitochondrial energy metabolism are characterized by the fluxes through these pathways: that is, the number of molecules turned over per unit time (*Stephanopoulos, 1999*). However, despite the wealth of knowledge concerning mitochondrial biochemistry, the spatiotemporal dynamics of cellular energy usage remains elusive and it is still unclear how cells partition energy across different cellular processes (*Dumollard et al., 2007*; *Van Blerkom, 2011*; *Yellen, 2018*; *Yang et al., 2021*) and how energy metabolism is misregulated in diseases (*Brand and Nicholls, 2011*; *Lin and Beal, 2006*; *Wallace, 2012*; *Bratic and Larsson, 2013*; *Lowell and Shulman, 2005*; *Mick et al., 2020*). Metabolic heterogeneities, between and within individual cells, are believed to be widespread, but remain poorly characterized (*Takhaveev and Heinemann, 2018*; *Aryaman et al., 2018*). Mitochondria have been observed to associate with the cytoskeleton (*Lawrence et al., 2016*), spindle (*Wang et al., 2020*), and endoplasmic reticulum (*Dumollard et al., 2004*) and display subcellular heterogeneities in mtDNA sequence (*Morris et al., 2017*) and mitochondrial membrane potential (*Smiley et al., 1991*). These observations suggest the potential existence of subcellular patterning of mitochondrial metabolic fluxes that could be critical in processes such as oocyte maturation (*Yu et al., 2010*) and embryo development (*Sanchez et al., 2019*). The limitations of current techniques for measuring mitochondrial metabolic fluxes with sufficient spatiotemporal resolution present a major challenge. In particular, there is a lack of techniques to measure mitochondrial metabolic fluxes with single cell and subcellular resolution.

Bulk biochemical techniques for measuring metabolic fluxes, such as oxygen consumption and nutrient uptake rates (*Ferrick et al., 2008*; *Houghton et al., 1996*; *Lopes et al., 2005*), and isotope tracing by mass spectrometry (*Wiechert, 2001*), require averaging over large populations of cells. Such techniques cannot resolve cellular, or subcellular, metabolic heterogeneity (*Takhaveev and Heinemann, 2018*; *Aryaman et al., 2018*). Biochemical approaches for measuring mitochondrial metabolic fluxes, such as mass spectrometry, are also often destructive (*Wiechert, 2001*; *Saks et al., 1998*), and thus cannot be used to observe continual changes in fluxes over time. Fluorescence microscopy provides a powerful means to measure cellular and subcellular metabolic heterogeneity continuously and non-destructively, with high spatiotemporal resolution. However, while fluorescent probes can be used to measure mitochondrial membrane potential (*Perry et al., 2011*) and the concentration of key metabolites (*Imamura et al., 2009*; *Berg et al., 2009*; *Díaz-García et al., 2017*; *San Martín et al., 2014*), it is not clear how to relate those observables to mitochondrial metabolic fluxes.

NADH is an important cofactor that is involved in many metabolic pathways, including the TCA and ETC in mitochondria. NADH binds with enzymes and acts as an electron carrier that facilitates redox reactions. In the ETC, for example, NADH binds to complex I and donates its electron to ubiquinone and ultimately to oxygen, becoming oxidized to NAD$^+$. Endogenous NADH has long been used to non-invasively probe cellular metabolism because NADH is autofluorescent, while NAD$^+$ is not (*Heikal, 2010*). Fluorescence lifetime imaging microscopy (FLIM) of NADH autofluorescence allows quantitative measurements of the concentration of NADH, the fluorescence lifetimes of NADH, and the fraction of NADH molecules bound to enzymes (*Becker, 2012*; *Becker, 2019*; *Bird et al., 2005*; *Skala et al., 2007*; *Heikal, 2010*; *Sharick et al., 2018*; *Sanchez et al., 2018*; *Sanchez et al., 2019*; *Ma et al., 2019*). It has been observed that the fraction of enzyme-bound NADH and NADH fluorescence lifetimes are correlated with the activity of oxidative phosphorylation, indicating that there is a connection between NADH enzyme-binding and mitochondrial metabolic fluxes (*Bird et al., 2005*; *Skala et al., 2007*). The mechanistic basis of this empirical correlation has been unclear.

Here, we developed a generic coarse-grained NADH redox model that enables the inference of ETC flux with subcellular resolution from FLIM measurements. We validated this model in mouse oocytes and human tissue culture cells subject to a wide range of perturbations by comparing predicted ETC fluxes from FLIM to direct measurements of oxygen consumption rate (OCR), and by a self-consistency criterion. Using this method, we discovered that perturbing nutrient supply and energy demand significantly impacts NADH metabolic state but does not change ETC flux. We also discovered a subcellular spatial gradient of ETC flux in mouse oocytes and found that this flux gradient is primarily due to a spatially heterogeneous mitochondrial proton leak. We concluded from these observations that ETC flux in mouse oocytes is not controlled by energy demand or supply, but by the intrinsic rates of mitochondrial respiration. Thus, FLIM of NADH can be used to non-invasively and

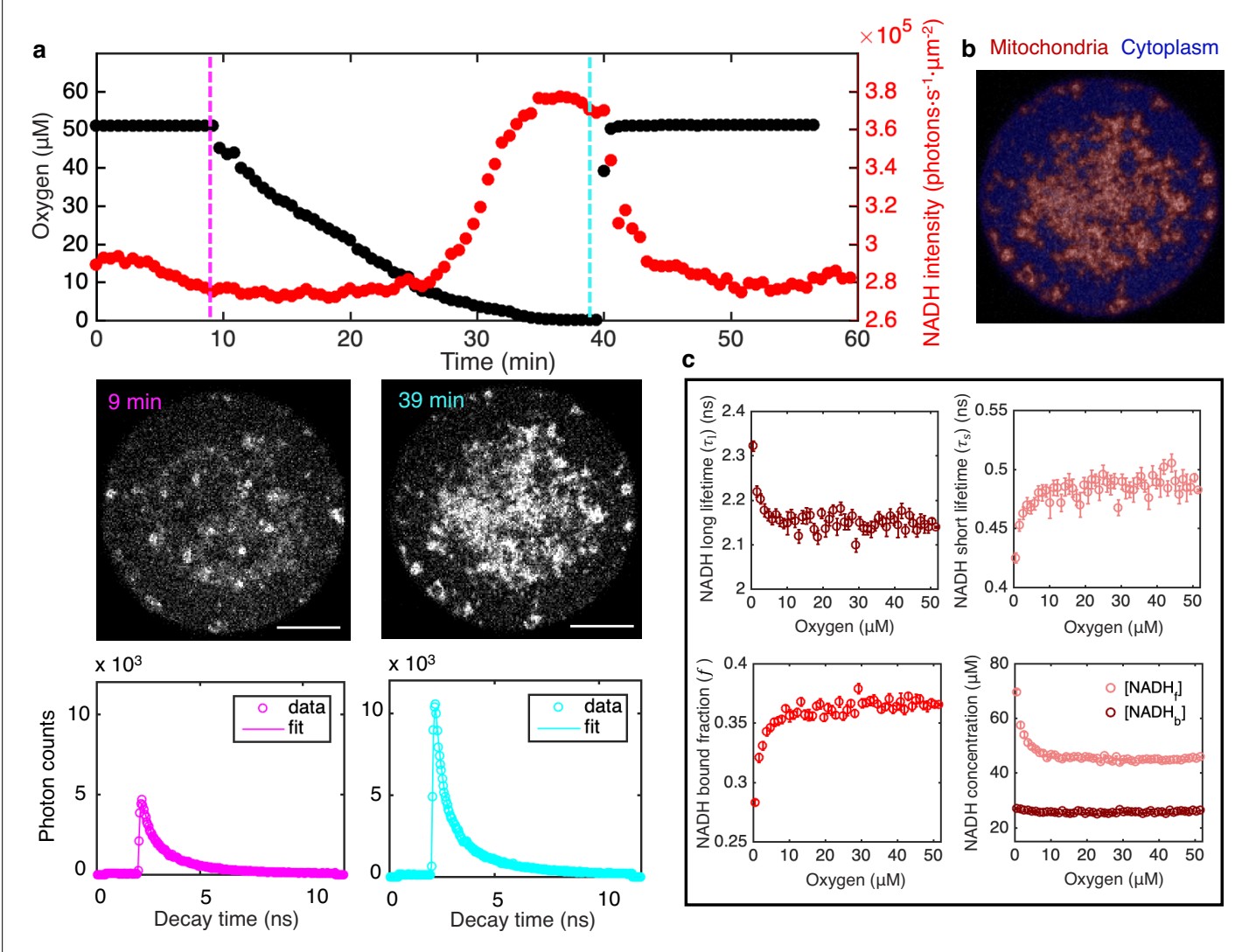

**Figure 1.** FLIM measurements of the response of mitochondrial NADH as a function of oxygen level. (**a**) Top row: oxygen level (black circles) and mitochondrial NADH intensity (red circles) as a function of time. Middle row: NADH intensity images of MII mouse oocyte at high and low oxygen levels corresponding to times indicated by the vertical lines. Scale bar, 20 μm. Bottom row: NADH fluorescence decay curves of the corresponding oocyte at low and high oxygen levels, with corresponding fits. (**b**) NADH-intensity-based segmentation of mitochondria and cytoplasm. (**c**) Mitochondrial NADH long fluorescence lifetime $\tau_1$ (upper left), short fluorescence lifetime $\tau_s$ (upper right), and bound fraction $f$ (lower left) as a function of oxygen level ($n$=68 oocytes). These FLIM parameters can be used in combination with intensity, $I$, and proper calibration, to obtain the concentration of free NADH, $[\mathrm{NADH_f}]$, and the concentration of enzyme-bound NADH, $[\mathrm{NADH_b}]$, in mitochondria as a function of oxygen (lower right). Error bars are standard error of the mean (s.e.m) across individual oocytes. FLIM, fluorescence lifetime imaging microscopy.

The online version of this article includes the following figure supplement(s) for figure 1:

**Figure supplement 1.** Machine learning based segmentation of mitochondria from NADH intensity images.

**Figure supplement 2.** Calibration and conversion of NADH concentrations from fluorescence intensities and lifetimes in vitro.

**Figure supplement 3.** Measurement of concentrations of free and bound NADH in vitro from FLIM of NADH.

continuously measure mitochondrial ETC fluxes with subcellular resolution and provides novel insights into the spatiotemporal regulation of metabolic fluxes in cells.

## Results

### Quantifying response of mitochondrial metabolism to changing oxygen levels and metabolic inhibitors using FLIM of NADH

We used meiosis II arrested mouse oocytes as a model system. MII oocytes are in a metabolic steady-state, which eases interpretations of metabolic perturbations. ATP synthesis in mouse oocytes occurs primarily through oxidative phosphorylation using pyruvate, without an appreciable contribution from glycolysis (*Houghton et al., 1996*), providing an excellent system to study mitochondrial metabolism. Mouse oocytes can be cultured in vitro using chemically well-defined media (*Biggers and Racowsky, 2002*). In our work, we used AKSOM as the culturing media (*Summers, 2013*). The oocytes can directly take up pyruvate supplied to them or derive it from lactate through the activity of lactate dehydrogenase (LDH) (*Lane and Gardner, 2000*), and they can remain in a steady-state for hours with constant metabolic fluxes. While NADH and NADPH are difficult to distinguish with fluorescence measurements due to their overlapping fluorescence spectrum, the concentration of NADH in mouse oocytes is 40 times greater than the concentration of NADPH for the whole cell (*Bustamante et al., 2017*) and potentially even greater in mitochondria (*Zhao et al., 2011*), so the autofluorescence signal from these cells (particularly from mitochondria) can be safely assumed to result from NADH.

To investigate how FLIM measurements vary with mitochondrial activities, we performed quantitative metabolic perturbations. We first continually varied the concentration of oxygen in the media, from 50±2 μM to 0.26±0.04 μM, over a course of 30 min while imaging NADH autofluorescence of oocytes with FLIM (*Figure 1a*, top, black curve; *Video 1*). NADH is present in both mitochondria and cytoplasm where it is involved in different metabolic pathways. To specifically study the response of NADH in mitochondria, we used a machine learning-based algorithm to segment mitochondria from the NADH intensity images (*Berg et al., 2019*; *Figure 1b* and *Figure 1—figure supplement 1*). We

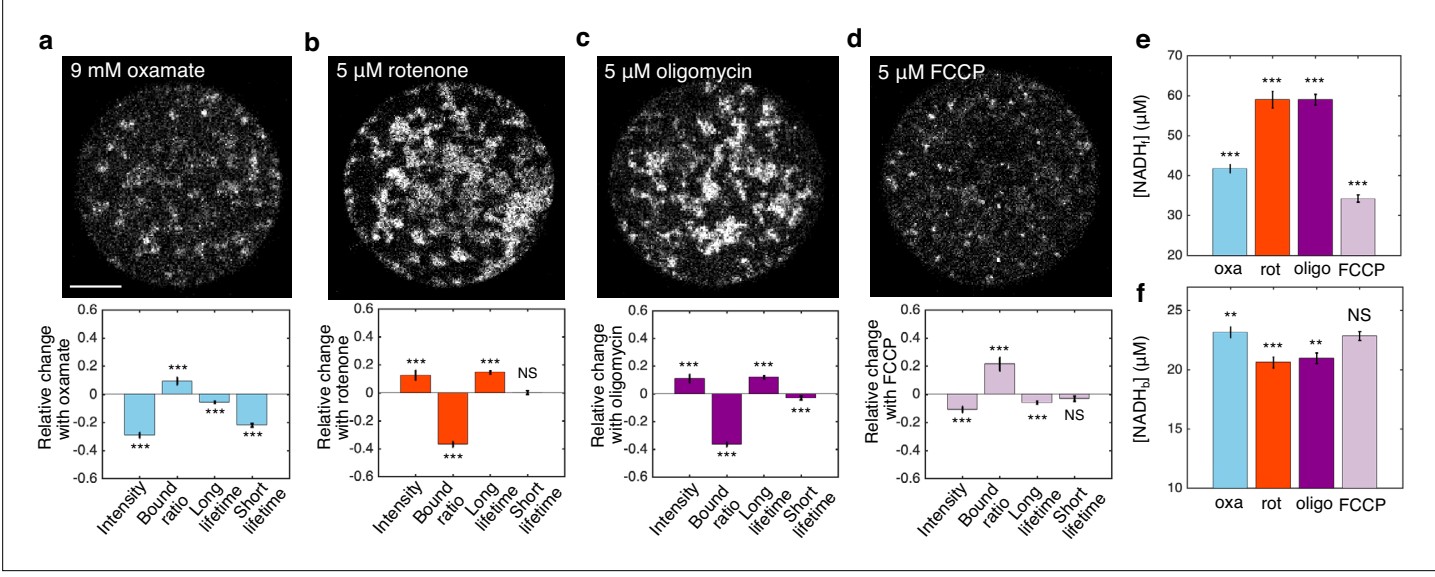

**Figure 2.** FLIM measurements of mitochondrial NADH under the impact of metabolic inhibitors. (**a–d**) NADH intensity images (scale bar, 20 μm) and the corresponding changes of FLIM parameters in response to 9 mM oxamate (**a**) (*n*=28), and with an additional 5 μM rotenone (**b**) (*n*=28), 5 μM oligomycin (**c**) (*n*=37), and 5 μM FCCP (**d**) (*n*=31) perturbations. *n* is the number of oocytes. 15–30 min have elapsed between the administration of the drugs and the measurements. (**e**) Free NADH concentrations ($[\mathrm{NADH_f}]$). (**f**) Bound NADH concentrations ($[\mathrm{NADH_b}]$). Error bars represent standard error of the mean (s.e.m) across different oocytes. Student's *t*-test is performed between parameters before and after the perturbation. \**p*<0.05, \*\**p*<0.01, \*\*\**p*<0.001. FLIM, fluorescence lifetime imaging microscopy.

The online version of this article includes the following source data for figure 2:

**Source data 1.** Excel spreadsheet of single-oocyte FLIM data used for *Figure 2a–f*.

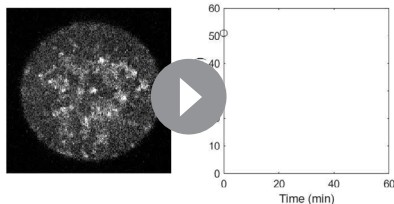

**Video 1.** NADH intensity in mouse oocyte as a function of oxygen level. Left: imaging of NADH from autofluorescence of mouse oocyte. Right: real time measurement of oxygen level in the imaging chamber. https://elifesciences.org/articles/73808/figures#video1

verified the accuracy of the segmentation with a mitochondrial labeling dye, MitoTracker Red FM, which showed a 80.6±1% (SEM) accuracy of the segmentation (Appendix 1).

Using the segmentation mask, we obtained the intensity of NADH, $I$, in mitochondria by averaging the photon count over all mitochondrial pixels. The intensity increased with decreasing oxygen concentration (*Figure 1a*, top, red), as is readily seen from the raw images (*Figure 1a*, middle). Restoring oxygen to its original level caused a recovery of NADH intensity, indicating that the observed changes are reversible (*Figure 1a*; *Video 1*). These observations are consistent with the expectation that NADH concentration will increase at low oxygen levels due to oxygen's role as the electron acceptor in the ETC. In addition to intensity, FLIM can be used to determine the enzyme engagement of NADH by measuring the photon arrival times, from which fluorescence lifetimes can be fitted. Enzyme-bound NADH has a much longer fluorescence lifetime than free NADH (*Sharick et al., 2018*), allowing bound and free NADH to be separately resolved, but the precise fluorescence lifetimes of NADH depend on a range of factors, including viscosity, pH, and the identity of the enzyme NADH binds to (*Sharick et al., 2018*; *Ghukasyan and Heikal, 2015*). To fit NADH fluorescence lifetimes, we grouped all detected photons from mitochondria to form histograms of photon arrival times from NADH autofluorescence for each time point (*Figure 1a*, lower). We fitted the histograms using a model in which the NADH fluorescence decay, $F(\tau)$, is described by the sum of two exponentials,

$$F(\tau) = f \cdot \exp\left(-\frac{\tau}{\tau_1}\right) + \left(1 - f\right) \cdot \exp\left(-\frac{\tau}{\tau_s}\right), \tag{1}$$

where $\tau_l$ and $\tau_s$ are long and short fluorescence lifetimes, corresponding to enzyme-bound NADH and free NADH, respectively, and $f$ is the fraction of enzyme-bound NADH (*Sanchez et al., 2018*; *Sanchez et al., 2019*) (Materials and methods).

We repeated the oxygen drop experiments for a total of 68 oocytes. Since the oxygen drop is much slower than the NADH redox reactions (30 min compared to a timescale of seconds), the oxygen perturbation can be safely assumed to be quasistatic, allowing the FLIM measurements to be determined as a function of oxygen levels. We averaged data from all oocytes to obtain a total of four FLIM parameters: mitochondrial NADH intensity, $I$, long and short fluorescence lifetimes, $\tau_l$ and $\tau_s$, and the fraction of enzyme-bound NADH, $f$. We determined how these parameters varied with oxygen level (*Figure 1a and c*). All parameters are insensitive to oxygen level until oxygen drops below ~10 µM. This observation is consistent with previous studies that showed mitochondria have a very high apparent affinity for oxygen (*Chance and Williams, 1955*; *Gnaiger et al., 1998*).

We next explored the relationship between the measured FLIM parameters and the concentration of NADH. Since bound and free NADH have different fluorescence lifetimes, and hence different molecular brightnesses, the NADH concentration is not generally proportional to NADH intensity. Assuming molecular brightness is proportional to fluorescence lifetime (*Lakowicz, 2006*), we derived a relation between NADH intensity, fluorescence lifetimes, and concentrations as

$$\left[\text{NADH}_\text{f}\right] = \frac{I(1-f)}{c_\text{s}[(\tau_l - \tau_\text{s})f + \tau_\text{s}]}, \tag{2a}$$

$$\left[\text{NADH}_\text{b}\right] = \left[\text{NADH}_\text{f}\right] \frac{f}{1-f}, \tag{2b}$$

where $c_\text{s}$ is a calibration factor that relates intensities and concentrations (see Appendix 1). We measured the calibration factor by titrating free NADH in vitro and acquiring FLIM data (*Figure 1—figure supplement 2*, *Equation S4*). To test the validity of this approach, we used *Equations 2a,b* to measure concentrations of free and bound NADH in solutions with different concentrations of purified LDH, an enzyme to which NADH can bind. The measured NADH bound concentration increases with

LDH concentration while the sum of free and bound NADH concentration remains a constant and equal to the amount of NADH added to the solution (*Figure 1—figure supplement 3*). This result demonstrates that *Equations 2a,b* can be used to measure free and bound NADH concentrations from NADH intensity and lifetimes. It is well established that FLIM can be used to distinguish bound and free NADH in vivo based on the large change of fluorescence lifetime when NADH binds to enzymes (*Skala et al., 2007*; *Heikal, 2010*). Even though the exact amount that the lifetime changes depend on the specific enzyme NADH binds to (*Sharick et al., 2018*), enzyme-bound NADH always has a much longer fluorescence lifetime than free NADH. Therefore, the method to calculate free and bound concentrations of NADH from FLIM measurements is expected to hold in vivo. We next used this method to study NADH in mitochondria in oocytes. We applied *Equations 2a,b* to our FLIM data from oocytes and determined how the concentrations of free NADH, $[\mathrm{NADH_f}]$, and enzyme-bound NADH, $[\mathrm{NADH_b}]$, depended on oxygen level (*Figure 1c*, lower right). Interestingly, $[\mathrm{NADH_f}]$ increased as oxygen fell below ~10 μM, while $[\mathrm{NADH_b}]$ did not vary with oxygen level.

We next explored the impact of metabolic inhibitors on mitochondrial NADH. We first inhibited LDH by adding 9 mM of oxamate to the AKSOM media. This led to a decrease of NADH intensity in the mitochondria (*Figure 2a*, upper) and significant changes in all FLIM parameters (*Figure 2a*, lower, p<0.001). We next inhibited complex I of the ETC using 5 μM of rotenone (in the presence of 9 mM of oxamate, to reduce cytoplasmic NADH signal for better mitochondrial segmentation). This resulted in a dramatic increase of NADH intensity in the mitochondria (*Figure 2b*, upper) and significant changes in NADH bound ratio and long lifetime (*Figure 2b*, lower, *p*<0.001). Then we inhibited ATP synthase with 5 μM of oligomycin (in the presence of 9 mM of oxamate), which, similar to rotenone, resulted in an increase of mitochondrial NADH intensity (*Figure 2c*, upper) and significant changes in all FLIM parameters (*Figure 2c*, lower, *p*<0.001). Finally, we subjected the oocytes to 5 μM of FCCP (in the presence of 9 mM of oxamate), which uncouples proton translocation from ATP synthesis, and observed a decrease of mitochondrial NADH intensity (*Figure 2d*, upper) and significant changes in FLIM parameters (*Figure 2d*, lower, *p*<0.001). Interestingly, the direction of change of FLIM parameters under FCCP is opposite to those under rotenone and oligomycin. For each of these conditions, we used *Equations 2a,b* to calculate the concentrations of free NADH, $[\mathrm{NADH_f}]$, (*Figure 2e*) and bound NADH, $[\mathrm{NADH_b}]$, (*Figure 2f*) from the measured intensity and FLIM parameters. While rotenone and oligomycin significantly increased $[\mathrm{NADH_f}]$ and decreased $[\mathrm{NADH_b}]$, FCCP decreased $[\mathrm{NADH_f}]$. It remains unclear how to relate these changes of the free and bound concentrations of NADH to the activities of mitochondrial respiration.

## Developing a coarse-grained NADH redox model to relate FLIM measurements of NADH to activities of mitochondrial metabolic pathways

We next developed a mathematical model of NADH redox reactions to relate these quantitative FLIM measurements to activities of mitochondrial metabolic pathways. NADH is a central coenzyme that binds to enzymes and facilitates redox reactions by acting as an electron carrier. There are two categories of enzymes associated with NADH redox reactions, which together form a redox cycle: oxidases that oxidize NADH to NAD$^+$ and reductases that reduce NAD$^+$ to NADH. The major NADH oxidase in mitochondria is complex I of ETC for mammalian cells. There are many NADH reductases in mitochondria because NADH can be reduced through different pathways depending on the energy substrate. These pathways include the TCA cycle, fatty acid catabolism via beta oxidation, amino acid catabolism such as glutaminolysis and the malate-aspartate shuttle (*Salway, 2017*). A comprehensive NADH redox model will include all the oxidases and reductases. For generality, we consider N oxidases and M reductases.

For convenience, we introduced a reduced notation to describe models of the enzyme kinetics of these oxidases and reductases. We began by illustrating our reduced notation using reversible Michaelis-Menten kinetics as an example (*Keleti, 1986*; *Miller and Alberty, 2002*; *Smith, 1992*). The conventional, full notation for these kinetics (*Figure 3a*, left) explicitly displays all chemical species that are modeled in this reaction scheme — free NADH, $\mathrm{NADH_f}$, free enzyme, $\mathrm{Ox}_i$, free NAD$^+$, $\mathrm{NAD_f^+}$, and NADH bound to the enzyme — as well as the forward and reverse reaction rates — $k_{-1}$, $k_1$, $k_{-2}$, and $k_2$ Our reduced notation for reversible Michaelis-Menten kinetics (*Figure 3a*, right) is an alternative way of representing the same mathematical model. In this reduced notation, only free NADH, free

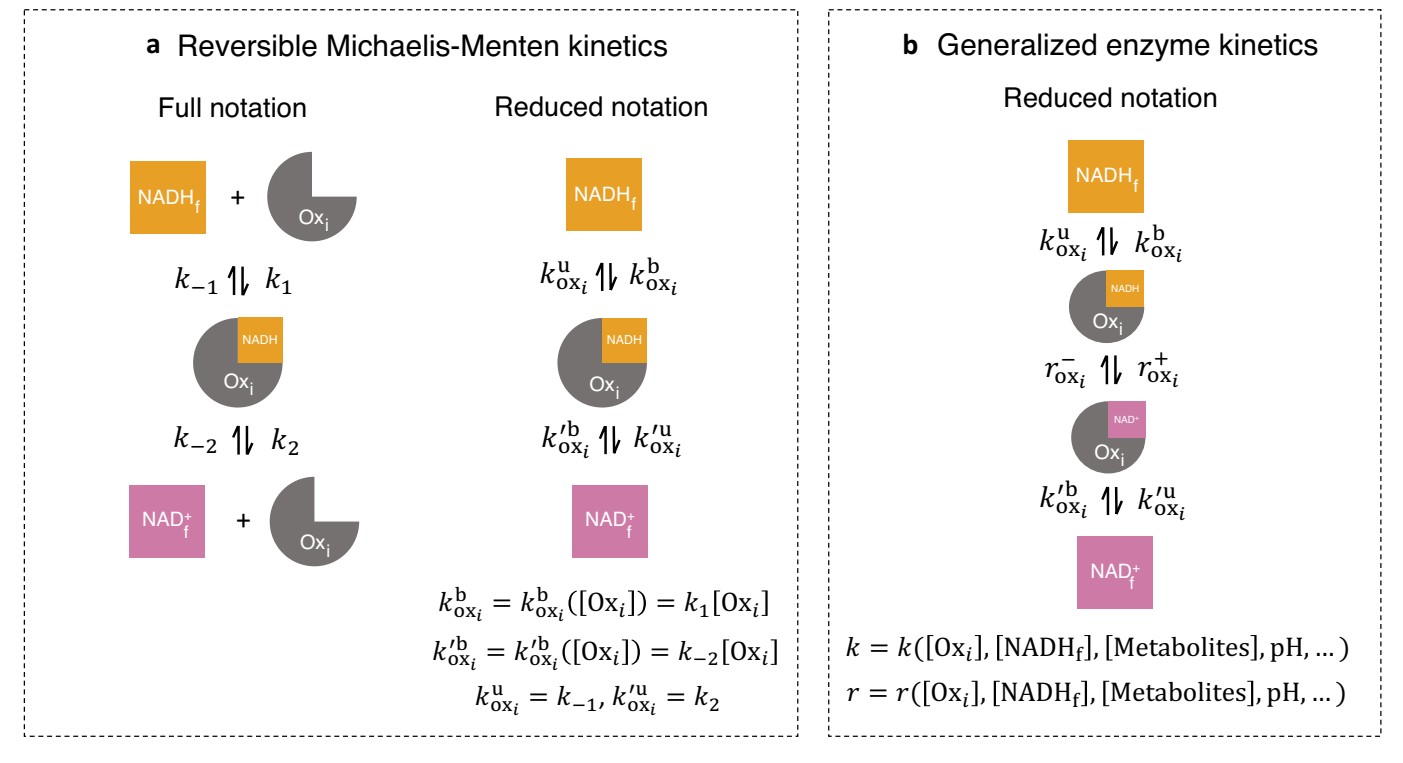

**Figure 3.** Generalized enzyme kinetics with reduced notation. (**a**) (left) Full notation for reversible Michaelis-Menten kinetics. (right) A mathematically equivalent reduced notation, in which the free enzyme concentration, is incorporated into the binding rates. (**b**) Generalized enzyme kinetics where all kinetic rates are general functions of enzyme and metabolite concentrations and other factors.

$NAD^+$, and NADH bound to the enzyme are explicitly shown, while the free enzyme concentration is only represented as entering through the effective binding rates $k^b_{ox_i}$ and $k'^b_{ox_i}$. The conventional, full notation, and the reduced notation are alternative ways of representing the same mathematical model, but the reduced notation is convenient to use in the derivation that follows (see Appendix 2).

We next introduced a generalized enzyme kinetics using our reduced notation (**Figure 3b**), which contains not only free NADH, free $NAD^+$, and NADH bound to the enzyme, but also $NAD^+$ bound to the enzyme, and the reaction rates for oxidation and reduction of the bound coenzymes. In this reduced notation, all the binding and unbinding rates, and the reaction rates, can be functions of metabolite concentrations, protein concentrations, and other factors such as pH and membrane potential. As in the reversible Michaelis-Menten kinetics example, these rates can depend on the concentration of the free enzyme itself. This dependency on free enzyme concentration can be nonlinear, as could occur if the enzyme oligomerizes. Furthermore, the rates may depend on the concentration of free NADH, free $NAD^+$, and the enzyme complexes. Thus, while the reduced notation for the generalized enzyme might appear to describe a first-order reaction, it can actually be used to represent reactions of any order, with arbitrary, nonlinear dependencies on the concentration of the enzyme itself, as well as arbitrary, nonlinear dependencies on other factors. In order to model the dynamics of enzymes described by such generalized kinetics, it is necessary to specify the functional form of all the rates, as well as specify mathematical models for all the variables that enter these rates (i.e., free enzyme concentration, membrane potential, pH, etc.) (Appendix 2). However, in what follows, we will derive results that hold true, irrespective of the functional form of the rates or the presence of additional, implicit variables. Thus, remarkably, these quantitative predictions are valid for enzyme kinetics of any order, with arbitrary nonlinearities in the rates.

To begin our derivation, we note that under this generalized enzyme kinetics (**Figure 3b**), the net flux through the $i$th oxidase at steady-state is:

$$J_{\text{ox}_i} \equiv r^+_{\text{ox}_i} \left[ \text{NADH} \cdot \text{Ox}_i \right] - r^-_{\text{ox}_i} \left[ \text{NAD}^+ \cdot \text{Ox}_i \right] = k^{\text{b}}_{\text{ox}_i} \left[ \text{NADH}_{\text{f}} \right] - k^{\text{u}}_{\text{ox}_i} \left[ \text{NADH} \cdot \text{Ox}_i \right], \qquad (3)$$

where $\left[ \text{NADH} \cdot \text{Ox}_i \right]$, $\left[ \text{NAD}^+ \cdot \text{Ox}_i \right]$, and $\left[ \text{NADH}_{\text{f}} \right]$ are the concentrations of the $i$th oxidase-bound NADH, NAD$^+$, and free NADH, respectively. $r^+_{\text{ox}_i}$ and $r^-_{\text{ox}_i}$ are the forward and reverse oxidation rates. $k^{\text{b}}_{\text{ox}_i}$ and $k^{\text{u}}_{\text{ox}_i}$ are the binding and unbinding rates. The second equality in *Equation 3* results from the steady-state condition, where the net binding and unbinding flux equals the net oxidation flux.

We next considered a redox cycle between NADH and NAD$^+$ with multiple oxidases and reductases. To account for all possible NADH redox pathways, we developed a detailed NADH redox model with N oxidases and M reductases described by the generalized enzyme kinetics (*Figure 4a* and *Figure 4—figure supplement 1*). In this model, NADH and NAD$^+$ can bind and unbind to each oxidase and reductase. Once bound, NADH can be reversibly oxidized to NAD$^+$ by the oxidases, and NAD$^+$ can be reversibly reduced to NADH by the reductases, forming a redox cycle. The functional dependencies of the binding and unbinding rates, and the reaction rates, can be different for each oxidase and reductase, and each of these rates can be nonlinear functions of free enzyme concentrations, NADH concentration, and other factors such as pH and membrane potential. Modeling the dynamics of this redox cycle requires specifying the precise number of oxidases and reductases, the functional forms of the rates, and mathematical models for all the variables the rates implicitly depend on. However, we will show that quantitative predictions regarding the interpretation of FLIM measurements can be made that generally hold, independent of these modeling choices.

FLIM cannot resolve the association of NADH with individual enzymes in cells, but rather, provides quantitative information on the global states of bound and free NADH. Thus, to facilitate comparison to FLIM experiments, we coarse-grained the detailed redox model by mapping all N oxidases into a single effective oxidase and all M reductases into a single effective reductase (*Figure 4b* and Appendix 3). This coarse-graining is mathematically exact and involves no approximations or assumptions.

In the coarse-grained redox model, NADH can be bound to the effective oxidase, $\text{NADH} \cdot \text{Ox}$, bound to the effective reductase, $\text{NADH} \cdot \text{Re}$, or can be free, $\text{NADH}_{\text{f}}$. Hence, the concentration of NADH bound to all enzymes is, $\left[ \text{NADH}_{\text{b}} \right] = \left[ \text{NADH} \cdot \text{Ox} \right] + \left[ \text{NADH} \cdot \text{Re} \right]$, and the total concentration of NADH is, $\left[ \text{NADH} \right] = \left[ \text{NADH}_{\text{b}} \right] + \left[ \text{NADH}_{\text{f}} \right]$. The kinetics of the effective oxidase and reductase are represented by the coarse-grained forward, $r^+_{\text{ox}}$, and reverse, $r^-_{\text{ox}}$, oxidation rates, and the forward, $r^+_{\text{re}}$, and reverse, $r^-_{\text{re}}$, reduction rates. The global flux through all the oxidases in the detailed redox model equals the global flux through the coarse-grained oxidase, which at steady-state is:

$$J_{\text{ox}} \equiv \sum_{i=1}^{\text{N}} J_{\text{ox}_i} = r^+_{\text{ox}} \left[ \text{NADH} \cdot \text{Ox} \right] - r^-_{\text{ox}} \left[ \text{NAD}^+ \cdot \text{Ox} \right] = k^{\text{b}}_{\text{ox}} \left[ \text{NADH}_{\text{f}} \right] - k^{\text{u}}_{\text{ox}} \left[ \text{NADH} \cdot \text{Ox} \right], \qquad (4)$$

where $k^{\text{b}}_{\text{ox}}$ is the rate that free NADH binds the effective oxidase, $k^{\text{u}}_{\text{ox}}$ is the rate that NADH unbinds the effective oxidase, and the last equality results because the coarse-grained redox loop is a linear pathway so the global oxidative flux must equal the global binding and unbinding flux at steady-state. The conservation of global flux explicitly relates the effective binding and unbinding rates and the reaction rates of the coarse-grained model to those of the detailed model (Appendix 3, *Figure 4—figure supplement 1*). The binding and unbinding kinetics of NADH and NAD$^+$ to the effective oxidase and reductase are described by eight coarse-grained binding and unbinding rates (*Figure 4b*). The coarse-grained reaction rates and binding and unbinding rates can be arbitrary functions of metabolite concentrations, enzyme concentrations, and other factors (i.e., pH, membrane potential, etc.). These effective rates can even be functions of $\left[ \text{NADH}_{\text{f}} \right]$, $\left[ \text{NAD}^+_{\text{f}} \right]$, and the concentration of other variables, and thus can include reactions of arbitrary order. Hence, this coarse-grained model is a generic model of NADH redox reactions. Fully specifying this model would require explicitly choosing the functional form of all the rates and incorporating additional equations to describe the dynamics of all the implicit variables that the rates depend on (Appendix 2). We next demonstrate that quantitative predictions regarding the interpretation of FLIM measurements of NADH can be made that are valid irrespective of the form of the rates or the presence of implicit variables.

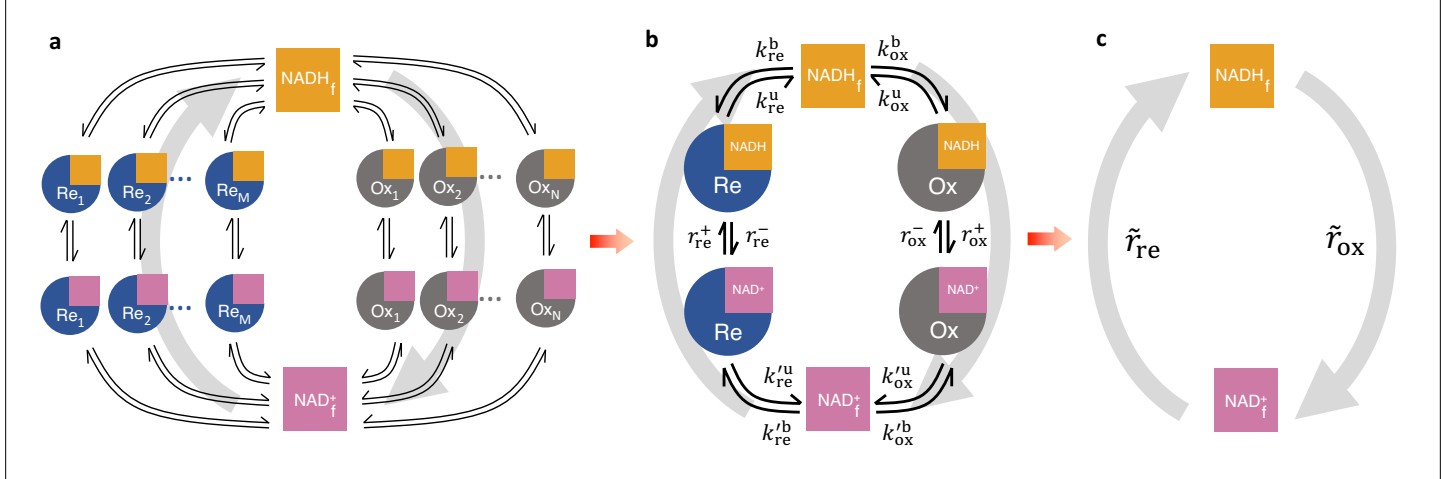

**Figure 4.** Coarse-graining the NADH redox model. (**a**) Schematic of the detailed NADH redox model. We consider all possible NADH redox pathways by modeling N oxidases (Ox) and M reductases (Re). Free NADH, $\mathbf{NADH_f}$, and free NAD$^+$, $\mathbf{NAD_f^+}$, can bind and unbind with each oxidase and reductase. Once bound, NADH can be oxidized reversibly to NAD$^+$ by the oxidases, and NAD$^+$ can be reduced reversibly to NADH by the reductases, forming a redox cycle. Gray arrows represent the total fluxes through all oxidases and reductases of the redox cycle. (**b**) Coarse-grained NADH redox model. All oxidases and reductases are coarse-grained into a single effective oxidase and reductase, respectively. $r_{ox}^+$ and $r_{ox}^-$ are the coarse-grained forward and reverse oxidation rates of the oxidase; $r_{re}^+$ and $r_{re}^-$ are the coarse-grained forward and reverse reduction rates of the reductase. $k_{ox}^b$, $k_{ox}^u$, $k_{re}^b$, $k_{re}^u$ and $k_{ox}'^b$, $k_{ox}'^u$, $k_{re}'^b$, $k_{re}'^u$ are the coarse-grained binding and unbinding rates of NADH and NAD$^+$, respectively, to the oxidase and reductase. (**c**) At steady-state, all the kinetics of the model can be further coarse-grained into the turnover rate of free NADH, $\tilde{r}_{ox}$, and the turnover rate of free NAD$^+$, $\tilde{r}_{re}$, characterizing the two branches of the cycle.

The online version of this article includes the following figure supplement(s) for figure 4:

**Figure supplement 1.** The detailed NADH redox model.

## Accurately predicting ETC flux from FLIM of NADH using the NADH redox model

At steady-state, the model can be further coarse-grained, without approximation, to consider only free NADH, with a turnover rate of $\tilde{r}_{ox}$, and free NAD$^+$, with a turnover rate of $\tilde{r}_{re}$ (*Figure 4c*). Our key prediction is that the steady-state global oxidative flux of NADH is (Appendix 4):

$$J_{ox} = k_{ox}^b \left[\text{NADH}_f\right] - k_{ox}^u \left[\text{NADH} \cdot \text{Ox}\right] = \tilde{r}_{ox} \left[\text{NADH}_f\right], \tag{5a}$$

where

$$\tilde{r}_{ox} = \alpha \left(\beta - \beta_{eq}\right), \tag{5b}$$

and

$$\beta = \frac{f}{1-f}. \tag{5c}$$

This prediction results from the steady-state assumption where the net binding and unbinding flux of NADH from the oxidase balances the net oxidative flux through the oxidase (*Equation 4* and Appendix 4). The turnover rate of free NADH, $\tilde{r}_{ox}$, is proportional to the difference between the NADH bound ratio $\beta$, that is, the ratio between bound and free NADH concentrations, and the equilibrium NADH bound ratio, $\beta_{eq}$ (i.e., what the bound ratio would be if the global oxidative flux is zero). $\beta_{eq}$ and the prefactor $\alpha$ are independent of the reaction rates of the oxidase and reductase and can be explicitly related to the binding and unbinding rates of the coarse-grained model (Appendix 4, *Equation S43 and S45*).

In mitochondria, the major NADH oxidation pathway is the ETC. Thus, *Equations 5a-c* predict that there is a direct connection between quantities that can be measured by FLIM of NADH in mitochondria (i.e., $\beta$ and $\left[\text{NADH}_f\right]$) and the flux through the ETC (i.e., $J_{ox}$). *Equations 5a-c* suggest a procedure for using FLIM to infer flux through the ETC: if a condition can be found under which there is no net

flux through the ETC, then $\beta_{eq}$ can be measured with FLIM. Once $\beta_{eq}$ is known, then subsequent FLIM measurements of $\beta$ allows $\tilde{r}_{ox}$, and hence $J_{ox}$, to be inferred (up to a constant of proportionality $\alpha$) (Appendix 5).

*Equations 5a-c* are valid irrespective of the functional forms of the rate laws, which may have nonlinear dependencies on metabolite concentrations, enzyme concentrations, and other factors. While *Equation 5a* seems to imply first-order kinetics in $[\mathrm{NADH_f}]$, the rates can also be arbitrary functions of $[\mathrm{NADH_f}]$, so *Equations 5a–c* hold for kinetics of any order. *Equations 5a-c* are also applicable if the rates depend on additional variables that have their own dynamical equations (as long as the system is at steady-state): as an example, Appendix 7 shows that *Equations 5a-c* result when the N oxidases and M reductases are each described by reversible Michaelis-Menten kinetics, a model in which the rates depend on the concentration of free enzymes (which is a dynamical variable). More generally, if detailed biophysical models of the NADH oxidases are available, then parameters of these models can be explicitly mapped to the coarse-grained parameters of the NADH redox model. Appendix 7 and *Appendix 7—table 1* contain mappings between the coarse-grained model and a number of previously proposed detailed biophysical models of NADH oxidation in the ETC (*Beard, 2005*; *Korzeniewski and Zoladz, 2001*; *Hill, 1977*; *Jin and Bethke, 2002*; *Chang et al., 2011*). However, since *Equations 5a-c* are valid for a broad set of models, they can be used for flux inference without the need to specify the functional form of the rates or the variables they depend on. This is because the rates are coarse-grained into two effective parameters $\alpha$ and $\beta_{eq}$, which can be experimentally determined. This generality results from the steady-state assumption and the topology of the reactions resulting in the net binding and unbinding flux of NADH from the oxidase balancing the net oxidative flux. Thus, *Equations 5a–c* provide a general procedure to infer the ETC flux from FLIM measurements of NADH in mitochondria.

We applied this procedure to analyze our oxygen drop experiments (*Figure 1*) by assuming that there was no net flux through the ETC at the lowest oxygen level achieved for each oocyte (implying that the measured value of $\beta$ at that oxygen concentration corresponds to $\beta_{eq}$ for that oocyte). We also assumed that $\alpha$ and $\beta_{eq}$ do not change with oxygen levels, which is reasonable since, as noted above, they are independent of the reaction rates of the oxidase and reductase. The measured value of $\beta_{eq}$ allowed us to obtain a prediction for $J_{ox}$ as a function of oxygen concentration for the oocytes (*Figure 5a*). To test these predictions, we directly determined $J_{ox}$ as a function of oxygen concentration by measuring the OCR of the oocytes using a nanorespirometer (*Lopes et al., 2005*) (Materials and methods). The direct measurements of $J_{ox}$ from OCR quantitatively agree with the predictions of $J_{ox}$ from FLIM for all oxygen concentrations (*Figure 5a*), strongly arguing for the validity of the model and the inference procedure. This agreement supports the assumption that $\alpha$ and $\beta_{eq}$ are independent of oxygen levels.

So far, we have inferred the ETC flux up to a constant of proportionality $\alpha$, allowing the relative changes of ETC flux to be inferred from FLIM of NADH. $\alpha$ cannot be determined by FLIM alone. If an absolute measurement of the ETC flux can be obtained at one condition, then $\alpha$ can be calibrated to predict absolute ETC fluxes for other conditions. OCR measurement provides a means to calibrate $\alpha$ (Appendix 5, *Equation S48*). We used oocytes cultured in AKSOM media at 50±2 µM oxygen as a reference state, which, from our OCR measurements yielded $J_{ox} = 56.6 \pm 2\ \mu M/s$ (SEM) and hence a constant of proportionality of $\alpha = 5.4 \pm 0.2\ \mathrm{s}^{-1}$. Using this value of $\alpha$, we can predict absolute values of $J_{ox}$ under various perturbations assuming $\alpha$ remains a constant. We note that $J_{ox}$ is a flux density with units of concentration per second, an intensive quantity that does not depend on the mitochondrial volume. Multiplying $J_{ox}$ by the volume of mitochondria in an oocyte gives the total ETC flux, proportional to OCR, in that oocyte. In all subsequent discussions, ETC flux refers to flux density unless otherwise noted.

We next applied the inference procedure and a constant of $\alpha = 5.4 \pm 0.2\ \mathrm{s}^{-1}$ to analyze the experiments of oxamate, FCCP, rotenone and oligomycin perturbations (*Figure 2*). We dropped oxygen levels to determine $\beta_{eq}$ in the presence of oxamate (*Figure 5—figure supplement 1h*) and applied *Equations 5a-c* to infer the impact of oxamate on $J_{ox}$ at 50 µM oxygen (i.e., control levels of oxygen). Surprisingly, while the addition of oxamate greatly impacts FLIM parameters, including a 29±2% (SEM) decrease in intensity and a 10±3% increase in bound ratio (*Figure 2a*), this procedure revealed that the predicted ETC flux with oxamate ($J_{ox} = 55.2 \pm 3.2\ \mu M/s$) is the same as that without oxamate ($J_{ox} = 55.4 \pm 1.9\ \mu M/s$) (*Figure 5b*; *p*=0.95), which was confirmed by direct measurements of oocytes'

OCR that yielded $J_{ox} = 55.4 \pm 1.5 \,\mu M/s$ and $J_{ox} = 54.9 \pm 0.7 \,\mu M/s$ before and after the addition of oxamate, respectively (*Figure 5b*; *p*=0.85). We next analyzed the FCCP experiment. We obtained $\beta_{eq}$ by dropping oxygen in the presence of FCCP (*Figure 5—figure supplement 1h*) and applied *Equations 5a-c* to infer the impact of FCCP on $J_{ox}$ at 50 μM oxygen. We predicted that FCCP increased the flux to $J_{ox} = 67.7 \pm 1.5 \,\mu M/s$, which was confirmed by the directly measured $J_{ox} = 74.0 \pm 3.7 \,\mu M/s$ from OCR (*Figure 5b*; *p*=0.30). Following the same FLIM based inference procedures, we predicted that the addition of rotenone and oligomycin reduced the fluxes to $J_{ox} = 16.7 \pm 1.4 \,\mu M/s$ and $J_{ox} = 15.7 \pm 1.3 \,\mu M/s$, respectively, which was again confirmed by corresponding direct measurements of OCR that yielded $J_{ox} = 11.1 \pm 0.4 \,\mu M/s$ and $J_{ox} = 22.3 \pm 0.6 \,\mu M/s$ (*Figure 5b*; *p*=0.31 and *p*=0.17). The quantitative agreement between predicted fluxes from FLIM and directly measured fluxes from OCR under a variety of conditions (i.e., varying oxygen tension, sodium oxamate, FCCP, rotenone, and oligomycin) demonstrates that *Equations 5a-c* can be successfully used to infer flux through the ETC in mouse oocytes. This agreement also supports the assumption that $\alpha$ is a constant across these different perturbations.

The work described above used the relation $\tilde{r}_{ox} = \alpha \left( \beta - \beta_{eq} \right)$ to predict the flux through the ETC from FLIM measurements. We next show that the model also predicts a relationship between $\tilde{r}_{ox}$ and the fluorescence lifetime of enzyme-bound NADH, $\tau_l$, in mitochondria. This provides a second means to use the model to infer $\tilde{r}_{ox}$, and hence $J_{ox}$, from FLIM of NADH. Specifically, we assumed that NADH bound to the oxidases have a different average lifetime, $\tau_{ox}$, than NADH bound to the reductases, $\tau_{re}$, which is reasonable because NADH bound to different enzymes do exhibit different fluorescence lifetimes (*Sharick et al., 2018*). This assumption implies that the experimentally measured long lifetime of NADH in mitochondria, $\tau_l$, is a weighted sum of these two lifetimes,

$$\tau_l = \tau_{ox} \frac{[\text{NADH·Ox}]}{[\text{NADH·Ox}] + [\text{NADH·Re}]} + \tau_{re} \frac{[\text{NADH·Re}]}{[\text{NADH·Ox}] + [\text{NADH·Re}]}. \tag{6}$$

Using the coarse-grained NADH redox model at steady-state, *Equation 6* leads to a non-trivial prediction that $\tau_l$ is linearly related to $1/\beta$ (Appendix 5):

$$\tau_l = A \frac{1}{\beta} + B, \tag{7}$$

where the slope $A$ and offset $B$ can be explicitly related to $\tau_{ox}$, $\tau_{re}$, and the coarse-grained binding and unbinding rates. Such a linear relationship is indeed observed in individual oocytes subject to oxygen drops (*Figure 6a* and *Figure 6—figure supplement 1*), supporting the assumptions of the model. Combining *Equations 7* and *5b* leads to a predicted relationship between $\tilde{r}_{ox}$ and NADH long fluorescence lifetime (Appendix 5):

$$\tilde{r}_{ox} = \alpha \frac{A}{\tau_{eq} - B} \left( \frac{\tau_{eq} - \tau_l}{\tau_l - B} \right), \tag{8}$$

where $\tau_{eq}$ is the equilibrium NADH long fluorescence lifetime, that is, the value of the long lifetime when the global oxidative flux is zero. This provides a second means to infer $\tilde{r}_{ox}$ from FLIM measurements: dropping oxygen and plotting the relationship between $\tau_l$ and $1/\beta$ provides a means to measure $A$ and $B$ from *Equation 7*, while $\tau_{eq}$ can be obtained from the NADH long fluorescence lifetime obtained at the lowest oxygen level. Once $A$, $B$, and $\tau_{eq}$ are known, $\tilde{r}_{ox}$ can be inferred solely from NADH long fluorescence lifetime $\tau_l$, using *Equation 8*.

We next used the lifetime method (*Equation 8*) and the bound ratio method (*Equation 5b*) to separately infer $\tilde{r}_{ox}$ in oocytes subject to a wide variety of conditions (varying oxygen levels, with oxamate, FCCP, rotenone, and oligomycin). We obtained $A$, $B$, $\beta_{eq}$, and $\tau_{eq}$ for these different conditions (*Figure 5—figure supplement 1* and *Figure 6—figure supplement 1*), and used the two different methods to provide two independent measures of $\tilde{r}_{ox}$ (assuming $\alpha$ is constant across all conditions). The predictions of $\tilde{r}_{ox}$ from these two methods quantitatively agree under all conditions (*Figure 6b*, *p*=0.73), which is a strong self-consistency check that further supports the use of the model to infer ETC flux from FLIM measurements of NADH.

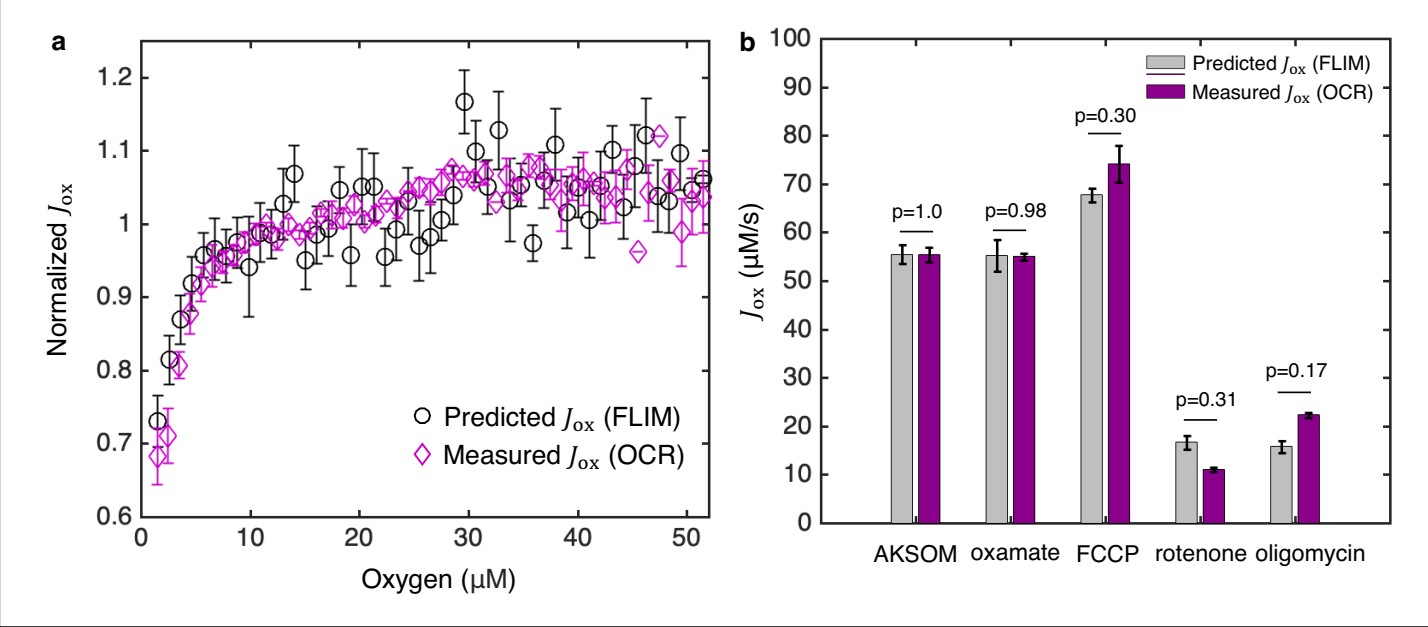

**Figure 5.** Coarse-grained NADH redox model enables accurate prediction of flux through the ETC from FLIM measurements of NADH. (**a**) Predicted flux through the ETC, $J_{ox}$, from the FLIM of NADH (n=68 oocytes) agrees quantitatively with $J_{ox}$ from oxygen consumption rate (OCR) measurements (N=3 measurements) for all oxygen concentrations. $J_{ox}$ is normalized by its value at 50 μM oxygen. (**b**) Predicted $J_{ox}$ from FLIM and measured $J_{ox}$ from OCR for AKSOM (n=68, N=4) and with perturbations of 9 mM oxamate (n=20, N=2), 5 μM FCCP (n=31, N=2), 5 μM rotenone (n=28, N=2) and 5 μM oligomycin (n=37, N=3). Predicted $J_{ox}$ agrees with measured $J_{ox}$ in all cases. n denotes number of oocytes for single-oocyte FLIM measurements. N denotes number of replicates for batch oocytes OCR measurements. Each batch contains 10–15 oocytes. p values are calculated from two-sided two-sample t-test. Error bars denote standard error of the mean across individual oocytes for FLIM measurements and across batches of oocytes for OCR measurements. ETC, electron transport chain; FLIM, fluorescence lifetime imaging microscopy; OCR, oxygen consumption rate.

The online version of this article includes the following source data and figure supplement(s) for figure 5:

**Source data 1.** Excel spreadsheet of single-oocyte FLIM data and batch OCR data used for *Figure 5b*.

**Figure supplement 1.** FLIM measurements of NADH in mitochondria under different biochemical perturbations.

**Figure supplement 1—source data 1.** Excel spreadsheet of single-oocyte FLIM data used for *Figure 5—figure supplement 1a-h*.

## The NADH redox model enables accurate prediction of ETC flux in human tissue culture cells

After thoroughly testing the NADH redox model and the inference procedure in mouse oocytes, we next investigated if it can be used in other cell types. We chose human tissue culture cells for this purpose, since they are widely used as model systems to study metabolic dysfunctions in human diseases including cancer (*Vander Heiden et al., 2009*) and neuropathology (*Lin and Beal, 2006*).

While mouse oocytes have a negligible level of NADPH compared to NADH (*Bustamante et al., 2017*), the concentrations of NADH and NADPH are similar in tissue culture cells (10–100 μM averaged over the whole cell) (*Lu et al., 2018*; *Park et al., 2016*; *Blacker et al., 2014*). Since NADPH and NADH have overlapping fluorescent spectra (*Patterson et al., 2000*), the presence of NADPH may complicate the interpretation of FLIM experiments. Thus, we investigated the impact of background fluorescence, such as from NADPH, on the flux inference procedure. If the background fluorescence does not change with the perturbations under study, then it can be treated as an additive offset that systematically makes the measured concentrations of free and bound NADH different from their actual values. In this case, a derivation in Appendix 5 demonstrates that the background fluorescence can be incorporated into the equilibrium bound ratio $\beta_{eq}$ and does not impact the flux inference procedures. In other words, if the modified $\beta_{eq}$ can be reliably determined, then the measured concentrations of free and bound fluorescent species can be used in place of the true values of NADH in *Equations 5a-c* to infer the ETC flux. An alternative possibility is that the background fluorescence does change with the perturbations under study, but in a manner that is proportional to the change in NADH. In this case, the background fluorescence can be incorporated into the equilibrium bound ratio $\alpha$ and, once

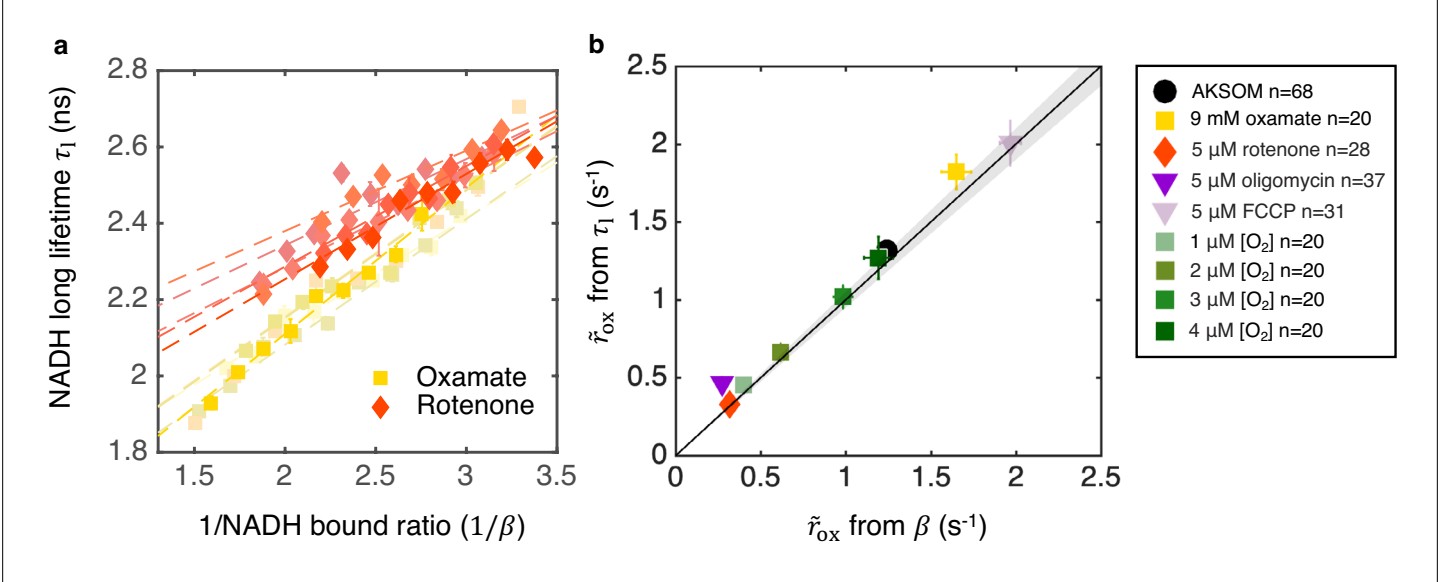

**Figure 6.** Coarse-grained NADH redox model self-consistently predicts NADH turnover rate from bound ratio and long fluorescence lifetime. (**a**) NADH long lifetime, $\tau_l$, is linearly related to the inverse of NADH bound ratio, $1/\beta$, from the oxygen drop experiment of individual oocytes treated with oxamate and rotenone (results from five representative oocytes are shown for each condition). Each shade corresponds to results from an individual oocyte (symbols are experimental measurements and dashed lines are linear fits). (**b**) NADH turnover rate $\tilde{r}_{ox}$ obtained from NADH long lifetime ($\tau_l$) using *Equation 8* agrees quantitatively with that from NADH bound ratio ($\beta$), obtained from *Equation 5b*, across all perturbations ($p=0.73$). The solid line denotes where $\tilde{r}_{ox}$ from lifetime equals that from bound ratio, the gray region denotes ±5% variation from equality. Error bars represent standard error of the mean (s.e.m) across different oocytes. $p$ value is calculated from Student's $t$-test.

The online version of this article includes the following source data and figure supplement(s) for figure 6:

**Source data 1.** Excel spreadsheet of single-oocyte FLIM data used for *Figure 6b*.

**Figure supplement 1.** NADH long fluorescence lifetime $\tau_l$ is linearly related to the inverse of the NADH bound ratio $1/\beta$.

**Figure supplement 1—source data 1.** Excel spreadsheet of single-oocyte FLIM data used for *Figure 6—figure supplement 1f,g*.

more, does not impact the flux inference procedures (Appendix 5). If the background fluorescence changes in some more complicated manner, then the inference procedure may no longer be valid. Thus, depending on the behavior of NADPH, it either might or might not interfere with the inference procedure: no impact if NADPH is either constant or proportional to changes in NADH, a possible impact otherwise. Therefore, the validity of the inference procedure in the presence of significant NADPH fluorescence must be established empirically.

We next tested the inference procedures experimentally in hTERT-RPE1 (hTERT-immortalized retinal pigment epithelial cell line) tissue culture cells. We started by exploring the impact of metabolic perturbations on mitochondrial NAD(P)H: the combined signal from NADH and NADPH (which are indistinguishable) from mitochondria. We first cultured the cells in DMEM with 10 mM galactose (Materials and methods). We then inhibited complex I of the ETC by adding 8 μM of rotenone to the media. This resulted in a significant increase of mitochondrial NAD(P)H intensity (*Figure 7a*, upper). We segmented mitochondria using a machine learning-based algorithm from the intensity images of NAD(P)H, and fitted the fluorescence decay curves of mitochondrial NAD(P)H to obtain changes in FLIM parameters (Materials and methods). All FLIM parameters displayed significant changes (*Figure 7a*, lower, $p<0.001$, and *Figure 7—figure supplement 1*). We next uncoupled proton translocation from ATP synthesis by adding 3.5 μM CCCP to the media. This led to a decrease of NAD(P)H intensity in the mitochondria (*Figure 7b*, upper) and significant changes in NAD(P)H bound ratio and short lifetime, but in opposite directions as compared to rotenone perturbation (*Figure 7b*, lower, $p<0.01$, and *Figure 7—figure supplement 1*). Finally, we perturbed the nutrient conditions by culturing the cells in DMEM with 10 mM glucose. FLIM imaging revealed an increase of mitochondrial NAD(P)H intensity (*Figure 7c*, upper) and significant changes in all FLIM parameters as compared to the galactose condition (*Figure 7c*, lower, $p<0.001$, and *Figure 7—figure supplement 1*).

Inference of the ETC flux from FLIM measurements requires a measurement of $\beta_{eq}$. Since rotenone is known to drastically decrease the OCR of hTERT-RPE1 cells to near zero (*MacVicar and Lane, 2014*), we used the NAD(P)H bound ratio measured in the presence of rotenone as $\beta_{eq}$. Different values of $\beta_{eq}$ were obtained for glucose and galactose conditions by adding 8 μM of rotenone to each condition (*Figure 7—figure supplement 1*). We next calculated the concentrations of free NAD(P)H, $[\mathrm{NAD(P)H_f}]$, from the FLIM parameters using *Equation 2a*. $[\mathrm{NAD(P)H_f}]$ displayed significant changes for all perturbations (*Figure 7d*). Using *Equation 5b* and assuming $\alpha$ is a constant, we calculated the NAD(P)H turnover rate, $\tilde{r}_{ox}$, from the FLIM measurements and $\beta_{eq}$. $\tilde{r}_{ox}$ changed significantly for all perturbations (*Figure 7e*). Multiplying $\tilde{r}_{ox}$ and $[\mathrm{NAD(P)H_f}]$, we obtained the predicted ETC flux, $J_{ox}$, which increased under FCCP, decreased under glucose and reduced to zero under rotenone (*Figure 7f*).

To test the model predictions, we compared the predicted ETC flux with previous direct OCR measurements of the same cell type that we used, under the same conditions (*MacVicar and Lane, 2014*). Remarkably, the predicted changes in ETC fluxes are in quantitative agreement with the directly measured OCR across all conditions as estimated from Figure 1A of *MacVicar and Lane, 2014*: CCCP is predicted to increase the ETC flux by 14±3% (SEM), in agreement with the 18±21% increase from OCR measurement ($p$=0.80); Glucose is predicted to decrease ETC flux by 33±3%, in agreement with the 46±9% decrease from OCR measurements ($p$=0.30), shifting metabolism from oxidative phosphorylation to aerobic glycolysis. Since we used $\beta$ from rotenone treatment as $\beta_{eq}$, the predicted decrease in ETC flux after the addition of rotenone is 101±2%, which is in agreement with the 82±2% decrease from OCR measurement ($p$=0.28). This quantitative agreement between predicted ETC fluxes and measured OCR across all perturbations demonstrated the applicability of the NADH redox model and the flux inference procedures to tissue culture cells, even though they contain substantial levels of NADPH.

## Homeostasis of ETC flux in mouse oocytes: perturbations of nutrient supply and energy demand impact NADH metabolic state but do not impact ETC flux

Having established the validity of the NADH redox model and the associated flux inference procedures, we next applied it to study energy metabolism in mouse oocytes. We began by investigating the processes that determine the ETC flux in MII mouse oocytes. Mitochondrial-based energy metabolism can be viewed as primarily consisting of three coupled cycles: the NADH/NAD$^+$ redox cycle (which our NADH redox model describes), the proton pumping/dissipation cycle, and the ATP/ADP production/consumption cycle (*Figure 8a*). At the most upstream portion of this pathway, the reduction of NAD$^+$ to NADH is powered by a supply of nutrients, while at the most downstream portion, energy-demanding cellular processes hydrolyze ATP to ADP. To test whether nutrient supply and energy demand set ETC flux, we investigated the effect of perturbing these processes. To perturb supply, we first varied the concentration of pyruvate in the media from 181 μM (which is standard for AKSOM) to either 18.1 μM or 1.81 mM, and observed significant changes in NADH intensity and FLIM parameters (*Figure 8b*, left), demonstrating that the NADH metabolic state is altered. To perturb demand, we began by adding 10 μM nocodazole to the media, which disassembled the meiotic spindle, an energy user, and resulted in significant changes in NADH FLIM parameters (*Figure 8b*, center). Similarly, the addition of 10 μM latrunculin A disassembled the actin cortex and also produced significant changes in NADH FLIM parameters (*Figure 8b*, right).

We next performed additional perturbations of nutrient supply, inhibiting the conversion of lactate to pyruvate by LDH (with 9 mM oxamate) and inhibiting the malate-aspartate shuttle (with 11 mM AOA). We performed additional perturbations of energy demand by inhibiting protein synthesis (with 1 mM cycloheximide) and ion homeostasis, by varying extracellular potassium concentrations from 0 mM to 15 mM, inhibiting the Na$^+$/K$^+$ pump (with 2 mM ouabain), and adding an ionophore (10 μM gramicidin). All perturbations resulted in significant changes in NADH FLIM parameters (*Figure 8—figure supplement 1*), showing that NADH metabolic state is generally impacted by varying nutrient supply and cellular energy demand. We next used the NADH redox model and the measured FLIM parameters to infer the concentration and effective turnover rate of free NADH for these perturbations. The free NADH concentrations, $[\mathrm{NADH_f}]$, and turnover rates, $\tilde{r}_{ox}$, displayed large variations across the perturbations, ranging from 33.5±1 μM (SEM) to 56.0±2.8 μM and from 1.0±0.05 s$^{-1}$ to 1.65±0.09 s$^{-1}$,

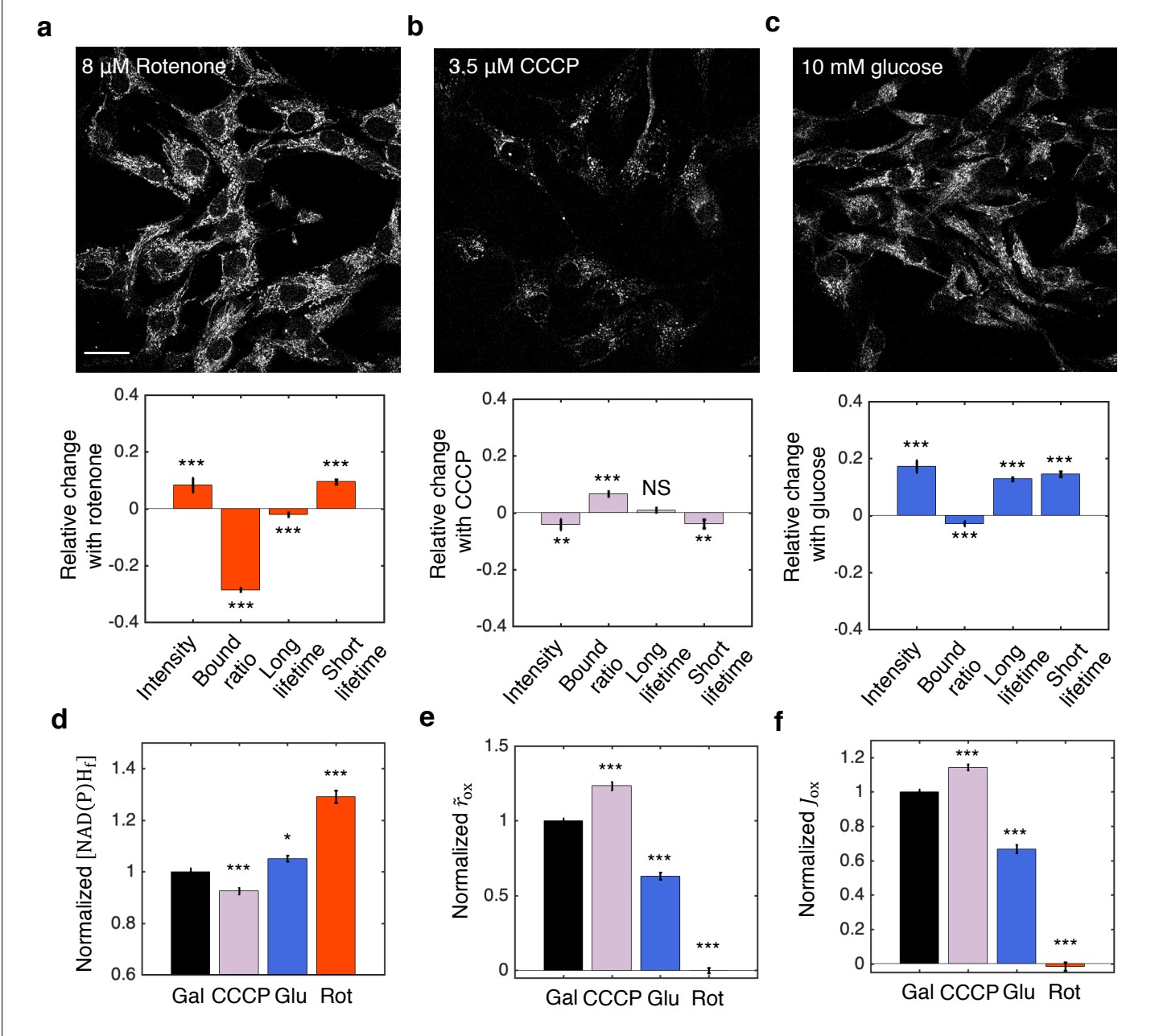

**Figure 7.** NADH redox model accurately predicts ETC flux in hTERT-RPE1 human tissue culture cells. (**a–c**) NAD(P)H intensity images (scale bar, 30 μm) and the corresponding changes of FLIM parameters in response to metabolic perturbations with the addition of 8 μM rotenone (**a**) (*N*=61), 3.5 μM CCCP (**b**) (*N*=72), and the change of nutrients from 10 mM galactose to 10 mM glucose (**c**) (*N*=77). Rotenone and CCCP are added to culturing media with 10 mM galactose (*N*=145). Measurements were taken within 30 min after the addition of the drugs. *N* specifies the number of images analyzed for each condition. A typical image contains dozens of cells as shown in (**a–c**). (**d–f**) Free NAD(P)H concentrations ($[\mathrm{NAD(P)H_f}]$) (**d**), NAD(P)H turnover rate ($\tilde{r}_{\mathrm{ox}}$) (**e**), and inferred ETC flux ($J_{\mathrm{ox}}$) (**f**) in response to CCCP, rotenone, and glucose perturbations. Student's *t*-test is performed pairwise between perturbations and the 10 mM galactose condition. \**p*<0.05, \*\**p*<0.01, \*\*\**p*<0.001. Error bars represent standard error of the mean (s.e.m) across different images. ETC, electron transport chain; FLIM, fluorescence lifetime imaging microscopy.

The online version of this article includes the following source data and figure supplement(s) for figure 7:

**Source data 1.** Excel spreadsheet of single image FLIM data used for *Figure 7a–f*.

**Figure supplement 1.** NAD(P)H FLIM parameters and TMRM measurements in response to mitochondrial inhibitors and nutrient perturbations for hTERT-RPE1 cells.

**Figure supplement 1—source data 1.** Excel spreadsheet of single image TMRM data used for *Figure 7—figure supplement 1*.

respectively (*Figure 8c*). Surprisingly, the changes in $[\mathrm{NADH_f}]$ and $\tilde{r}_{\mathrm{ox}}$ were highly anti-correlated such that the data points primarily fell within a region where the inferred ETC flux, $J_{\mathrm{ox}} = \tilde{r}_{\mathrm{ox}} [\mathrm{NADH_f}]$, is a constant 55.5 µM·s⁻¹ (*Figure 8c*, solid line, shaded region indicates 5% error). Indeed, ANOVA tests confirmed that perturbing nutrient supplies and cellular energy demand lead to no significant change in either the inferred ETC flux (*Figure 8d*, *p*=0.20) or directly measured OCR (*Figure 8e*, *p*=0.07). Thus, while nutrient supply and cellular energy demand strongly affect mitochondrial NADH redox metabolism, they do not impact ETC flux. In contrast, ETC flux is impacted by perturbing proton leak and ATP synthesis (*Figure 5*). Taken together, this suggests that the ETC flux in mouse oocytes is set by the intrinsic properties of their mitochondria, which can adjust their NADH redox metabolism to maintain a constant flux when nutrient supplies and cellular energy demand are varied. The mechanistic basis of this homeostasis of ETC flux is unclear and will be an exciting topic for future research.

## Subcellular spatial gradient of ETC flux in mouse oocytes: spatially inhomogeneous mitochondrial proton leak leads to a higher ETC flux in mitochondria closer to cell periphery

Our results presented so far were performed by averaging together FLIM measurements from all mitochondria within an oocyte. However, FLIM data is acquired with optical resolution, enabling detailed subcellular measurements. To see if there are spatial variations in FLIM measurements within individual oocytes, we computed the mean NADH fluorescence decay time for each mitochondrial pixel. The mean NADH fluorescence decay time displays a clear spatial gradient, with higher values closer to the oocyte center (*Figure 9a*).

To quantify this gradient in more detail, we partitioned mouse oocytes into equally spaced concentric regions (*Figure 9b*) and fitted the fluorescence decay curves from mitochondrial pixels within each region to obtain FLIM parameters as a function of distance from the oocyte center. NADH intensity, bound ratio, and long lifetime in mitochondria all display significant spatial gradient within oocytes (*Figure 9c*). Next, using *Equations 5a-c* and $\beta_{\mathrm{eq}}$ obtained at the lowest oxygen level, and confirming that $\beta_{\mathrm{eq}}$ is uniform within the oocyte with complete inhibition of ETC using high concentration of rotenone (*Figure 9—figure supplement 1*), we predicted the ETC flux, $J_{\mathrm{ox}}$, as a function of distance from the oocyte's center. The ETC flux displayed a strong spatial gradient within oocytes, with a higher flux closer to the cell periphery (*Figure 9d*). Note that, as described above, $J_{\mathrm{ox}}$ is actually a flux density with units of concentration per second. Thus, the measured flux gradient is not merely a reflection of variations in mitochondrial density, but instead indicates the existence of subcellular spatial heterogeneities in mitochondrial activities.

To investigate the origin of this flux gradient, we inhibited ATP synthase using 5 µM of oligomycin and repeated measurements of subcellular spatial variations in inferred fluxes. After inhibition, $J_{\mathrm{ox}}$ decreased at all locations throughout the oocytes and displayed an even more dramatic flux gradient (*Figure 9e*). If oligomycin completely blocks ATP synthase, then the remaining flux must be the result of proton leak. If it is further assumed that proton leak remains the same with and without oligomycin, then the flux due to ATP synthase in control oocytes can be determined by subtracting the flux after oligomycin inhibition (i.e., the proton leak) from the flux before inhibition. Performing this procedure throughout oocytes indicates that proton leak greatly increases in mitochondria near the periphery of oocytes, where ATP production decreases (*Figure 9f*). This implies that the subcellular gradient in ETC flux is primarily caused by a gradient in proton leak and that mitochondria near the periphery of oocytes are less active in ATP production than those in the middle of the oocyte.

We hypothesized that a gradient in proton leak would result in a gradient of mitochondrial membrane potential, with lower membrane potential closer to the cell periphery where proton leak is the greatest. To test this, we measured mitochondrial membrane potential using the membrane potential-sensitive dye TMRM, which preferentially accumulates in mitochondria with higher membrane potential (*Al-Zubaidi et al., 2019*). We observed a strong spatial gradient of the intensity of TMRM in mitochondria within oocytes, with dimmer mitochondria near the cell periphery (*Figure 9g and h*), indicating that mitochondria near the periphery of the oocyte have a lower membrane potential. This result is robust to locally normalizing TMRM intensity by mitochondrial mass using a membrane potential insensitive dye (Mitotracker Red FM), or using an alternative membrane potential-sensitive dye, JC-1 (*Figure 9—figure supplement 2*). The predicted flux of proton leak and mitochondrial TMRM intensity shows a strong negative correlation (*Figure 9i*), confirming our hypothesis.

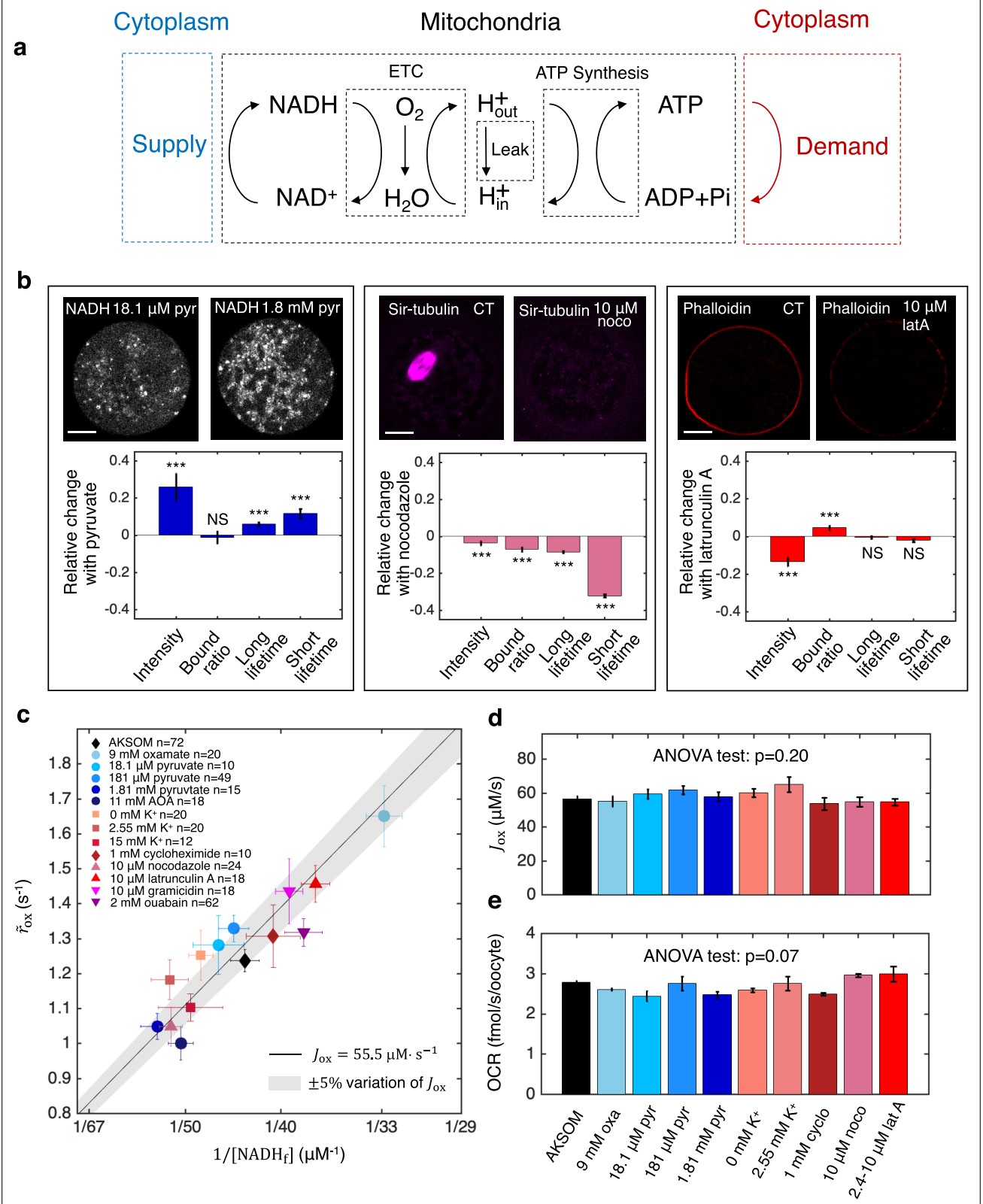

**Figure 8.** Homeostasis of ETC flux in mouse oocytes: perturbations of nutrient supply and energy demand impact NADH metabolic state but do not impact ETC flux. (**a**) The three coupled cycles of mitochondrial-based energy metabolism: the NADH/NAD⁺ redox cycle, the proton pumping/ dissipation cycle, and the ATP/ADP production/consumption cycle. Nutrients supplied from the cytoplasm (blue) power the reduction of NAD⁺ to NADH. Energy-demanding cellular processes in the cytoplasm (red) hydrolyze ATP to ADP. (**b**) Oocyte images (top) and change in NADH FLIM

*Figure 8 continued on next page*

*Figure 8 continued*

parameters relative to control (bottom) for changing pyruvate concentration (left), addition of 10 µM nocodazole (center) and addition of 10 µM latrunculin A (right). Student's *t*-test were performed for the change of FLIM parameters (\**p*<0.05, \*\**p*<0.01, \*\*\**p*<0.001). The spindle disassembles after addition of 10 µM nocodazole (top, center) and the actin cortex disassembles after addition of 10 µM latrunculin A (top, right). (**c**) NADH turnover rate ($\tilde{r}_{ox}$) and NADH free concentrations ($\left[\mathrm{NADH_f}\right]$) inferred from FLIM measurements under a variety of perturbations of nutrient supply and energy demand. Error bars are standard error of the mean (s.e.m) across oocytes. The black line corresponds to $\tilde{r}_{ox}$ and $\left[\mathrm{NADH_f}\right]$ values with an inferred flux of $J_{ox} = 55.5\ \mu\mathrm{M} \cdot \mathrm{s}^{-1}$, and the gray shaded region corresponds to a variation of ±5% around that value. (**d**) The inferred ETC flux and (**e**) measured OCR show no change across different perturbations of nutrient supply and energy demand (ANOVA, *p*=0.20 and *p*=0.07, respectively). ETC, electron transport chain; FLIM, fluorescence lifetime imaging microscopy; OCR, oxygen consumption rate.

The online version of this article includes the following source data and figure supplement(s) for figure 8:

**Source data 1.** Excel spreadsheet of single-oocyte FLIM data and batch OCR data used for *Figure 8c–e*.

**Figure supplement 1.** NADH FLIM parameters for mouse oocytes under all nutrient supply and energy demand perturbations.

**Figure supplement 1—source data 1.** Excel spreadsheet of single-oocyte FLIM data used for *Figure 8—figure supplement 1*.

Taken together, these results show that MII mouse oocytes contain subcellular spatial heterogeneities of mitochondrial metabolic activities. The observation that proton leak is responsible for the gradient of ETC flux suggests that the flux heterogeneity is a result of intrinsic mitochondrial heterogeneity. This is consistent with our conclusion from the homeostasis of ETC flux (*Figure 8*) that it is the intrinsic rates of mitochondrial respiration, not energy demand or supply, that controls the ETC flux. The causes and consequences of the subcellular spatial variation in mitochondrial activity remain unclear and are an exciting topic for future research.

## Discussion

### The NADH redox model is a general model to relate FLIM measurements of NADH to ETC fluxes

Despite extensive studies and applications of FLIM in metabolic research (*Bird et al., 2005*; *Skala et al., 2007*; *Heikal, 2010*; *Sharick et al., 2018*; *Sanchez et al., 2018*; *Liu et al., 2018*; *Sanchez et al., 2019*; *Ma et al., 2019*), it remains a challenge to relate FLIM measurements to the activities of the underlying metabolic pathways in cells. We overcame this challenge by developing a coarse-grained NADH redox model that leads to quantitative predictions for the relationship between FLIM measurements and the flux through the ETC. The model was constructed by explicitly coarse-graining a detailed NADH redox model with an arbitrary number of oxidases and reductases that represent all the possible enzymes involved in NADH redox reactions. The reactions in the detailed NADH redox model can be of arbitrary order and depend on implicit variables (i.e., free enzyme concentration, membrane potential, pH, etc.), which obey their own dynamical equations. The dynamics of the redox model will, of course, depend on the precise number of oxidases and reductases, the functional forms of the rates, and specific mathematical models for all the variables the rates implicitly depend on. However, the quantitative predictions relating FLIM measurements and ETC flux are independent of these modeling choices. Coarse-graining the detailed NADH redox model reduces all oxidases to an effective oxidase and all reductases to an effective reductase. The kinetic rates of the coarse-grained model can be related to those of the detailed model by keeping the global fluxes through the oxidases and the reductases the same in both models. The coarse-grained model predicts that the flux through the ETC is a product of the turnover rate and the concentration of free NADH (*Equation 5a*). The turnover rate is proportional to the difference between the nonequilibrium and the equilibrium NADH bound ratio (*Equation 5b*), which are measurable by FLIM of NADH (*Equation 5c*). Thus, this model provides a generic framework to relate FLIM measurements of NADH to the flux through the ETC in mitochondria.

The central assumption required for the validity of *Equations 5a-c* is that the redox reactions, and binding and unbinding processes, can be approximated as being at steady-state (i.e., undergoing only quasistatic changes over perturbations or development). At steady-state, the net binding and unbinding flux balances the oxidative flux of NADH. Therefore, the measurement of binding and unbinding state of NADH from FLIM allows the inference of the ETC flux, irrespective of the detailed behaviors of the oxidative reactions.

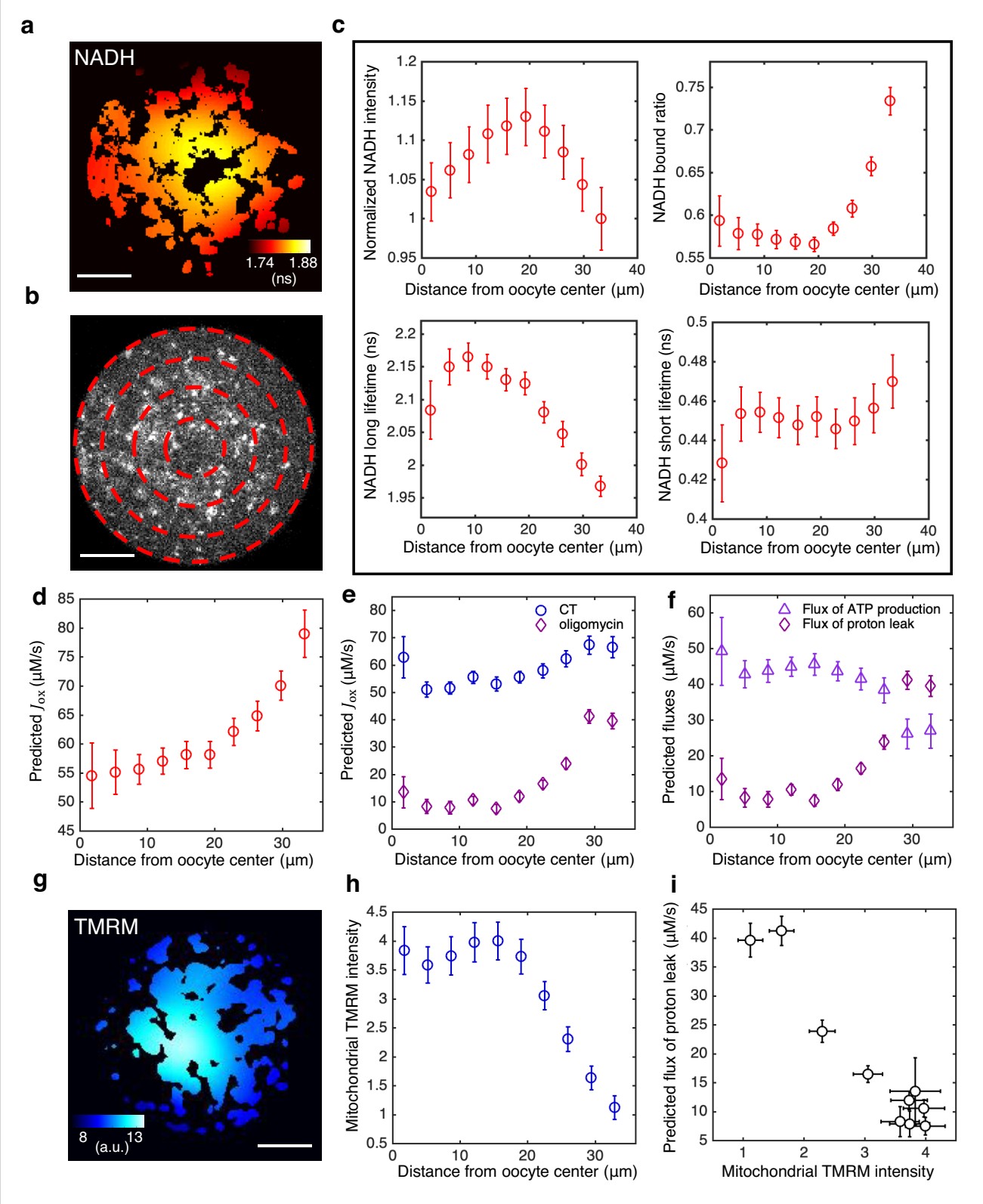

**Figure 9.** Subcellular mitochondrial heterogeneity in mouse oocytes: spatially inhomogeneous mitochondrial proton leak leads to a higher ETC flux in mitochondria closer to cell periphery. (**a**) Heatmap of the mean NADH fluorescence decay time in mitochondria exhibits a subcellular spatial gradient within oocytes. (**b**) NADH intensity image of the oocyte partitioned with equally spaced concentric rings. (**c**) Mitochondrial normalized NADH intensity (upper left), bound ratio $\beta$ (upper right), long fluorescence lifetime $\tau_l$ (lower left), and short fluorescence lifetime $\tau_s$ (lower right) as a function of distance

*Figure 9 continued on next page*

*Figure 9 continued*

from the oocyte center (*n*=67). (**d**) Predicted ETC flux from FLIM of NADH as a function of distance from the oocyte center (*n*=67). (**e**) ETC flux gradient is enhanced by 5 μM oligomycin (*n*=37), suggesting the flux gradient is determined by proton leak. CT is AKSOM with oxamate (*n*=32). 9 mM oxamate is present in oligomycin condition to reduce cytoplasmic NADH signal for better mitochondrial segmentation. (**f**) Opposing flux gradients of proton leak and ATP production, where proton leak (ATP production) is maximal (minimal) at the cell periphery. (**g**) Heatmap of the TMRM intensity in mitochondria, which increases with mitochondrial membrane potential, exhibits a subcellular spatial gradient within oocytes. (**h**) Mitochondrial TMRM intensity as a function of distance from the oocyte center (*n*=16). (**i**) Predicted flux of proton leak correlates negatively with mitochondrial membrane potential as measured by mitochondrial TMRM intensity. Scale bar, 20 μm. Error bars represent standard error of the mean (s.e.m) across different oocytes. ETC, electron transport chain; FLIM, fluorescence lifetime imaging microscopy.

The online version of this article includes the following figure supplement(s) for figure 9:

**Figure supplement 1.** $\beta_{eq}$ is uniform within the oocyte.

**Figure supplement 2.** Subcellular spatial gradient of mitochondrial membrane potential.

Remarkably, all the binding and unbinding rates of the NADH redox model are coarse-grained into two effective parameters: $\alpha$ and $\beta_{eq}$, which can be experimentally measured. We determined the value of $\alpha$ from an OCR measurement (Appendix 5, *Equation S48*), and we determined the value of $\beta_{eq}$ from FLIM of NADH at low oxygen levels or from rotenone perturbation (Appendix 5, *Figure 5—figure supplement 1h*). In MII mouse oocytes, $\alpha$ do not significantly vary in response to oxygen or drug and nutrient perturbations. This is demonstrated by the agreement between the predicted ETC flux and the measured OCR with a constant $\alpha$ of 5.4±0.2 s⁻¹ across a variety of conditions (*Figure 5*). $\alpha$ is predicted to depend only on the coarse-grained unbinding rates of NADH from the enzymes (*Equation S43*), so the observed constancy of $\alpha$ implies that the perturbations in this study primarily impacted the reduction/oxidation reaction rates (and not the unbinding rates). In other scenarios, such as when the concentrations of enzymes change, the coarse-grained unbinding rates might change, so $\alpha$ might not be a constant. In contrast, $\beta_{eq}$ does vary with drug and nutrient perturbations, but not with oxygen level, allowing $\beta_{eq}$ to be obtained at the lowest oxygen level for different drug and nutrient conditions (*Figure 5—figure supplement 1h* and *Figure 8—figure supplement 1*). Using these two parameters, we inferred the effective turnover rate of free NADH, $\tilde{r}_{ox}$, from FLIM measurements of NADH. By multiplying this turnover rate with the concentration of free NADH, $[\mathrm{NADH_f}]$ (also obtained from FLIM measurements using *Equation 2a*), we inferred the ETC flux from *Equation 5a*. Thus, all the complex behaviors of the binding and unbinding and reaction rates are captured by the variations in FLIM parameters of NADH, and our coarse-grained model provides a generic way to interpret these variations.

While we found that $\beta_{eq}$ is smaller than $\beta$ in mouse oocytes, this does not generically have to be true. Thus, if a perturbation is observed to decrease the NADH bound ratio $\beta$, it does not necessarily imply a decrease of the ETC flux. Similarly, a decrease of NADH long lifetime is not necessarily associated with an increase of the ETC flux. Therefore, measurements of $\alpha$ and $\beta_{eq}$ are required to use *Equations 5a-c* to infer ETC flux from FLIM measurements of NADH.

## The underlying assumptions and limitations of the NADH redox model

In this section, we clarify the underlying assumptions and limitations of the model to facilitate the accurate interpretation of FLIM measurements of NADH in different biological contexts.

To use the coarse-grained NADH model, segmentation needs to be performed to separate the mitochondrial NADH signal from the cytoplasmic NADH signal, because they encode different metabolic fluxes. In mouse oocytes, the segmentation can be reliably performed based on NADH images due to the higher NADH intensity in mitochondria than cytoplasm. Mitochondrial movements are also slow in MII oocytes (*Video 1*); hence, long exposure times can be used to obtain high contrast NADH images. For cells where NADH contrast is low, such as in yeast cells (*Papagiannakis et al., 2017*; *Shaw and Nunnari, 2002*), MitoTracker dye (Appendix 1 *Figure 1—figure supplement 1*) or mitochondrial associated fluorescent proteins (*Westermann and Neupert, 2000*) will likely be needed for reliable segmentation of mitochondria.

One of the most important assumptions that enables the coarse-grained model to be used to predict fluxes is that the NADH redox cycle can be well approximated as being at steady-state, that is, the rate of change of NADH concentrations is much slower than the kinetic rates, including the binding/unbinding rates and the reaction rates. This is true for mouse oocytes, where the NADH intensity does not significantly change over the course of hours. This assumption also holds for slow processes such as the cell cycle (*Papagiannakis et al., 2017*), which occurs on the timescale of hours

compared to timescales of seconds for the kinetic rates. This claim is supported by the success of the model on human tissue culture cells. The steady-state approximation could fail for rapid dynamics of NADH, such as the transient overshoot of NADH in neurons induced by acute external stimulus (*Díaz-García et al., 2021*), but this needs to be tested experimentally.

While NADH and NADPH share the same fluorescence spectrum, NADH concentration is 40 times greater than the concentration of NADPH for the whole mouse oocytes and presumably even higher for mitochondria (*Bustamante et al., 2017*). NADPH concentration can be comparable to that of NADH for other cell types such as tissue culture cells (*Park et al., 2016*). However, we have shown that the presence of NADPH signal and other background fluorescence signals only affect the equilibrium bound ratio $\beta_{eq}$ or the prefactor $\alpha$, and hence does not affect the flux inference procedure if $\beta_{eq}$ can be reliably determined and $\alpha$ remains a constant (Appendix 5). This was validated in tissue culture cells by comparing predicted ETC flux (*Figure 7*) with previous OCR measurements (*MacVicar and Lane, 2014*).

Finally, when relating NADH FLIM measurements to the ETC flux, we did not explicitly consider the contribution to the flux through $FADH_2$. This is a valid approximation when the $FADH_2$ oxidative flux is much smaller than the NADH oxidative flux, as is often the case since pyruvate dehydrogenase plus the TCA cycle yields four NADH molecules but only one $FADH_2$ molecule per cycle. Alternatively, if the $FADH_2$ flux is proportional to the NADH flux, then a rescaled value of $\alpha$ can be used in *Equation 5b* to effectively account for both fluxes. The proportionality of $FADH_2$ flux and NADH flux is expected when NADH and $FADH_2$ are produced from the same redox cycle with fixed stoichiometry, such as the pyruvate dehydrogenase and TCA cycle. This proportionality will break down if significant amounts of NADH and $FADH_2$ are produced in independent cycles where the stoichiometry varies, for example, when the glycerol phosphate shuttle acts as a reductase in mitochondria for $FADH_2$ but not for NADH.

Given these underlying assumptions, the model needs to be tested before being applied to other biological systems. The present study provides an example for such tests in mouse oocytes and human tissue culture cells by comparing the predicted ETC flux from FLIM with direct measurements of OCR across a wide range of perturbations.

## Towards spatiotemporal regulations of metabolic fluxes in cells

Cells transduce energy from nutrients to power various cellular processes. The ETC flux represents the total rate of energy conversion by mitochondria. Despite the detailed knowledge of the biochemistry of mitochondrial metabolism, it is still unclear what cellular processes determine ETC flux or how cells partition energetic fluxes to different cellular processes, including biosynthesis, ion pumping, and cytoskeleton assemblies. Energetic costs of specific cellular processes have been estimated from theoretical calculations (*Stouthamer, 1973*) or through inhibition experiments (*Mookerjee et al., 2017*). The latter typically involves measurements of the change of metabolic fluxes, such as OCR, upon inhibition of specific cellular processes, and interpreting this change as the energetic cost of the inhibited process. This interpretation is valid if metabolic flux is determined by the energy demand of different cellular processes in an additive manner. This assumption has not been thoroughly tested. Using the NADH redox model, we discovered a homeostasis of ETC flux in mouse oocytes where perturbing energy demand and supply do not impact ETC flux despite significantly changing NADH metabolic state. On the other hand, perturbing ATP synthesis and proton leak greatly impacted the ETC flux. From these results, we concluded that it is the intrinsic rates of mitochondrial respiration, rather than energy supply or demand, that controls the ETC flux in mouse oocytes. While NADH metabolic state significantly changed in response to perturbing energy demand and supply, indicating cell metabolism was indeed impacted, it is unclear if these perturbations also influenced ATP, ADP, or AMP levels. Future work, including direct measurements of ATP, ADP, and AMP levels, will be required to uncover the mechanism of flux homeostasis. More broadly, our work demonstrates that it is a prerequisite to understand the regulation of ETC fluxes in order to correctly interpret the changes of ETC flux upon inhibiting subcellular processes.

The mechanism of the homeostasis of ETC flux is unclear. One possibility is the presence of flux buffering pathways, where the change of ATP fluxes induced by process inhibition is offset by the opposing change of fluxes through the buffering pathways. Enzymes such as adenylate kinase are known to buffer concentrations of adenine nucleotide (*De la Fuente et al., 2014*), but it is unclear if they also buffer fluxes. Another possibility is a global coupling of cellular processes, where the change of ATP consumption by one process is offset by the change of others. Changes in proton leak could also compensate for changes in ATP production. Additional work will be required to distinguish between these (and other) possibilities.

FLIM data is obtained with optical resolution, enabling subcellular measurements of NADH metabolic state. Interpreting these measurements using the NADH redox model enables inference of metabolic fluxes with subcellular resolution. Using this method, we discovered a subcellular spatial gradient of ETC flux in mouse oocytes, where the ETC flux is higher in mitochondria closer to the cell periphery. We found that this flux gradient is primarily a result of a spatially heterogeneous mitochondrial proton leak. It will be an exciting aim for future research to uncover the causes and consequences of the subcellular spatial variation in mitochondrial activity.

# Materials and methods

**Key resources table**

| Reagent type (species) or resource | Designation | Source or reference | Identifiers | Additional information |
|---|---|---|---|---|
| Cell line (*Homo sapiens*) | hTERT-RPE1 | Iain Cheeseman Lab | ATCC Cat# CRL-4000, RRID:CVCL_4388 | |
| Biological sample (mouse) | MII oocytes | EmbryoTech | Strain: B6C3F1 | |
| Commercial assay or kit | MitoTracker Red FM | Thermo Fisher Scientific | Cat.#: M22425 | |
| Commercial assay or kit | TMRM | Sigma-Aldrich | Cat.#: T5428 CAS: 115532-50-8 | |
| Commercial assay or kit | JC-1 | Thermo Fisher Scientific | Cat.#: T3168 | |
| Commercial assay or kit | SiR-Tubulin | Cytoskeleton Inc | Cat.#: CY-SC006 | |
| Commercial assay or kit | Phalloidin | Thermo Fisher Scientific | Cat.#: F432 | |
| Chemical compound, drug | Sodium oxamate | Sigma-Aldrich | Cat.#: O2751 CAS: 565-73-1 | |
| Chemical compound, drug | Rotenone | Sigma-Aldrich | Cat.#: R8875 CAS: 83-79-4 | |
| Chemical compound, drug | Oligomycin A | Sigma-Aldrich | Cat.#: 75351 CAS: 579-13-5 | |
| Chemical compound, drug | FCCP | Sigma-Aldrich | Cat.#: C2920 CAS: 370-86-5 | |
| Chemical compound, drug | CCCP | Sigma-Aldrich | Cat.#: C2759 CAS: 555-60-2 | |
| Chemical compound, drug | Glucose | Sigma-Aldrich | Cat.#: D9434 CAS: 50-99-7 | |
| Chemical compound, drug | Galactose | Millipore | Cat.#: 48260 CAS: 59-23-4 | |
| Chemical compound, drug | Pyruvate | Sigma-Aldrich | Cat.#: P2256 CAS: 113-24-6 | |
| Chemical compound, drug | Cycloheximide | Sigma-Aldrich | Cat.#: C4859 CAS: 66-81-9 | |
| Chemical compound, drug | Nocodazole | Sigma-Aldrich | Cat.#: M1404 CAS: 31430-18-9 | |
| Chemical compound, drug | Latrunculin A | Sigma-Aldrich | Cat.#: L5163 CAS: 76343-93-6 | |
| Chemical compound, drug | Gramicidin | Sigma-Aldrich | Cat.#: 50845 CAS: 11029-61-1 | |
| Chemical compound, drug | Ouabain | Sigma-Aldrich | Cat.#: O3125 CAS: 11018-89-6 | |
| Chemical compound, drug | Aminooxyacetic acid (AOA) | Sigma-Aldrich | Cat.#: C13408 CAS: 2921-14-4 | |
| Software, algorithm | FLIM data acquisition (SPCM) | Becker & Hickl | RRID:SCR_018310 | |

*Continued on next page*

*Continued*

| Reagent type (species) or resource | Designation | Source or reference | Identifiers | Additional information |
|---|---|---|---|---|
| Software, algorithm | FLIM data acquisition (Labview) | National Instruments | RRID:SCR_014325 | |
| Software, algorithm | FLIM data analysis (MATLAB R2015b) | MathWorks | RRID:SCR_001622 | |
| Software, algorithm | OCR data acquisition (SensorTrace Profiling) | Unisense | | |

## Culturing of mouse oocytes

Frozen MII mouse oocytes (Strain B6C3F1) were purchased from EmbryoTech. Oocytes were thawed and cultured in droplets of AKSOM media purchased from MilliporeSigma in plastic petri dish. Mineral oil from VitroLife was applied to cover the droplets to prevent evaporation of the media. Oocytes were then equilibrated in an incubator at 37°C, with 5% $CO_2$ and air saturated oxygen before imaging. For imaging, oocytes were transferred to a 2-µl media droplet in a 35-mm glass bottom FluoroDish from WPI covered with 400–500 µl of oil. The glass bottom dish was placed in an Ibidi chamber with temperature and gas control during imaging. Temperature was maintained at 37°C via heated chamber and objective heater. $CO_2$ was maintained at 5% using gas tanks from Airgas.

## Cell lines

The hTERT-RPE1 cell line is an established wild-type cell line received from the Cheeseman lab that has been validated based on behavior and properties. The hTERT-RPE1 cell line was maintained and tested for mycoplasma contamination in the Needleman lab on a regular basis (Southern Biotech).

## Culturing of hTERT-RPE1 cells

Cell lines were maintained at 37°C and 5% $CO_2$. Cells were grown in Dulbecco's modified Eagle's medium (DMEM) (11966025, Gibco) supplemented with 10% fetal bovine serum (FBS), 0.5 mM sodium pyruvate, 5 mM HEPES, 1% penicillin and streptomycin, and either 10 mM glucose or 10 mM galactose. Cells were passaged in glucose or galactose at least three times before imaging. Cells were plated on 35 mm glass bottom FluoroDishes from WPI for imaging. Right before imaging, the media was replaced with 1 ml of phenol red-free DMEM (A1443001, Gibco) supplemented with 0.5 mM sodium pyruvate, 4 mM L-glutamine, 10 mM HEPES, and either 10 mM glucose or 10 mM galactose.

## FLIM measurements

Our FLIM system consists of a two-photon confocal microscope with a 40× 1.25 NA water immersion Nikon objective, Becker and Hickle Time Correlated Single Photon Counting (TCSPC) acquisition system and a pulsed MaiTai DeepSee Ti:Sapphire laser from Spectra-Physics. NADH autofluorescence was obtained at 750 nm excitation wavelength with a 460/50 nm emission filter. Laser power at the objective was maintained at 3 mW. The scanning area was 512 by 512 pixels with a pixel size of 420 nm. Acquisition time was 30 s per frame. Oocytes were imaged with optical sectioning across their equators. A histogram of NADH fluorescence decay times was obtained at each pixel of the image.

## Oxygen measurements

Oxygen level was measured in the Ibidi chamber with an electrode-based oxygen sensor (GasLab). Since the oil layer covering the media droplet was very thin, the oxygen level in the droplet was assumed to be in instant equilibrium with the chamber.

## Image and FLIM data analysis

To separate mitochondrial NADH signal from cytoplasmic signal, we performed machine learning-based segmentation algorithms on NADH intensity images. We used the freeware Ilastik (*Berg et al., 2019*), which implements a supervised learning algorithm for pixel classification. The classifiers were trained to separate mitochondrial pixels from cytoplasmic pixels with a greater than 80% accuracy, as tested by MitoTracker Red FM (Appendix 1, *Figure 1—figure supplement 1*). We grouped photons

from all mitochondrial pixels to obtain a histogram of NADH decay times for each oocyte and for each image of tissue culture cells. To extract the FLIM parameters of NADH bound fraction $f$, long lifetime $\tau_1$ and short lifetime $\tau_s$, we fitted the histogram with $G = \mathrm{IRF} * (C_1 F + C_2)$, where $*$ indicates a convolution, and $\mathrm{IRF}$ is the instrument response function of the FLIM system, measured using a urea crystal. $F(\tau) = f \cdot \exp\left(-\frac{\tau}{\tau_1}\right) + (1-f) \cdot \exp\left(-\frac{\tau}{\tau_s}\right)$ is the two-exponential model for the NADH fluorescence decay. $C_1$ is the amplitude of the decay and $C_2$ is the background noise. The fitting was performed with a custom MATLAB code using a Levenberg-Marquardt algorithm (*Yoo, 2018*). To obtain the intensity, $I$, of mitochondrial NADH, we first measured the average number of photons per mitochondrial pixel, and divided it by the pixel area, 0.185 μm², and pixel scanning time 4.09 μs. The flux of ETC is inferred using *Equations 5a-c* for each oocyte and for tissue culture cells in a single image. Heatmaps of mean NADH fluorescence decay times were obtained by computing NADH fluorescence decay time of each mitochondrial pixel and averaging over neighboring mitochondrial pixels weighted by a Gaussian kernel with a standard deviation of 20 pixels. All FLIM measurements were taken from distinct individual oocytes and distinct images of tissue culture cells. Error bars in all figures of FLIM represent standard error of the mean across different individual oocytes or across different images for tissue culture cells. Number of oocytes is reported with *n*. Number of images for tissue culture cells is reported with *N*.

## Error analysis

FLIM curves were independently fit for each individual oocyte. The reported error bars in this manuscript are standard errors of the mean (SEMs) across these measurements, which depends on the level of variation (the standard deviation) between the oocytes. Two sources of variation in FLIM measurements across the oocytes are: (1) true biological variations between oocytes and (2) fitting errors in the FLIM analysis. To estimate the error of fitting, we performed bootstrapping with randomly drawn points with substitution from each fluorescence decay curve for 53 oocytes. There are ~66,000 photons per oocyte, from which we generated 10 bootstrapped decay curves per oocyte to estimate the fitting error. The fitting error is computed as the variance and covariance of the fitted parameters across bootstrapped decay curves and averaged over 53 oocytes.

At high oxygen level in the AKSOM condition, the bootstrapping yields a variance of $2.2 \times 10^{-4}$, $4.6 \times 10^{-3}$ ns², and $6.0 \times 10^{-4}$ ns² for bound fraction, long lifetime, and short lifetime, respectively. The cell-to-cell variances obtained from a single fit per oocyte are $4.4 \times 10^{-4}$, $9.5 \times 10^{-3}$ ns², and $1.6 \times 10^{-3}$ ns² for bound fraction, long lifetime, and short lifetime, respectively. Hence the bootstrapping error accounts for 50%, 49%, and 40% of the cell-to-cell variance in bound fraction, long lifetime, and short lifetime, respectively. The bootstrapping yields a covariance of $-1.0 \times 10^{-3}$ ns between bound fraction and long lifetime, which only accounts for ~20% of the covariance between these two variables during oxygen drop experiment. The inferred mean flux for oocytes at high oxygen levels in AKSOM is $\langle J_{\mathrm{ox}} \rangle = 56.6 \ \mu\mathrm{M} \cdot \mathrm{s}^{-1}$. Propagating the error of fitting in all parameters from the bootstrapping analysis to the inferred flux gives a standard error of the mean in $J_{\mathrm{ox}}$ of 1.1 μM·s⁻¹. The standard error of the mean in $J_{\mathrm{ox}}$ obtained from a single fit per oocyte was 2.0 μM·s⁻¹. Thus, fitting errors account for ~50% of the standard error of the mean in $J_{\mathrm{ox}}$.

## Metabolic and demand perturbations

Oxygen drop experiments for oocytes were performed by mixing nitrogen-balanced 5% $O_2$ gas with 0% $O_2$ gas at different ratios to create a continuous oxygen drop profile. $CO_2$ was maintained at 5%. Oocytes were imaged for 10 min at 5% $O_2$, 30 min during the continuous drop from 5% $O_2$ to approximately 0% $O_2$, and 20 min after quickly returning to 5% $O_2$. Oxygen levels were simultaneously monitored with an electrode-based oxygen sensor in the Ibidi chamber. 5% $O_2$ corresponds to ~50 μM of oxygen concentration in the culturing media. All the drug perturbations for oocytes were performed by equilibrating oocytes in the AKSOM media containing the corresponding drug for 15–30 min before the oxygen drop experiments. Pyruvate and potassium perturbations were performed by making KSOM media following Cold Spring Harbor Laboratory protocols with varying concentrations of sodium pyruvate and potassium, respectively. For oligomycin, FCCP, rotenone and pyruvate perturbations, 9 mM of sodium oxamate was also added to the media to suppress cytoplasmic NADH signal for better mitochondrial segmentation. The addition of the oxamate does not change the ETC flux of the mitochondria (*Figure 5b*).

For hTERT-RPE1 cells, drug perturbations were performed by replacing the media with drug-containing media through pipetting. Cells were imaged for 20–30 min immediately after drug perturbations.

All drugs were purchased from Sigma-Aldrich. Temperature was maintained at 37°C. $CO_2$ was maintained at 5%.

## Oxygen consumption rate measurement

The OCR of the oocytes was measured using the nanorespirometer from Unisense (*Lopes et al., 2005*). A batch of 10–15 oocytes was placed at the bottom of a glass capillary with a diameter of 0.68 mm and a height of 3 mm. The capillary well is filled with AKSOM media or drug-containing media for metabolic perturbations. After an equilibration time of ~2 hr, a steady-state linear oxygen gradient is established in the capillary well due to the balance of oocyte respiration and oxygen diffusion. A motor-controlled electrode-based oxygen sensor (Unisense) is used to measure the oxygen gradient. The OCR is calculated as the product of the oxygen gradient, diffusivity of oxygen in the media, taken to be $3.37 \times 10^{-5}$ cm$^2$/s, and the cross-sectional area of the capillary well, which was 0.36 mm$^2$. The entire system was enclosed in a custom-built chamber with temperature and gas control. Temperature was maintained at 37°C. Oxygen level was continuously varied during oxygen drop experiments by slowly mixing 20% $O_2$ with 0% $O_2$ from gas tanks, and maintained at the air saturation level for drug and pyruvate perturbations. OCR was measured on a group of 10–15 oocytes at a time. Single-oocyte OCR was obtained by dividing the measured OCR by the number of oocytes in the group. Error bars in all figures of OCR represent standard error of the mean across different groups of oocytes normalized by the number of oocytes in each group. Number of oocytes is reported with $n$. Number of groups is reported with $N$.

## Statistical analysis

For the comparison between inferred ETC flux and measured ETC flux of the oocytes, two-sample t-test was performed on the vectors of inferred single-cell ETC flux (with $n$ elements, where $n$ is the number of oocytes) and the batch OCR measurements (with $N$ elements, where $N$ is the number of batch groups). For the comparison between inferred ETC flux and measured ETC flux of the tissue culture cells, two-sample t-test was performed on the vectors of inferred relative change of ETC flux (with $n$ elements, where $n$ is the number of images) and the relative change of OCR estimated from Figure 1A of *MacVicar and Lane, 2014* (with $N$ elements, where $N$ is the estimated number of OCR data points).

## Mitochondrial membrane potential measurement

The spatial distribution of mitochondrial membrane potential within oocytes was measured with a potential-sensitive dye TMRM (Sigma-Aldrich). Oocytes were cultured in AKSOM with 100 nM TMRM for 30 min before imaging. TMRM signal was obtained at 830 nm excitation wavelength with 560/40 nm emission filter. Mitochondrial TMRM intensity in different regions of the oocyte was computed by dividing the total number of photons from that region by the number of pixels in the same region. Heatmaps of mitochondrial TMRM intensity were obtained by computing photon counts for each mitochondrial pixel and averaging over neighboring mitochondrial pixels weighted by a Gaussian kernel with a standard deviation of 20 pixels. To normalize TMRM intensity by mitochondrial mass, we cultured oocytes in AKSOM with 100 nM MitoTracker Red FM and 25 nM TMRM for 30 min before imaging. We also cultured oocytes in AKSOM with 1 µg/ml JC-1 dye for 3 hr before imaging.

Mitochondrial membrane potential of hTERT-RPE1 cells was measured with TMRM. The cells were cultured in DMEM with 100 nM TMRM for 15–30 min before imaging. To measure membrane potential under drug perturbations, the original media was pipetted out and replaced with media containing both 100 nM TMRM and the drug. The cells were imaged for 20–30 min immediately after drug perturbations. TMRM intensity ratio was obtained by normalizing the mitochondrial TMRM intensity by the cytoplasmic TMRM intensity.

## Acknowledgements

The authors thank Easun Arunachalam, Yu-Chen Chao, Will Conway, Carlos Manlio Díaz-García, Peter Foster, Bill Ireland, Denis Titov, and Gary Yellen for suggestions, advice, and comments on the manuscript. This work is supported by the National Institutes of Health (R01HD092550-01) and the National

Science Foundation (PFI-TT-1827309, PHY-2013874, and MCB-2052305). G.H. acknowledges support from the NSF-Simons Center for Mathematical and Statistical Analysis of Biology at Harvard (Award number #1764269) and the Harvard Quantitative Biology Initiative. X.Y. and D.J.N. acknowledge discussions with participants of the 'Cellenergy19' and the 'Active20' KITP programs, supported in part by the National Science Foundation Grant no. NSF PHY-1748958, NIH Grant no. R25GM067110, and the Gordon and Betty Moore Foundation Grant no. 2919.02.

## Additional information

### Funding

| Funder | Grant reference number | Author |
| --- | --- | --- |
| National Institutes of Health | R01HD092550-01 | Dan Needleman |
| National Science Foundation | PFI-TT-1827309 | Dan Needleman |
| National Science Foundation | PHY-2013874 | Dan Needleman |
| National Science Foundation | MCB-2052305 | Dan Needleman |
| National Science Foundation | PHY-1748958 | Xingbo Yang Dan Needleman |
| National Institutes of Health | R25GM067110 | Xingbo Yang Dan Needleman |
| Gordon and Betty Moore Foundation | 2919.02 | Xingbo Yang Dan Needleman |
| National Science Foundation | 1764269 | Gloria Ha |

The funders had no role in study design, data collection and interpretation, or the decision to submit the work for publication.

### Author contributions

Xingbo Yang, Conceptualization, Data curation, Formal analysis, Investigation, Methodology, Resources, Software, Validation, Visualization, Writing – original draft, Writing – review and editing; Gloria Ha, Resources, Writing – review and editing, cultured and prepared the hTERT-RPE1 cells for FLIM imaging; Daniel J Needleman, Conceptualization, Funding acquisition, Supervision, Writing – original draft, Writing – review and editing

### Author ORCIDs

Xingbo Yang http://orcid.org/0000-0002-5798-4448
Gloria Ha http://orcid.org/0000-0002-4076-5337

### Decision letter and Author response

Decision letter https://doi.org/10.7554/eLife.73808.sa1
Author response https://doi.org/10.7554/eLife.73808.sa2

## Additional files

### Supplementary files

• Transparent reporting form

## Data availability

All data generated or analysed during this study are included in the manuscript and supporting file; Source Data files have been provided for Figures 2, Figure 5, Figure 5—figure supplement 1, Figure 6, Figure 6—figure supplement 1, Figure 7, Figure 8, Figure 8—figure supplement 1.

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

## Appendix 1

### Segmentation of mitochondria and calculation of NADH concentrations

#### Segmentation of mitochondria

We used Ilastik, a machine learning-based software for image analysis, to classify pixels in the NADH intensity images containing mitochondria (*Berg et al., 2019*). For each experiment, we generated a time-lapse movie of NADH (*Video 1*). We used a few images in the movie as the training data set to train the software to classify mitochondrial pixels by manually selecting clustered high brightness pixels. Other pixels are classified as either cytoplasm or background. We then applied the trained pixel classifier to generate a mitochondrial probability map for each image in the entire movie, with each pixel assigned a probability between 0 and 1 to be mitochondrial pixel. Pixels with a probability higher than 0.7 were considered to be mitochondrial pixels.

To test the accuracy of this segmentation algorithm, we immersed the oocytes in AKSOM media containing MitoTracker Red FM, a dye that specifically labels mitochondria. Pixels with intensity above 60 percentile in the MitoTracker image were considered to be mitochondrial pixels. We imaged NADH and MitoTracker for the same oocyte and compared the resulting distribution of mitochondria (*Figure 1—figure supplement 1*). We defined the accuracy of the NADH-based segmentation as the fraction of photons originating from true mitochondrial pixels. The accuracy of the segmentation is 80.6±1% (SEM) for the control condition as averaged over seven oocytes. We repeated the analysis for oxamate, oligomycin, FCCP and rotenone perturbations, and obtained an accuracy of segmentation of 78.6±1.4%, 84.1±1.6%, 83.7±0.5%, and 81.7±2%, respectively, similar to the control condition.

#### Converting NADH intensity to NADH concentrations

Since the molecular brightness of NADH depends on the fluorescence lifetime of NADH, which changes drastically upon binding enzymes, the NADH concentration is not linearly proportional to NADH intensity. FLIM provides an accurate way of measuring NADH concentrations by simultaneously measuring fluorescence intensity and lifetime. We now derive the NADH intensity-concentration relation from the FLIM measurements. Assuming molecular brightness is proportional to the fluorescence lifetime, and therefore that free and bound NADH have different contributions to the measured intensity, we have

$$I = c_s \tau_s \left[ \text{NADH}_f \right] + c_s \tau_l \left[ \text{NADH}_b \right], \tag{S1}$$

where $I$ is the intensity of NADH and $c_s$ is a calibration factor that depends on the laser power. From *Equation S1*, we obtained the concentrations of free and bound NADH:

$$\left[ \text{NADH}_f \right] = \frac{I(1-f)}{c_s \left[ (\tau_l - \tau_s)f + \tau_s \right]}, \tag{S2}$$

$$\left[ \text{NADH}_b \right] = \left[ \text{NADH}_f \right] \frac{f}{1-f}, \tag{S3}$$

where $f$ is the fraction of bound NADH.

To get the calibration factor $c_s$, we titrated NADH in AKSOM solutions and fitted the calibration curve using:

$$I = c_s \tau_{sol} \left[ \text{NADH}_{sol} \right], \tag{S4}$$

where $\tau_{sol}$ is the lifetime of NADH in solution. $\tau_{sol}$ was directly measured by FLIM (*Figure 1—figure supplement 2*), allowing us to obtain $c_s$ from the fit (*Figure 1—figure supplement 2*).

### FLIM can be used to accurately measure concentrations of bound and free NADH in vitro

To test if absolute concentrations of free and bound NADH can be accurately measured from FLIM of NADH, we prepared solutions with known total concentration of NADH, and titrated the concentration of LDH, an enzyme to which NADH can bind. We prepared the solutions with 50 mM TRIS buffer, 150 mM NaCl at pH 7.6 and 37°C. We added a total concentration of 50 µM NADH to the solution and titrated LDH concentrations at 0, 1.4, and 3.5 µM. We first performed single exponential fitting of the NADH decay curve at 0 µM LDH, where all NADH are free, to obtain the

NADH short lifetime (*Figure 1—figure supplement 3d*). From the NADH intensity (*Figure 1—figure supplement 3a*), we obtained the calibration factor $c_s$ using *Equation S4* with $[\mathrm{NADH_{sol}}] = 50\mu M$. We then fixed the short lifetime and performed two-exponential fitting of the NADH decay curve at LDH concentrations of 1.4 μM and 3.5 μM to obtain the bound ratio (*Figure 1—figure supplement 3b*) and long lifetime (*Figure 1—figure supplement 3c*). As expected, NADH bound ratio increases with LDH concentrations, as there is more enzyme for NADH to bind. Finally, we calculated free NADH concentration $[\mathrm{NADH_f}]$ and bound NADH concentration $[\mathrm{NADH_b}]$ using *Equations S2 and S3* from the FLIM parameters. Remarkably, the free and bound concentrations of NADH both change with LDH concentrations but the total concentration remains at 50 μM (*Figure 1—figure supplement 3e*). This result shows that *Equations S2-S3* can be used to accurately measure the concentrations of free and bound NADH from FLIM measurements of NADH.

## Appendix 2

### Reversible Michaelis-Menten kinetics, full and reduced notations

The kinetic equations of the reversible Michaelis-Menten kinetics (*Figure 3a*, left) for the $i$th oxidase are

$$\frac{d\left[\text{NADH}_\text{f}\right]}{dt} = k_{-1}\left[\text{NADH} \cdot \text{Ox}_i\right] - k_1\left[\text{Ox}_i\right]\left[\text{NADH}_\text{f}\right], \tag{S5}$$

$$\frac{d\left[\text{NADH} \cdot \text{Ox}_i\right]}{dt} = k_1\left[\text{Ox}_i\right]\left[\text{NADH}_\text{f}\right] - k_{-1}\left[\text{NADH} \cdot \text{Ox}_i\right] + k_{-2}\left[\text{Ox}_i\right]\left[\text{NAD}_\text{f}^+\right] - k_2\left[\text{NADH} \cdot \text{Ox}_i\right], \tag{S6}$$

$$\frac{d\left[\text{NAD}_\text{f}^+\right]}{dt} = k_2\left[\text{NADH} \cdot \text{Ox}_i\right] - k_{-2}\left[\text{Ox}_i\right]\left[\text{NAD}_\text{f}^+\right], \tag{S7}$$

$$\frac{d\left[\text{Ox}_i\right]}{dt} = k_{-1}\left[\text{NADH} \cdot \text{Ox}_i\right] - k_1\left[\text{Ox}_i\right]\left[\text{NADH}_\text{f}\right] + k_2\left[\text{NADH} \cdot \text{Ox}_i\right] - k_{-2}\left[\text{Ox}_i\right]\left[\text{NAD}_\text{f}^+\right]. \tag{S8}$$

where $\left[\text{NADH}_\text{f}\right]$ is the concentration of free NADH, $\left[\text{NAD}_\text{f}^+\right]$ is the concentration of free NAD$^+$, $\left[\text{Ox}_i\right]$ is the concentration of free oxidase, and $\left[\text{NADH} \cdot \text{ox}_i\right]$ is the concentration of the NADH-oxidase complex. $k_{-1}$, $k_1$, $k_{-2}$, and $k_2$ are the forward and reverse reaction rates.

In the reduced notation as introduced in *Figure 3a* (right), the same enzyme kinetics is described by

$$\frac{d\left[\text{NADH}_\text{f}\right]}{dt} = k_{\text{ox}_i}^\text{u}\left[\text{NADH} \cdot \text{Ox}_i\right] - k_{\text{ox}_i}^\text{b}\left[\text{NADH}_\text{f}\right], \tag{S9}$$

$$\frac{d\left[\text{NADH} \cdot \text{Ox}_i\right]}{dt} = k_{\text{ox}_i}^\text{b}\left[\text{NADH}_\text{f}\right] - k_{\text{ox}_i}^\text{u}\left[\text{NADH} \cdot \text{Ox}_i\right] + k_{\text{ox}_i}^{\prime\text{b}}\left[\text{NAD}_\text{f}^+\right] - k_{\text{ox}_i}^{\prime\text{u}}\left[\text{NADH} \cdot \text{Ox}_i\right], \tag{S10}$$

$$\frac{d\left[\text{NAD}_\text{f}^+\right]}{dt} = k_{\text{ox}_i}^{\prime\text{u}}\left[\text{NADH} \cdot \text{Ox}_i\right] - k_{\text{ox}_i}^{\prime\text{b}}\left[\text{NAD}_\text{f}^+\right], \tag{S11}$$

where $k_{\text{ox}_i}^\text{b}$ and $k_{\text{ox}_i}^\text{u}$ are the effective binding and unbinding rates of NADH to the oxidase, and $k_{\text{ox}_i}^{\prime\text{b}}$ and $k_{\text{ox}_i}^{\prime\text{u}}$ are the effective binding and unbinding rates of NAD$^+$ to the oxidase. In this reduced notation, the concentration of the free oxidase $\left[\text{Ox}_i\right]$ is absorbed into the effective binding rates: that is, $k_{\text{ox}_i}^\text{u} = k_{-1}$, $k_{\text{ox}_i}^\text{b} = k_1\left[\text{Ox}_i\right]$, $k_{\text{ox}_i}^{\prime\text{u}} = k_2$, and $k_{\text{ox}_i}^{\prime\text{b}} = k_{-2}\left[\text{Ox}_i\right]$. Hence $\left[\text{Ox}_i\right]$ becomes an implicit variable whose behavior is not evident from the reduced notation diagram (*Figure 3a*, right). Modeling the full dynamics of a reversible Michalis-Menten enzyme requires specifying the equation for $\left[\text{Ox}_i\right]$:

$$\frac{d\left[\text{Ox}_i\right]}{dt} = k_{-1}\left[\text{NADH} \cdot \text{Ox}_i\right] - k_1\left[\text{Ox}_i\right]\left[\text{NADH}_\text{f}\right] + k_2\left[\text{NADH} \cdot \text{Ox}_i\right] - k_{-2}\left[\text{Ox}_i\right]\left[\text{NAD}_\text{f}^+\right]. \tag{S12}$$

The reduced notation (*Figure 3a*, right; *Equations S9–S12*) and the full notation (*Figure 3a*, left; *Equations S5-S8*) are mathematically identical and describe the exact same kinetics.

### Generalized enzyme kinetics, reduced notation

The reduced notation for the $i$th oxidase displaying generalized enzyme kinetics (*Figure 3b*) refers to the following class of mathematical models:

$$\frac{d\left[\text{NADH}_\text{f}\right]}{dt} = k_{\text{ox}_i}^\text{u}\left[\text{NADH} \cdot \text{Ox}_i\right] - k_{\text{ox}_i}^\text{b}\left[\text{NADH}_\text{f}\right], \tag{S13}$$

$$\frac{d\left[\text{NADH} \cdot \text{Ox}_i\right]}{dt} = k_{\text{ox}_i}^\text{b}\left[\text{NADH}_\text{f}\right] - k_{\text{ox}_i}^\text{u}\left[\text{NADH} \cdot \text{Ox}_i\right] - r_{\text{ox}_i}^+\left[\text{NADH} \cdot \text{Ox}_i\right] + r_{\text{ox}_i}^-\left[\text{NAD}^+ \cdot \text{Ox}_i\right], \tag{S14}$$

$$\frac{d\left[\text{NAD}^+ \cdot \text{Ox}_i\right]}{dt} = r_{\text{ox}_i}^+\left[\text{NADH} \cdot \text{Ox}_i\right] - r_{\text{ox}_i}^-\left[\text{NAD}^+ \cdot \text{Ox}_i\right] - k_{\text{ox}_i}^{\prime\text{u}}\left[\text{NAD}^+ \cdot \text{Ox}_i\right] + k_{\text{ox}_i}^{\prime\text{b}}\left[\text{NAD}_\text{f}^+\right], \tag{S15}$$

$$\frac{d\left[\text{NAD}_\text{f}^+\right]}{dt} = k_{\text{ox}_i}^{\prime\text{u}}\left[\text{NAD}^+ \cdot \text{Ox}_i\right] - k_{\text{ox}_i}^{\prime\text{b}}\left[\text{NAD}_\text{f}^+\right]. \tag{S16}$$

where $\left[\text{NADH}_\text{f}\right]$ is the concentration of free NADH, $\left[\text{NAD}_\text{f}^+\right]$ is the concentration of free NAD$^+$, $\left[\text{Ox}_i\right]$ is the concentration of free oxidase, $\left[\text{NADH} \cdot \text{Ox}_i\right]$ is the concentration of the NADH-oxidase complex, and $\left[\text{NAD}^+ \cdot \text{Ox}_i\right]$ is the concentration of the NAD$^+$-oxidase complex. $k_{\text{ox}_i}^\text{b}$ and $k_{\text{ox}_i}^\text{u}$ are the

effective binding and unbinding rates of NADH to the oxidase, $k_{\mathrm{ox}_i}^{\prime\mathrm{b}}$ and $k_{\mathrm{ox}_i}^{\prime\mathrm{u}}$ are the effective binding and unbinding rates of NAD$^+$ to the oxidase, and $r_{\mathrm{ox}_i}^+$ and $r_{\mathrm{ox}_i}^-$ are the forward and reverse oxidation rates. These rates can be arbitrary functions of implicit variables, such as the concentration of free oxidase, $[\mathrm{Ox}_i]$, the mitochondrial membrane potential, $\Delta G_\mathrm{H}$, pH, and other factors:

$$k_{\mathrm{ox}_i}^\mathrm{u} = k_{\mathrm{ox}_i}^\mathrm{u}\left([\mathrm{Ox}_i], \Delta G_\mathrm{H}, \mathrm{pH}, [\mathrm{NADH_f}], \ldots\right), \tag{S17a}$$

$$k_{\mathrm{ox}_i}^\mathrm{b} = k_{\mathrm{ox}_i}^\mathrm{b}\left([\mathrm{Ox}_i], \Delta G_\mathrm{H}, \mathrm{pH}, [\mathrm{NADH_f}], \ldots\right), \tag{S17b}$$

$$r_{\mathrm{ox}_i}^+ = r_{\mathrm{ox}_i}^+\left([\mathrm{Ox}_i], \Delta G_\mathrm{H}, \mathrm{pH}, [\mathrm{NADH_f}], \ldots\right), \tag{S17c}$$

$$r_{\mathrm{ox}_i}^- = r_{\mathrm{ox}_i}^-\left([\mathrm{Ox}_i], \Delta G_\mathrm{H}, \mathrm{pH}, [\mathrm{NADH_f}], \ldots\right), \tag{S17d}$$

$$k_{\mathrm{ox}_i}^{\prime\mathrm{u}} = k_{\mathrm{ox}_i}^{\prime\mathrm{u}}\left([\mathrm{Ox}_i], \Delta G_\mathrm{H}, \mathrm{pH}, [\mathrm{NADH_f}], \ldots\right), \tag{S17e}$$

$$k_{\mathrm{ox}_i}^{\prime\mathrm{b}} = k_{\mathrm{ox}_i}^{\prime\mathrm{b}}\left([\mathrm{Ox}_i], \Delta G_\mathrm{H}, \mathrm{pH}, [\mathrm{NADH_f}], \ldots\right). \tag{S17f}$$

Thus, while *Equations S13-S16* superficially appear to be linear and first order, they can actually refer to nonlinear reactions of any order because the rate can depend on $[\mathrm{NADH_f}]$ and other variables (*Equations S17a–f*).

The implicit variables that these rates depend on can each be governed by their own dynamics that are arbitrary functions of other variables:

$$\frac{\mathrm{d}[\mathrm{Ox}_i]}{\mathrm{dt}} = \ldots; \frac{\mathrm{d}[\Delta G_\mathrm{H}]}{\mathrm{dt}} = \ldots; \frac{\mathrm{d}[\mathrm{pH}]}{\mathrm{dt}}\ldots; \text{etc.} \tag{S18}$$

Describing the dynamics of the enzyme requires specifying the implicit variables that the rates depend on, the functional form of these dependencies, and the additional equations for the dynamics of the implicit variables (*Equations S18*). However, we will show that the predicted relationship between FLIM measurements of NADH and fluxes does not depend on these modeling choices. Thus, the reduced notation is convenient for deriving these relations for a broad class of models.

## Appendix 3

## Coarse-graining the detailed NADH redox model

We consider an NADH redox loop consisting of M reductases and N oxidases (*Figure 4—figure supplement 1*), each of which is described by the generalized enzyme kinetics (*Figure 3b*; *Equations S13-S16*). We coarse-grain this detailed NADH redox model by coarse-graining all oxidases into a single effective oxidase and all reductases into a single effective reductase (*Figure 4b*). We relate the kinetic rates of the coarse-grained model to those of the detailed model by keeping the global binding and unbinding fluxes and the global reaction fluxes through the oxidases and reductases the same as the detailed model.

We first coarse-grain the oxidases and reductases:

$$\left[\text{NADH} \cdot \text{Ox}\right] = \sum_{i=1}^{N} \left[\text{NADH} \cdot \text{Ox}_i\right], \quad \left[\text{NADH} \cdot \text{Re}\right] = \sum_{i=1}^{M} \left[\text{NADH} \cdot \text{Re}_i\right]. \tag{S19}$$

We require the global binding and unbinding fluxes of NADH to the effective oxidase and reductase to be equal to the sum of their binding and unbinding fluxes to all of the individual oxidases and reductases:

$$J_{\text{ox}}^{\text{b}} = \left[\text{NADH}_{\text{f}}\right] \sum_{i=1}^{N} k_{\text{ox}_i}^{\text{b}} = k_{\text{ox}}^{\text{b}} \left[\text{NADH}_{\text{f}}\right], \tag{S20}$$

$$J_{\text{ox}}^{\text{u}} = \sum_{i=1}^{N} k_{\text{ox}_i}^{\text{u}} \left[\text{NADH} \cdot \text{Ox}_i\right] = k_{\text{ox}}^{\text{u}} \left[\text{NADH} \cdot \text{Ox}\right], \tag{S21}$$

$$J_{\text{re}}^{\text{b}} = \left[\text{NADH}_{\text{f}}\right] \sum_{i=1}^{M} k_{\text{re}_i}^{\text{b}} = k_{\text{re}}^{\text{b}} \left[\text{NADH}_{\text{f}}\right], \tag{S22}$$

$$J_{\text{re}}^{\text{u}} = \sum_{i=1}^{M} k_{\text{re}_i}^{\text{u}} \left[\text{NADH} \cdot \text{Re}_i\right] = k_{\text{re}}^{\text{u}} \left[\text{NADH} \cdot \text{Re}\right], \tag{S23}$$

which leads to

$$k_{\text{ox}}^{\text{b}} = \sum_{i=1}^{N} k_{\text{ox}_i}^{\text{b}}, \quad k_{\text{re}}^{\text{b}} = \sum_{i=1}^{M} k_{\text{re}_i}^{\text{b}}, \tag{S24}$$

$$k_{\text{ox}}^{\text{u}} = \sum_{i=1}^{N} k_{\text{ox}_i}^{\text{u}} \frac{\left[\text{NADH} \cdot \text{Ox}_i\right]}{\left[\text{NADH} \cdot \text{Ox}\right]}, \quad k_{\text{re}}^{\text{u}} = \sum_{i=1}^{M} k_{\text{re}_i}^{\text{u}} \frac{\left[\text{NADH} \cdot \text{Re}_i\right]}{\left[\text{NADH} \cdot \text{Re}\right]}. \tag{S25}$$

We require the global forward and reverse reaction flux through the effective oxidase and reductase to be equal to the sum of the reaction fluxes through all of the individual oxidases and reductases:

$$J_{\text{ox}}^{+} = \sum_{i=1}^{N} r_{\text{ox}_i}^{+} \left[\text{NADH} \cdot \text{Ox}_i\right] = r_{\text{ox}}^{+} \left[\text{NADH} \cdot \text{Ox}\right], \tag{S26}$$

$$J_{\text{ox}}^{-} = \sum_{i=1}^{N} r_{\text{ox}_i}^{-} \left[\text{NAD}^{+} \cdot \text{Ox}_i\right] = r_{\text{ox}}^{-} \left[\text{NAD}^{+} \cdot \text{Ox}\right], \tag{S27}$$

which leads to

$$r_{\text{ox}}^{+} = \sum_{i=1}^{N} r_{\text{ox}_i}^{+} \frac{\left[\text{NADH} \cdot \text{Ox}_i\right]}{\left[\text{NADH} \cdot \text{Ox}\right]}, \tag{S28}$$

$$r_{\text{ox}}^{-} = \sum_{i=1}^{N} r_{\text{ox}_i}^{-} \frac{\left[\text{NAD}^{+} \cdot \text{Ox}_i\right]}{\left[\text{NAD}^{+} \cdot \text{Ox}\right]}. \tag{S29}$$

By applying the same procedure to $\mathrm{NAD}^+$, we can obtain the effective reduction rates $r_{\mathrm{re}}^+$, $r_{\mathrm{ox}}^-$ and the effective binding and unbinding rates of $\mathrm{NAD}^+$ : $k_{\mathrm{ox}}^{'\mathrm{b}}$, $k_{\mathrm{re}}^{'\mathrm{b}}$, $k_{\mathrm{ox}}^{'\mathrm{u}}$, $k_{\mathrm{re}}^{'\mathrm{u}}$. We omit the derivation here because these rates are not needed to infer ETC flux. We hence explicitly related the kinetic rates of the coarse-grained model (*Figure 4b*) to those of the detailed model (*Figure 4—figure supplement 1*). We note that under the generalized enzyme kinetics (*Figure 3b*; *Equations S13-S16*), all kinetic rates are considered to be general functions of enzyme concentrations, metabolite concentrations, and other factors, and thus, all of the rates in the coarse-grained model can also depend on all of those factors. These implicit variables can obey their own dynamical equations (*Equation S18*). The coarse-graining presented here is mathematically exact and independent of both the functional forms of these rates and the functional form of the dynamic equations of the implicit variables.

## Appendix 4

### Predicting the ETC flux using the coarse-grained NADH redox model

#### The coarse-grained NADH redox model

We start with the equations characterizing the dynamics of the coarse-grained NADH redox model as described in *Figure 4b*:

$$\frac{d[\text{NADH}\cdot\text{Re}]}{dt} = k_{\text{re}}^{\text{b}}\left[\text{NADH}_{\text{f}}\right] - k_{\text{re}}^{\text{u}}\left[\text{NADH}\cdot\text{Re}\right] + r_{\text{re}}^{+}\left[\text{NAD}^{+}\cdot\text{Re}\right] - r_{\text{re}}^{-}\left[\text{NADH}\cdot\text{Re}\right],\tag{S30}$$

$$\frac{d[\text{NADH}_{\text{f}}]}{dt} = k_{\text{re}}^{\text{u}}\left[\text{NADH}\cdot\text{Re}\right] + k_{\text{ox}}^{\text{u}}\left[\text{NADH}\cdot\text{Ox}\right] - k_{\text{re}}^{\text{b}}\left[\text{NADH}_{\text{f}}\right] - k_{\text{ox}}^{\text{b}}\left[\text{NADH}_{\text{f}}\right],\tag{S31}$$

$$\frac{d[\text{NADH}\cdot\text{Ox}]}{dt} = k_{\text{ox}}^{\text{b}}\left[\text{NADH}_{\text{f}}\right] - k_{\text{ox}}^{\text{u}}\left[\text{NADH}\cdot\text{Ox}\right] - r_{\text{ox}}^{+}\left[\text{NADH}\cdot\text{Ox}\right] + r_{\text{ox}}^{-}\left[\text{NAD}^{+}\cdot\text{Ox}\right],\tag{S32}$$

$$\frac{d[\text{NAD}^{+}\cdot\text{Ox}]}{dt} = k_{\text{ox}}^{\prime\text{b}}\left[\text{NAD}_{\text{f}}^{+}\right] - k_{\text{ox}}^{\prime\text{u}}\left[\text{NAD}^{+}\cdot\text{Ox}\right] + r_{\text{ox}}^{+}\left[\text{NADH}\cdot\text{Ox}\right] - r_{\text{ox}}^{-}\left[\text{NAD}^{+}\cdot\text{Ox}\right],\tag{S33}$$

$$\frac{d[\text{NAD}_{\text{f}}^{+}]}{dt} = k_{\text{re}}^{\prime\text{u}}\left[\text{NAD}^{+}\cdot\text{Re}\right] + k_{\text{ox}}^{\prime\text{u}}\left[\text{NAD}^{+}\cdot\text{Ox}\right] - k_{\text{re}}^{\prime\text{b}}\left[\text{NAD}_{\text{f}}^{+}\right] - k_{\text{ox}}^{\prime\text{b}}\left[\text{NAD}_{\text{f}}^{+}\right],\tag{S34}$$

$$\frac{d[\text{NAD}^{+}\cdot\text{Re}]}{dt} = k_{\text{re}}^{\prime\text{b}}\left[\text{NAD}_{\text{f}}^{+}\right] - k_{\text{re}}^{\prime\text{u}}\left[\text{NAD}^{+}\cdot\text{Re}\right] - r_{\text{re}}^{+}\left[\text{NAD}^{+}\cdot\text{Re}\right] + r_{\text{re}}^{-}\left[\text{NADH}\cdot\text{Re}\right],\tag{S35}$$

where $[\text{NADH}_{\text{f}}]$ and $[\text{NAD}_{\text{f}}^{+}]$ are the concentrations of free NADH and free NAD$^{+}$; $[\text{NADH}\cdot\text{Re}]$ and $[\text{NAD}^{+}\cdot\text{Re}]$ are concentrations of reductase-bound NADH and NAD$^{+}$; $[\text{NADH}\cdot\text{Ox}]$ and $[\text{NAD}^{+}\cdot\text{Ox}]$ are concentrations of oxidase-bound NADH and NAD$^{+}$; $k$ denotes binding (b) and unbinding (u) rates, with subscript re and ox denoting reductase and oxidase, respectively; $r_{\text{re}}^{+}$ and $r_{\text{re}}^{-}$ are the forward and reverse reaction rates of the reductase; $r_{\text{ox}}^{+}$ and $r_{\text{ox}}^{-}$ are the forward and reverse reaction rates of the oxidase. The reaction rates, and binding and unbinding rates, can be arbitrary functions of metabolite concentrations, enzyme concentrations, and other variables (such as membrane potential, oxygen concentration, etc, each of which can obey their own dynamical equations).

#### Predicting the ETC flux

The flux through the ETC is

$$J_{\text{ox}} \equiv r_{\text{ox}}^{+}\left[\text{NADH}\cdot\text{Ox}\right] - r_{\text{ox}}^{-}\left[\text{NAD}^{+}\cdot\text{Ox}\right].\tag{S36}$$

At steady-state (or in the quasistatic limit), all the time derivatives are zero. Setting d[NADH · Ox]/dt (*Equation S32*) to zero, we obtained the steady-state flux through the ETC:

$$J_{\text{ox}} = k_{\text{ox}}^{\text{b}}\left[\text{NADH}_{\text{f}}\right] - k_{\text{ox}}^{\text{u}}\left[\text{NADH}\cdot\text{Ox}\right].\tag{S37}$$

Setting d[NADH$_{\text{f}}$]/dt (*Equation S31*) to zero gives:

$$\left(k_{\text{ox}}^{\text{b}} + k_{\text{re}}^{\text{b}}\right)\left[\text{NADH}_{\text{f}}\right] = k_{\text{ox}}^{\text{u}}\left[\text{NADH}\cdot\text{Ox}\right] + k_{\text{re}}^{\text{u}}\left[\text{NADH}\cdot\text{Re}\right],\tag{S38}$$

and using:

$$\left[\text{NADH}\cdot\text{Re}\right] + \left[\text{NADH}\cdot\text{Ox}\right] = \left[\text{NADH}_{\text{b}}\right],\tag{S39}$$

from which we solved for $[\text{NADH}\cdot\text{Ox}]$ :

$$\left[\text{NADH}\cdot\text{Ox}\right] = \frac{k_{\text{re}}^{\text{b}} + k_{\text{ox}}^{\text{b}}}{k_{\text{ox}}^{\text{u}} - k_{\text{re}}^{\text{u}}}\left[\text{NADH}_{\text{f}}\right] - \frac{k_{\text{re}}^{\text{u}}}{k_{\text{ox}}^{\text{u}} - k_{\text{re}}^{\text{u}}}\left[\text{NADH}_{\text{b}}\right].\tag{S40}$$

Substituting $[\text{NADH}\cdot\text{Ox}]$ in *Equation S37* with *Equation S40*, we obtained our central result:

$$J_{\text{ox}} = \tilde{r}_{\text{ox}}\left[\text{NADH}_{\text{f}}\right].\tag{S41}$$

From *Equation S41*, we see that the flux through the ETC is a product of the turnover rate of free NADH, $\tilde{r}_{\text{ox}}$, and the concentration of free NADH, $[\text{NADH}_{\text{f}}]$, where

$$\tilde{r}_{\text{ox}} = \alpha \left( \beta - \beta_{\text{eq}} \right), \tag{S42}$$

and

$$\alpha = \frac{k_{\text{ox}}^{\text{u}} \, k_{\text{re}}^{\text{u}}}{k_{\text{ox}}^{\text{u}} - k_{\text{re}}^{\text{u}}}, \tag{S43}$$

where we defined the NADH bound ratio and its equilibrium counterpart as:

$$\beta = \frac{\left[ \text{NADH}_{\text{b}} \right]}{\left[ \text{NADH}_{\text{f}} \right]}, \tag{S44}$$

$$\beta_{\text{eq}} = \beta_{\text{eq}}^{\text{ox}} + \beta_{\text{eq}}^{\text{re}}, \tag{S45}$$

$$\beta_{\text{eq}}^{\text{ox}} = \frac{k_{\text{ox}}^{\text{b}}}{k_{\text{ox}}^{\text{u}}}, \tag{S46}$$

$$\beta_{\text{eq}}^{\text{re}} = \frac{k_{\text{re}}^{\text{b}}}{k_{\text{re}}^{\text{u}}}. \tag{S47}$$

## Appendix 5

### Flux inference procedures using the coarse-grained NADH redox model

*Equations S41-S42*, or equivalently *Equations 5a-c* from the main text, can be used to infer the flux through the ETC, $J_{\mathrm{ox}}$, from FLIM measurements of NADH across a wide range of metabolic perturbations (*Figure 5—figure supplement 1*). To do so, we infer the turnover rate of free NADH, $\tilde{r}_{\mathrm{ox}}$, and the concentration of free NADH, $[\mathrm{NADH_f}]$. The product of $\tilde{r}_{\mathrm{ox}}$ and $[\mathrm{NADH_f}]$ gives $J_{\mathrm{ox}}$. $[\mathrm{NADH_f}]$ can be obtained using *Equation S2* (*Figure 5—figure supplement 1*). In this section, we describe two procedures to obtain $\tilde{r}_{\mathrm{ox}}$: one from the measurement of NADH bound ratio $\beta$, and the other from the measurement of NADH long fluorescence lifetime $\tau_{\mathrm{l}}$.

### Inferring $\tilde{r}_{\mathrm{ox}}$ from NADH bound ratio $\beta$

*Equation S42*, $\tilde{r}_{\mathrm{ox}} = \alpha\left(\beta - \beta_{\mathrm{eq}}\right)$, provides a method to obtain $\tilde{r}_{\mathrm{ox}}$. We measure the NADH bound ratio, $\beta$, using $\beta = f/(1-f)$, where $f$ is the NADH bound fraction obtained by fitting the fluorescence decay curve of NADH (see Materials and methods). We obtain the equilibrium bound ratio, $\beta_{\mathrm{eq}}$, by dropping the oxygen level to the lowest achievable value with our setup: $[\mathrm{O_2}] = 0.26 \pm 0.04\ \mu\mathrm{M}$, and assuming $\beta_{\mathrm{eq}}$ does not change with oxygen levels. Note that $\beta_{\mathrm{eq}}$ does change with drug perturbations, and therefore needs to be separately determined for each condition (*Figure 5—figure supplement 1h*). We obtain $\alpha$ using direct measurement of $J_{\mathrm{ox}}$ from OCR measurements:

$$\alpha = \frac{J_{\mathrm{ox}}}{(\beta - \beta_{\mathrm{eq}})[\mathrm{NADH_f}]} = 2\frac{\mathrm{OCR}}{(\beta - \beta_{\mathrm{eq}})[\mathrm{NADH_f}]V_{\mathrm{m}}}, \tag{S48}$$

where $V_{\mathrm{m}} = 9.5 \times 10^4\ \mu\mathrm{m}^3$ is the average volume of mitochondria per oocyte approximated from the area fraction of mitochondria based on the segmentation, where the mitochondrial area fraction is estimated at 46% and oocyte volume at 2×10⁵ μm³. Using $\mathrm{OCR} = 2.68 \pm 0.06\ \mathrm{fmol/s}$ per oocyte in the control condition (AKSOM media at 50 μM oxygen level), we get $\alpha = 5.4 \pm 0.2\ \mathrm{s}^{-1}$. $\alpha$ is approximated as a constant that does not vary with perturbations, hence $\alpha$ calibrated at one condition can be used for all other conditions (as confirmed by the agreement between FLIM based inference and OCR measurements in *Figures 5 and 8*).

Once $\alpha$ is calibrated at the control condition using *Equation S48*, and $\beta_{\mathrm{eq}}$ is determined from an oxygen drop experiment, then subsequent FLIM measurements of $\beta$ and $[\mathrm{NADH_f}]$ can be used with *Equation S42 and S41* to determine the absolute value of $\tilde{r}_{\mathrm{ox}}$ and $J_{\mathrm{ox}}$ for all conditions (*Figure 5*).

### Inferring $\tilde{r}_{\mathrm{ox}}$ from NADH long fluorescence lifetime $\tau_{\mathrm{l}}$

In this section, we derive an alternative procedure for determining the turnover rate of free NADH $\tilde{r}_{\mathrm{ox}}$, and hence $J_{\mathrm{ox}}$, using changes in the NADH long fluorescence lifetime. The NADH long fluorescence lifetime, $\tau_{\mathrm{l}}$, is associated with enzyme-bound NADH (*Sharick et al., 2018*). In the coarse-grained NADH redox model described above, and in *Figure 4b*, the enzyme-bound NADH consists of reductase-bound NADH ($[\mathrm{NADH \cdot Re}]$) and oxidase-bound NADH ($[\mathrm{NADH \cdot Ox}]$). We therefore assume that the experimentally measured NADH long lifetime, $\tau_{\mathrm{l}}$, is a linear combination of the lifetimes of $[\mathrm{NADH \cdot Ox}]$ and $[\mathrm{NADH \cdot Re}]$:

$$\tau_{\mathrm{l}} = \tau_{\mathrm{ox}}\frac{[\mathrm{NADH \cdot Ox}]}{[\mathrm{NADH \cdot Ox}] + [\mathrm{NADH \cdot Re}]} + \tau_{\mathrm{re}}\frac{[\mathrm{NADH \cdot Re}]}{[\mathrm{NADH \cdot Ox}] + [\mathrm{NADH \cdot Re}]}, \tag{S49}$$

where $\tau_{\mathrm{ox}}$ and $\tau_{\mathrm{re}}$ are the fluorescence lifetimes corresponding to the oxidase-bound NADH and reductase-bound NADH, respectively. Solving for $[\mathrm{NADH \cdot Ox}]$ and $[\mathrm{NADH \cdot Re}]$ as a function of $\beta$ using *Equations S39-S40 and S44*, and substituting into *Equation S49*, we predict that the NADH long fluorescence lifetime $\tau_{\mathrm{l}}$ is linearly related to the inverse of the NADH bound ratio $1/\beta$:

$$\tau_{\mathrm{l}} = A\frac{1}{\beta} + B, \tag{S50}$$

with

$$A = (\tau_{\mathrm{ox}} - \tau_{\mathrm{re}})\frac{k_{\mathrm{ox}}^{\mathrm{b}} + k_{\mathrm{re}}^{\mathrm{b}}}{k_{\mathrm{ox}}^{\mathrm{u}} - k_{\mathrm{re}}^{\mathrm{u}}}, \tag{S51}$$

$$B = \frac{k_{\text{ox}}^{\text{u}} \tau_{\text{re}} - k_{\text{re}}^{\text{u}} \tau_{\text{ox}}}{k_{\text{ox}}^{\text{u}} - k_{\text{re}}^{\text{u}}}. \tag{S52}$$

The predicted linear relationship between $\tau_l$ and $1/\beta$ is empirically observed during oxygen drop experiments, as shown in *Figure 6a* and *Figure 6—figure supplement 1*. This is a self-consistency check that argues for the validity of the assumption in *Equation S49*.

At equilibrium, when there is no flux through the ETC (i.e., $J_{\text{ox}} = 0$), *Equation S50* gives:

$$\tau_{\text{eq}} = A \frac{1}{\beta_{\text{eq}}} + B, \tag{S53}$$

where $\tau_{\text{eq}}$ is the NADH long lifetime at equilibrium. Solving for $\beta$ and $\beta_{\text{eq}}$ as a function of $\tau_l$ and $\tau_{\text{eq}}$ from *Equations S50 and S53* and substituting into *Equation S42*, we obtain $\tilde{r}_{\text{ox}}$ in terms of $\tau_l$ :

$$\tilde{r}_{\text{ox}} = \alpha \frac{A}{\tau_{\text{eq}} - B} \left( \frac{\tau_{\text{eq}} - \tau_l}{\tau_l - B} \right), \tag{S54}$$

where $A$ and $B$ are the slope and offset of the linear relation between $\tau_l$ and $1/\beta$ in *Equation S50*.

We experimentally measured $A$ and $B$ for each oocyte from the slope and offset of a linear fit between $\tau_l$ and $1/\beta$ during oxygen drop experiments across all drug perturbations (*Figure 6a*; *Figure 6—figure supplement 1*). We obtained the equilibrium long lifetime, $\tau_{\text{eq}}$, by FLIM measurements at the lowest achievable oxygen level in our set up: $[O_2] = 0.26 \pm 0.04 \,\mu\text{M}$. Once $A$, $B$, and $\tau_{\text{eq}}$ are measured, *Equation S54* can be used to determine $\tilde{r}_{\text{ox}}$ from FLIM measurements of $\tau_l$. If $\alpha$ is not known, this procedure can only be used to obtain $\tilde{r}_{\text{ox}}$ up to a constant of proportionality. If $\alpha$ is independently measured from *Equation S48* at one condition, then *Equation S54* can be used to determine the absolute value of $\tilde{r}_{\text{ox}}$ for all conditions (*Figure 6b*).

As described in the main text, $\tilde{r}_{\text{ox}}$ inferred from $\tau_l$ using *Equation S54* produces the same results as $\tilde{r}_{\text{ox}}$ inferred from $\beta$ using *Equation S42* (*Figure 6b*). The agreement between these two methods is a strong self-consistency check of the NADH redox model.

## Accounting for NADPH and other background fluorescence

*Equation S41* provides a method to infer ETC flux since all factors in it, except for the constant of proportionality, $\alpha$, depend only on $[\text{NADH}_b]$ and $[\text{NADH}_f]$, which can be measured from FLIM of NADH. One potential complication with this procedure is that NADPH, another autofluorescent electron carrier, shares a similar fluorescence spectrum with NADH, resulting in a mixed NAD(P)H signal from the autofluorescence measurement. While NADH concentration is 40 times greater than the concentration of NADPH for the whole mouse oocytes (*Bustamante et al., 2017*), and presumably even higher for mitochondria, NADPH concentration can be comparable to that of NADH for other cell types such as tissue culture cells (*Park et al., 2016*). In this section, we generalize *Equation S41* to predict the ETC flux by explicitly considering the potential contributions of other fluorescence species, such as NADPH, to the measured autofluorescence signal.

We start from the fact that concentrations of bound and free fluorescent species measured from FLIM using *Equations S2-S3*, $[\text{N}_f]$ and $[\text{N}_b]$, could be different from the actual concentrations of free and bound NADH, $[\text{NADH}_f]$ and $[\text{NADH}_b]$. If the signal from NADPH and other additional fluorescence species is additive, then:

$$[\text{N}_f] = [\text{NADH}_f] + C_f, \tag{S55}$$

$$[\text{N}_b] = [\text{NADH}_b] + C_b, \tag{S56}$$

where $C_f$ and $C_b$ are the non-NADH contributions to the measured concentrations of free and bound fluorescent species. We substitute *Equations S55-S56* to the predicted ETC flux in *Equation S41* and obtain

$$J_{\text{ox}} = \alpha \left( \frac{[\text{N}_b] - C_b}{[\text{N}_f] - C_f} - \beta_{\text{eq}} \right) \left( [\text{N}_f] - C_f \right). \tag{S57}$$

Rearranging, we obtain

$$J_{\text{ox}} = \alpha \left( \beta_{\text{N}} - \beta_{\text{N,eq}} \right) [\text{N}_f], \tag{S58}$$

where

$$\beta_{\mathrm{N}} = \frac{[\mathrm{N_b}]}{[\mathrm{N_f}]}, \tag{S59}$$

$$\beta_{\mathrm{N,eq}} = \beta_{\mathrm{eq}} + \left( \frac{C_\mathrm{b}}{C_\mathrm{f}} - \beta_{\mathrm{eq}} \right) \frac{C_\mathrm{f}}{[\mathrm{N_f}]}. \tag{S60}$$

Comparing *Equation S58* with *Equation S41*, we notice that the background fluorescence does not change the form of the equation of the predicted ETC flux because the concentrations of the background fluorescent species are incorporated into the equilibrium bound ratio $\beta_{\mathrm{N,eq}}$. If $\beta_{\mathrm{N,eq}}$ can be reliably measured, the background fluorescence will not affect the flux inference procedures. In other words, $[\mathrm{N_f}]$ and $[\mathrm{N_b}]$ can be used for flux inference in place of $[\mathrm{NADH_f}]$ and $[\mathrm{NADH_b}]$ in *Equations S41-S42*. Therefore, an additive offset to the measured concentrations of free and bound species will not affect the flux inference procedure, whether that additive offset comes from NADPH or from other sources of fluorescent background.

Alternatively, if the signal from background fluorescence changes proportionally with NADH, then:

$$\left[ \mathrm{N_f} \right] = \left[ \mathrm{NADH_f} \right] + C_\mathrm{f}' \left[ \mathrm{NADH_f} \right] = \left( 1 + C_\mathrm{f}' \right) \left[ \mathrm{NADH_f} \right], \tag{S61}$$

$$\left[ \mathrm{N_b} \right] = \left[ \mathrm{NADH_b} \right] + C_\mathrm{b}' \left[ \mathrm{NADH_b} \right] = \left( 1 + C_\mathrm{b}' \right) \left[ \mathrm{NADH_b} \right], \tag{S62}$$

we have

$$J_{\mathrm{ox}} = \alpha_{\mathrm{N}} \left( \beta_{\mathrm{N}} - \beta_{\mathrm{N,eq}}' \right) \left[ \mathrm{N_f} \right], \tag{S63}$$

where

$$\alpha_{\mathrm{N}} = \frac{\alpha}{1 + C_\mathrm{b}'}, \tag{S64}$$

$$\beta_{\mathrm{N}} = \frac{[\mathrm{N_b}]}{[\mathrm{N_f}]}, \tag{S65}$$

$$\beta_{\mathrm{N,eq}}' = \frac{1 + C_\mathrm{b}'}{1 + C_\mathrm{f}'} \, \beta_{\mathrm{eq}}. \tag{S66}$$

Comparing *Equation S63* with *Equation S41*, we again obtained the same form for $J_{\mathrm{ox}}$, but with a rescaled $\alpha_{\mathrm{N}}$. Therefore, a background fluorescence signal that changes proportionally with NADH will not affect the flux inference procedure, whether that background comes from NADPH or from other sources.

## Appendix 6

## Spatial gradient of mitochondrial metabolism in mouse oocytes

### $\beta_{eq}$ is uniform within the oocyte

To obtain subcellular ETC flux as a function of distance to the oocyte's center using *Equations 5a-c* in the main text, we need to know the spatial variation of $\beta_{eq}$. While the NADH bound ratio at the lowest oxygen level gives a good approximation for the average $\beta_{eq}$ of the cell (*Figure 5—figure supplement 1h*), subpopulations of mitochondria closer to the cell periphery are exposed to slightly higher oxygen level than those away from the cell periphery, obscuring the determination of the spatial variation of $\beta_{eq}$ from oxygen drop experiment. Hence to obtain the spatial variation of $\beta_{eq}$ throughout the oocyte, we inhibited the ETC completely using 15 µM of rotenone, an inhibitor of complex I in the ETC, for an extended period of time until the NADH bound ratio reaches the lowest level. We then fitted the NADH decay curves from mitochondrial pixels within equal-distanced concentric rings (*Figure 9—figure supplement 1a*) to obtain $\beta_{eq}$ as a function of distance from the oocyte's center (*Figure 9—figure supplement 1b*). A linear fit yielded a slope of 0.001±0.0012 (SEM), which is statistically indistinguishable from 0 (*p*=0.42). Therefore, the resulting $\beta_{eq}$ is uniform throughout the oocyte and is equal to the average $\beta_{eq}$ obtained by fitting the decay curve from all mitochondrial pixels in the oocyte at the lowest oxygen level (*Figure 5—figure supplement 1h*). Hence, we used a constant $\beta_{eq}$ throughout the oocyte to compute the subcellular ETC flux (*Figure 9d*).

### Subcellular spatial gradient of mitochondrial membrane potential

As shown in the main text, we observed a strong spatial gradient of the intensity of TMRM in mitochondria in oocytes. TMRM is a potential-sensitive dye that preferentially accumulates in mitochondria with higher membrane potential (*Figure 9g and h*). To test whether this spatial gradient is due to the subcellular variation of mitochondrial membrane potential or the variation in mitochondrial mass, we labelled mitochondria with a potential-insensitive dye MitoTracker Red FM to quantify mitochondrial mass, together with TMRM. We did not observe a strong gradient of MitoTracker intensity (*Figure 9—figure supplement 2b, f*) as compared to TMRM intensity (*Figure 9—figure supplement 2a, e*) within the same oocyte, indicating the mitochondrial mass is uniformly distributed. We further normalized the TMRM intensity by the MitoTracker intensity, and observed a strong spatial gradient of the ratio (*Figure 9—figure supplement 2c, g*). These results suggest that the spatial gradient of TMRM is due to the variation of mitochondrial membrane potential, rather than the variation of mitochondrial mass. Finally, to test the robustness of the result, we used an alternative potential-sensitive dye JC-1, and observed a similar spatial gradient of mitochondrial membrane potential (*Figure 9—figure supplement 2d, h*). Taken together, these results show that the subcellular spatial gradient of mitochondrial membrane potential is a robust observation that does not depend on the variation of mitochondrial mass or the type of dye used.

# Appendix 7

## Flux prediction for a NADH redox model with each enzyme described by the reversible Michaelis-Menten kinetics

In this section, we derive the ETC flux for a NADH redox model where each of the N oxidase and M reductase obeys reversible Michaelis-Menten kinetics (*Equations S9-S11*). We achieve this by reducing the flux prediction of the generalized enzyme kinetics to that of the reversible Michaelis-Menten kinetics.

We first consider an NADH redox model with a single oxidase and a single reductase, each of which obeys reversible Michaelis-Menten kinetics (i.e., N=M=1). The flux through the oxidase is:

$$J_{\text{ox}} = k_1 \left[ \text{Ox}_1 \right] \left[ \text{NADH}_{\text{f}} \right] - k_{-1} \left[ \text{NADH} \cdot \text{Ox}_1 \right]. \tag{S67}$$

We show that *Equations S41-S43* that characterize the flux of the generalized NADH redox model can be reduced to *Equation S67* that characterizes the flux of the reversible Michaelis-Menten model in the limit:

$$k_{\text{re}}^{\text{u}} \gg k_{\text{ox}}^{\text{u}}, k_{\text{re}}^{\text{u}} \gg k_{\text{re}}^{\text{b}}, \quad k_{\text{re}}^{\text{u}} \gg k_{\text{ox}}^{\text{b}}. \tag{S68}$$

In this limit, we have from *Equation S43*:

$$\alpha \approx \alpha_{\text{MM}} = -k_{\text{ox}}^{\text{u}} = -k_{-1}. \tag{S69}$$

where 'MM' stands for 'Michaelis-Menten.' Similarly, from *Equation S45-S47*, we have

$$\beta_{\text{eq}} \approx \beta_{\text{eq}}^{\text{MM}} = \frac{k_{\text{ox}}^{\text{b}}}{k_{\text{ox}}^{\text{u}}} = \frac{k_1 \left[ \text{Ox}_1 \right]}{k_{-1}}. \tag{S70}$$

From *Equation S40*, we have in this limit:

$$\left[ \text{NADH}_{\text{b}} \right] \approx \left[ \text{NADH} \cdot \text{Ox}_1 \right]. \tag{S71}$$

Substituting the expressions for $\alpha_{\text{MM}}$, $\beta_{\text{eq}}^{\text{MM}}$, and $\left[ \text{NADH}_{\text{b}} \right]$ from *Equations S69-S71* to the predicted ETC flux in *Equation S41*, we obtain

$$J_{\text{ox}} = \alpha_{\text{MM}} \left( \beta - \beta_{\text{eq}}^{\text{MM}} \right) \left[ \text{NADH}_{\text{f}} \right] = k_1 \left[ \text{Ox}_1 \right] \left[ \text{NADH}_{\text{f}} \right] - k_{-1} \left[ \text{NADH} \cdot \text{Ox}_1 \right]. \tag{S72}$$

Thus, we have shown that the flux of the generalized NADH redox model reduces to the flux of the reversible Michaelis-Menten model in the limit where the unbinding rate of NADH from the reductase is much faster than any other rates in the model.

We note that the predicted flux-concentration relation for the reversible Michaelis-Menten model (*Equation S72*) remains exactly the same as the generalized model (*Equation S41*), but with different expressions for $\alpha$ and $\beta_{\text{eq}}$ as expressed in *Equations S69-S70*.

Next, we generalize the results in *Equation S72* to a detailed model with N oxidase and M reductase, each of which is described by the reversible Michaelis-Menten kinetics. Unpacking the coarse-grained binding and unbinding rates from *Equations S24-S25*, we obtain

$$\alpha_{\text{MM}} = -\sum_{i=1}^{\text{N}} k_{-1,i} \frac{\left[ \text{NADH} \cdot \text{Ox}_i \right]}{\left[ \text{NADH} \cdot \text{Ox} \right]}, \tag{S73}$$

$$\beta_{\text{eq}}^{\text{MM}} = \frac{\sum_{i=1}^{\text{N}} k_{1,i} \left[ \text{Ox}_i \right]}{\sum_{i=1}^{\text{N}} k_{-1,i} \frac{\left[ \text{NADH} \cdot \text{Ox}_i \right]}{\left[ \text{NADH} \cdot \text{Ox} \right]}}, \tag{S74}$$

where $k_{1,i}$ and $k_{-1,i}$ denote the binding and unbinding rates of NADH to the $i$th oxidase.

## Connecting the NADH redox model to detailed biophysical models of mitochondrial metabolism

In this section, we show that the coarse-grained NADH redox model described above, and in *Figure 4b* of the main text, can be directly related to detailed biophysical models of mitochondrial

metabolism, including previously published models (*Beard, 2005*; *Korzeniewski and Zoladz, 2001*; *Hill, 1977*; *Jin and Bethke, 2002*; *Chang et al., 2011*).

In mitochondria, NADH oxidation is catalyzed by complex I of the ETC, which has the overall reaction:

$$H^+ + NADH + Q \rightleftarrows NAD^+ + QH_2 + 4\Delta H^+, \tag{S75}$$

where two electrons are transferred from NADH to ubiquinone Q, and four protons are pumped out of the mitochondrial matrix. To connect our model with detailed model of complex I, we rewrite the flux through the ETC:

$$J_{ox} = r_{ox}^+ \left[ NADH \cdot Ox \right] - r_{ox}^- \left[ NAD^+ \cdot Ox \right], \tag{S76}$$

using

$$\left[ NADH \cdot Ox \right] = \alpha \left( \frac{\beta_{eq}^{re}}{k_{ox}^u} + \frac{\beta_{eq}^{ox}}{k_{re}^u} - \frac{\beta}{k_{ox}^u} \right) \left[ NADH_f \right], \tag{S77}$$

$$\left[ NAD^+ \cdot Ox \right] = \alpha' \left( \frac{\beta_{eq}'^{re}}{k_{ox}'^u} + \frac{\beta_{eq}'^{ox}}{k_{re}'^u} - \frac{\beta'}{k_{ox}'^u} \right) \left[ NAD_f^+ \right], \tag{S78}$$

where

$$\alpha = \frac{k_{ox}^u k_{re}^u}{k_{ox}^u - k_{re}^u}, \ \beta = \frac{[NADH_b]}{[NADH_f]}, \ \beta_{eq}^{ox} + \beta_{eq}^{re} = \beta_{eq}, \ \beta_{eq}^{ox} = \frac{k_{ox}^b}{k_{ox}^u}, \ \beta_{eq}^{re} = \frac{k_{re}^b}{k_{re}^u}, \tag{S79}$$

$$\alpha' = \frac{k_{ox}'^u k_{re}'^u}{k_{ox}'^u - k_{re}'^u}, \ \beta = \frac{[NAD_b^+]}{[NAD_f^+]}, \ \beta_{eq}'^{ox} + \beta_{eq}'^{re} = \beta_{eq}', \ \beta_{eq}'^{ox} = \frac{k_{ox}'^b}{k_{ox}'^u}, \ \beta_{eq}'^{re} = \frac{k_{re}'^b}{k_{re}'^u}, \tag{S80}$$

$$J_{ox} = \tilde{r}_{ox}^+ [NADH_f] - \tilde{r}_{ox}^- [NAD_f^+] = \tilde{r}_{ox} [NADH_f], \tag{S81}$$

where

$$\tilde{r}_{ox}^+ = \alpha \left( \frac{\beta_{eq}^{re}}{k_{ox}^u} + \frac{\beta_{eq}^{ox}}{k_{re}^u} - \frac{\beta}{k_{ox}^u} \right) r_{ox}^+, \tag{S82}$$

$$\tilde{r}_{ox}^- = \alpha' \left( \frac{\beta_{eq}'^{re}}{k_{ox}'^u} + \frac{\beta_{eq}'^{ox}}{k_{re}'^u} - \frac{\beta'}{k_{ox}'^u} \right) r_{ox}^-, \tag{S83}$$

$$\tilde{r}_{ox} = \alpha \left( \beta - \beta_{eq} \right) = \left( \tilde{r}_{ox}^+ + \tilde{r}_{ox}^- \frac{1+\beta}{1+\beta'} \right) - \frac{N}{[NADH_f]} \tilde{r}_{ox}^-. \tag{S84}$$

The last equality in *Equation S84* is obtained by assuming that the total concentration of NADH plus NAD$^+$ is constant:

$$N = \left[ NADH_f \right] + \left[ NAD_f^+ \right] + \left[ NADH_b \right] + \left[ NAD_b^+ \right]. \tag{S85}$$

*Equation S81* allows us to connect our coarse-grained model to previously published detailed models of complex I. By equating the flux through complex I, $J_{CI}$, in previous models to the flux through the ETC in our NADH redox model, $J_{ox}$, we can determine $\tilde{r}_{ox}$ (and $\tilde{r}_{ox}^+$ and $\tilde{r}_{ox}^-$) in terms of variables defined in those more detailed models. In *Appendix 7—table 1*, we summarize the relationship between the NADH redox model and several previously published models of complex I.

**Appendix 7—table 1.** Connection of the NADH redox model to detailed models of complex I. $\Delta G_H$ is the proton motive force. $\Delta G_{CI}$ is the free energy difference of the reaction at complex I. $\Delta G_{0,CI}$ is the standard free energy difference of the reaction at complex I. [Q], [QH$_2$], and [Q$_T$] are the concentrations of the oxidized, reduced, and total ubiquinone concentrations.

| Model | Flux | $\tilde{r}_{ox}^{+}$ | $\tilde{r}_{ox}^{-}$ | $\tilde{r}_{ox}$ |
|---|---|---|---|---|
| Beard, 2005 | $\begin{aligned} J_{CI} &= \tilde{r}_{ox}^{+}\left[NADH_f\right] \\ &\quad -\tilde{r}_{ox}^{-}\left[NAD^+\right] \end{aligned}$ | $\tilde{r}_{ox}^{+} = X_{C1}e^{\frac{\Delta\tilde{G}_{CI}}{RT}}$ | $\tilde{r}_{ox}^{-} = X_{C1}$ | $\tilde{r}_{ox} = \left(\tilde{r}_{ox}^{+} + \tilde{r}_{ox}^{-}\frac{1+\beta}{1+\beta'}\right) - \frac{N}{[NADH_f]}\tilde{r}_{ox}^{-}$ |
| Chang et al., 2011; Jin and Bethke, 2002 | $J_{CI} = \tilde{r}_{ox}\left[NADH_f\right]$ | N/A | N/A | $\begin{aligned} \tilde{r}_{ox} = V_{max} &\left(\frac{[N_T]}{[N_T]+K_{S,D}}\right)\left(\frac{[Q_T]}{[Q_T]+K_{S,A}}\right) \times \\ &\left(\frac{[Q]}{[Q]+[QH_2]K_{R,A}}\right)\left(\frac{1}{[NADH_f]+[NAD_f^+]K_{R,D}}\right) \times \\ &\left(1-e^{-\frac{\Delta G_{CI}}{RT}}\right) \end{aligned}$ |
| Hill, 1977 | $\begin{aligned} J_{CI} &= J_{max} \\ &\times\left(1-e^{\frac{-\Delta G_{CI}}{RT}}\right) \end{aligned}$ | N/A | N/A | $\tilde{r}_{ox} = J_{max}\left(1-e^{\frac{-\Delta G_{CI}}{RT}}\right) / \left[NADH_f\right]$ |
| Korzeniewski and Zoladz, 2001 | $J_{CI} = k_{CI}\Delta G_{CI}$ | N/A | N/A | $\tilde{r}_{ox} = k_{CI}\Delta G_{CI} / \left[NADH_f\right]$ |

$\Delta G_{CI} = -\left[\Delta G_{0,CI} + 4\Delta G_H - RT\ln\left(\frac{[H^+]}{10^{-7}}\right) - RT\ln\left(\frac{[Q]}{[QH_2]}\right) - RT\ln\left(\frac{[NADH_f]}{[NAD_f^+]}\right)\right];$

$\Delta\tilde{G}_{CI} = \Delta G_{CI} - RT\ln\left(\frac{[NADH_f]}{[NAD_f^+]}\right); \Delta G_{0,CI} = -69.37 \text{ kJ/mol}$

$\left[N_T\right] = \left[NADH_f\right] + \left[NAD_f^+\right], \left[Q_T\right] = \left[Q\right] + \left[QH_2\right]$

