## [Editor Report]

This paper describes the derivation and validation of a coarse-grained model to measure mitochondrial metabolism at cellular and subcellular resolution by exploiting fluorescence lifetime imaging of NADH. This technique is applied to mouse oocytes subjected to a variety of metabolic stresses and to human tissue culture cells, revealing spatial gradients in mitochondrial NADH oxidation. This method represents an exciting new approach to quantifying mitochondrial electron transport chain rates and provides for the first time a method to study mitochondrial metabolic flux with subcellular resolution.

---

## [Decision Letter]

[Editors’ note: the authors submitted for reconsideration following the decision after peer review. What follows is the decision letter after the first round of review.]

Thank you for submitting your work entitled "Coarse-grained model of mitochondrial metabolism enables subcellular flux inference from fluorescence lifetime imaging of NADH" for consideration by *eLife*. Your article has been reviewed by 4 peer reviewers, one of whom is a member of our Board of Reviewing Editors, and the evaluation has been overseen by a Senior Editor. The following individuals involved in review of your submission have agreed to reveal their identity: Jason W. Locasale (Reviewer #2); Edmund J. Crampin (Reviewer #3).

We are sorry to say that, after consultation with the reviewers, we have decided that your work will not be considered further for publication by *eLife* at this time. The reviewers raised three major issues regarding the clarity of the manuscript, the validity of the model, and the potential of the methodology to be used to generate biological insights in oocytes or additional cell systems. There were several concerns about the validity of the assumptions of the model, most notably the concern that as the coarse-grained model describes a substrate-enzyme binding reaction with a rate that depends only on the substrate concentration, this model would be valid only under conditions in which enzyme concentrations were orders of magnitude higher than substrate concentration. More broadly, the reviewers raised questions as to whether the assumptions in the model are valid in other systems and whether the methodology will be broadly applicable for the generation of novel biological insights. Given these significant concerns, we are returning the manuscript reviews to you so that they may guide you as you seek publication elsewhere. Should you wish to resubmit at a future date, we would be happy to reconsider a revised version that (1) more clearly justifies the validity of the approach and its assumptions and (2) provides an application of the new method to generate new biological insight.

*Reviewer #1:*

In this manuscript, the authors present a model to relate FLIM measurements to mitochondrial metabolic fluxes. Using mouse oocytes, which have little NADPH, the authors develop a coarse-grained model to infer mitochondrial NADH oxidation by exploiting NAD(P)H FLIM. Using this approach, the authors uncover regional variation in mitochondrial fluxes in mouse oocytes. The modeled mitochondrial flux shows a strong negative correlation with mitochondrial membrane potential and no correlation with mitochondrial content. While this is not the first paper to use NAD(P)H FLIM to show subcellular metabolic variability, this manuscript does present a model to connect NAD(P)H FLIM to mitochondrial redox cycles. Therefore, the major utility of the model lies in its ability to provide subcellular information about mitochondrial NAD(P)H oxidation. The authors provide a comprehensive and accessible discussion of the assumptions, caveats, and conclusions enabled by their modeling. At present, however, it is not clear to this reviewer how generalizable this method will prove beyond mouse oocytes. This concern stems from the potential difficulty in establishing key parameters of the model in other cell types in which assumptions safely made in mouse oocytes may not be appropriate.

To demonstrate the utility of their model, the authors should test key parameters in at least one additional cell type. In particular, the following issues should be addressed:

1. A key requirement of the model is the ability to determine the equilibrium NADH bound ratio. Here, the authors use low oxygen (or rotenone) to establish this parameter. Will this be feasible in other cell types, for example those with active NNT?

2. The authors note that the confounding signal from NADPH can be ignored in mouse oocytes, which have 40-fold higher NADH than NADPH. How generalizable is this? Will other mammalian cell types be amenable to this method?

3. The authors test the assumption that NADH signal originates in the mitochondria by comparing with mitotracker signal under control conditions. This should also be repeated for key conditions (e.g. low oxygen or oxamate treatment).

*Reviewer #2:*

In the manuscript "Coarse-grained model of mitochondrial metabolism enables subcellular flux inference from fluorescence lifetime imaging of NADH", the authors use fluorescence imaging to estimate NADH/NAD turnover flux and electron transfer rate in the mitochondria of mouse oocytes. Because of high spatial resolution of microscopy, the authors could also observe significant subcellular spatial gradient of oxidative flux in oocytes.

The fluorescence imaging and quantification of flux are generally solid and convincing, but there are issues that need to be addressed.

– In figure 1c, the author estimated two parameters τl and τs under different oxygen levels. They should be constant in all oxygen levels if this model is valid, but they vary a lot when oxygen level is below 10µM. However, the NADH concentration and bound fraction only vary a lot in this oxygen range. This should be addressed.

– The author used a mixture of LDH and NADH to prove the FLIM works in vitro. However, there are lots of different types of enzyme and complex in mitochondria that can bind NADH, and the author's model combines them together to do the calculations. Some justification of this is needed.

– In the spatial model, does variation of thickness of oocyte from center to periphery affect fluorescence levels? If yes, have author corrected this effect and how to correct?

– The nucleus will also cause heterogeneous distribution of mitochondria, which might also need to be considered in modeling spatial distribution.

*Reviewer #3:*

This paper describes an analysis of fluorescence lifetime imaging (FLIM) of NADH in mitochondria in intact mouse oocytes, using a mathematical model to interpret the fluorescence data to infer mitochondrial NADH redox fluxes. The authors measure FLIM data for varying oxygen concentrations and using several other perturbations to mitochondrial respiration, in order to infer consequential changes to key mitochondrial metabolic fluxes. One striking observation is of subcellular spatial gradients in the inferred metabolic flux across the oocytes.

The authors tackle an important issue in measurement and understanding mitochondrial function in intact cells. The analysis of the FLIM data is dependent on a mathematical model that the authors develop. The correctness and suitability of this model is not clear to me from the way it is described in the manuscript.

The analysis is based on a model, presented in Appendix 2, which considers 'course graining' of a 'detailed' NADH redox model. The latter considers N oxidases and M reductases acting on NADH and NAD+. The aim of the course graining approach is to reduce this model to an equivalent model with one effective oxidase and one effective reductase, and to calculate the effective binding and unbinding coefficients for this reduced model as functions of the binding and unbinding coefficients of the full model. This reduced model is then used to analyse the fluorescence lifetime imaging data, to infer mitochondrial redox fluxes, and to draw some conclusions on spatial gradients of mitochondrial function within the oocytes.

As the analysis and interpretation of the fluorescence lifetime imaging data depends closely on this model, it is necessary to (a) be convinced that the full model is an appropriate representation of the underlying system, and that (b) the course graining methodology is valid.

In relation to (a), the full model (Figure 2 in Appendix 2, but seemingly never presented as a system of mathematical equations), it is not clear why the authors have chosen this particular kinetic scheme, and there is seemingly no specific justification given for it. A simpler scheme would be (for example, for the ith oxidase)

NADH_f_ + Ox_i_ ⇌ Complex ⇌ NAD^+^_f_ + Ox_i_

i.e. a standard reversible Michaelis-Menten scheme. It isn't clear why the authors have chosen to represent these reactions with two complexes, when a simpler scheme might suffice.

A more serious concern is in relation to (b), the validity of the course graining procedure that is subsequently outlined. The schema that is presented in Appendix 2 has the binding rates to be independent of free enzyme concentration. Thus, in Equation S6, for example, the binding rate for NADH_f_ for the ith oxidase is k_oi_,_b_∙[NADH_f_]. But as per the standard analysis of enzyme reactions (Michaelis Menten, etc), following from basic mass action principles, this should depend on the free enzyme concentration, and should instead be given by k_oi_,_b_∙[NADH_f_]∙[Ox_i_] for the ith enzyme. No justification appears to be given as to why there is no dependence on the free enzyme concentration.

Unfortunately, a consequence of this is that the simple factorisation that allows equations S6-S9 to simplify into S10 and S11 is no longer possible. Essentially, by omitting the free enzyme concentration the authors end up considering a linear system which can be factored in the way they have presented, whereas the nonlinear system that is obtained when the free enzyme concentration is included does not allow this simple factorisation. It is this nonlinearity that generates the saturating behaviour in standard enzyme kinetics, for example.

Can the authors justify their linear model and omission of the free enzyme concentration? This does not appear to have been justified in the manuscript. One possibility that I can think of may be that the enzymes are all present in very high concentration, such that enzyme concentration is effectively constant. It is not clear that this is an appropriate regime. Alternatively, the authors should justify why the free enzyme concentration has been omitted in these equations. One way to demonstrate the validity of their approach would be through simulation, for example by selecting arbitrary parameters for a model with N and M oxidases and reductases and comparing full simulations of the nonlinear ODE system generated for this model with simulations of the reduced model derived from the course graining approach. As far as I could see there was no demonstration of the validity of the course grained model, however.

Otherwise, if the model is indeed incorrect, it is not clear what is the consequence of this will be on the subsequent data analysis (or indeed if the data can be analysed in this manner), given that Equation 1 in the main text and all subsequent analysis appear to follow directly from this assumption.

I hope that I am wrong, but if I am then I would strongly encourage the authors to provide further justification as to why their model and their course graining approach is correct and valid.

*Reviewer #4:*

Not every single cell is the same in terms of its metabolism. To study the causes of such cell-to-cell differences, we need microscopic tools to assess metabolic properties, such as metabolite levels and metabolic fluxes, on the single cell level or even beyond. While sensors exist to visualize certain metabolite levels, we still largely lack methods to assess metabolic fluxes in single cells. The work of Yang and Needleman presents a method that can assess -under certain assumptions- the flux through electron transport chain (ETC) in mitochondria of single mouse oocytes at quasi steady-states with subcellular resolution.

For their method, the authors use FLIM (fluorescence lifetime imaging microscopy) to determine the concentration of free and bound NADH in mitochondria, and these measurements are then used in a simple coarse-grained model to infer the flux through the ETC. This coarse-grained steady-state model describes the oxidation of NADH with one oxidase (resembling the ETC) and one NADH reductase (resembling all the 3 TCA cycle NADH dehydrogenases plus pyruvate dehydrogenase, but neglecting the FADH2-dependent succinate dehydrogenase) with only two free model parameters.

Strikingly, when fed with the FLIM data, this coarse-grained model could describe the outcomes of a number of perturbations, where the oxygen uptake rate (i.e. a proxy for the flux through the ETC) was independently measured with a different method. Applying the method, the authors also suggest that the ETC flux is higher in mitochondria that are rather located at the outside of the oocyte.

While FLIM measurements of bound and unbound NADH have been done before, the main strength of the paper is that it presents a method to infer metabolic activity in an oocyte, where the novelty resides on the development of the simple coarse-grained model and on showing that the model-based analysis of the FLIM data can allow to obtain quasi-steady-state ETC fluxes.

The main weakness of the paper is the following: Unfortunately, the work falls short on the application side. One would have wished that for a novel method like this, if it is indeed relevant, it should have been easy for the authors to add exciting application cases that would indeed generate novel biological insight.

While the main strength is the paper is the method (i.e. inference of ETC flux of model-based analysis of FLIM data), I feel that the description of the method, its assumptions etc falls short, which made assessment of the method and its potential limitations challenging. I feel that this is due to the fact that the writing of the manuscript is suboptimal. While the biochemistry is described/introduced on a very detailed textbook level, the methods, the measurements, the analyses of the measurement data in the result section and in the method section are described in a very short, condensed, and sometime convoluted, manner. As this is primarily meant to be a method paper, the authors need to do a better job in describing what they have done (i.e. model development, model assumptions, inference procedure, etc) in a clearer manner.

I felt that a strong point was that the two different versions of how the experimental data is used in the model, i.e. lifetime (tau) and bound ratio (β), leads to similarly inferred r_ox_. However, due to the above criticized too short explanations, I could not tell whether this would be trivial or not. Also, the whole method boils down to this equation J_ox_ = α * (β – β_eq_) * [NADH_f_], describing the full complexity of mitochondrial metabolism (TCA cycle, the electron transport chain, metabolite exchange between mitochondria and cytoplasm) with a single equation with only two free parameters (α, β_eq_). For this reviewer, also this part still remains somewhat elusive.

For publication of a new method in a journal such as *eLife*, I would expect that the manuscript also shows an application of the new method, which generates new biological insight, thereby demonstrating the power and value of the new method. With the measurements of the ETC flux in the spatially located mitochondria the authors start doing this, but unfortunately the manuscript does not develop this into anything interesting (i.e. unraveling of what could cause this gradient, where for instance one could do a dynamic addition of an inhibitor that would diffuse from the outside of the cell, etc). I think it is important to show a compelling case here.

[Editors’ note: further revisions were suggested prior to acceptance, as described below.]

Thank you for submitting your article "A coarse-grained NADH redox model enables inference of subcellular metabolic fluxes from fluorescence lifetime imaging" for consideration by *eLife*. Your article has been reviewed by 3 peer reviewers, one of whom is a member of our Board of Reviewing Editors, and the evaluation has been overseen by Naama Barkai as the Senior Editor. Overall, all 3 reviewers were very positive about your revised manuscript. We would be pleased to accept a revised version that includes the textual modifications suggested by Reviewer #3.

Because we could not reach two of the initial reviewers, we added an additional reviewer at this stage. The following individuals involved in review of your submission have agreed to reveal their identity: Jason W. Locasale (Reviewer #2); Denis V Titov (Reviewer #3).

Please see the suggestions made by Reviewer #3 for clarifying points in the text. No additional experiments - only text revisions - are required.

*Reviewer #1:*

The authors have significantly improved the manuscript, which is much clearer than before. I think it is very interesting, although I cannot fully evaluate the strength of the assumptions. The authors take great care to explicitly address the major concerns raised by reviewers with regards to the assumptions of the coarse-graining model. Whether these clarified assumptions are valid is not obvious to me, as such modeling is outside of my expertise, but the concern appears to be addressed as far as I can tell. At least, it is clear in the text what assumptions the authors are making.

The finding that ETC flux is not regulated by substrate availability but rather by intrinsic ATP synthesis and proton flux is extremely interesting. While not a criticism of the manuscript, it is worth noting, though, that this result could have been achieved by OCR alone - as the authors show. Therefore, a major strength of this method is the ability to quantify ETC flux with subcellular resolution.

*Reviewer #2:*

Accept

*Reviewer #3:*

I was not one of the original reviewers of this manuscript. I have carefully read the manuscript, previous comments by 4 reviewers and authors response. I feel that authors addressed all of the comments raised during original review.

Below I provide comments based on my review of this manuscript, but I want to highlight that I feel that the manuscript is strong in current form and that model is interesting and well validated and that this model will provide interesting insight into activity of ETC in single cells or even subcellular compartment that is difficult to measure currently.

1) Interpretation of Figure 8 data/ homeostasis of ETC flux in MII oocytes. I agree with the authors that Figure 8 clearly demonstrates that various perturbations can lead to changes in redox status of cells without affect Jox/OCR and it is impressive validation of the model that not of these changes affect Jox validated by no effect on OCR. It feels to this reviewer that it is still entirely possible that textbook view that Jox/OCR is regulated by ATP demand is still correct here and it just happened that none of the perturbation that authors used were enough to significantly change ATP consumption in this system and instead all perturbations somehow affected substrate supply without changing demand. Since no measurements of ATP, ADP, AMP have been reported by authors, it is difficult to be sure that any of the perturbation actually affected ATP demand in a significant way although I agree that would have been my expectation. Perhaps, authors should consider adding this caveat to the text that it is possible that none of the perturbations changes ATP demand or if authors strongly believe this hypothesis is correct then showing that large changes in ATP, ADP, AMP are observed without change in Jox/OCR would strongly support this hypothesis.

2) Why does lifetime of free NADH change with low oxygen and with other treatments in other figures)? I can imagine how protein bound lifetime might change as the fractional contributions of proteins to which NADH binds might change with changes in [NADH] but I would have expected free NADH lifetime to stay the same. Perhaps authors should comment on this in the text to clarify this for readers and maybe provide examples of factors that can change free NADH lifetime.

3) Figure 1c. What are the confidence intervals for fitting-based estimates of lifetime and fraction bound of each sample? I could be wrong, but it seems to me that fitting of double exponential equation to decay data might produce redundant values for estimates of lifetimes and fraction bound. I think it would be useful if authors could add confidence intervals (e.g., using bootstrapping with randomly drawn points with substitution from each lifetime curve) that estimate the uncertainty of the fitting-based estimates to show that the fitting procedure used by authors produces non-redundant values of lifetimes and fraction bound. These estimates of uncertainty could also be propagated to Jox to provide a better estimate of Jox uncertainty compared to using one set of values from each FLIM trace that authors use currently if I understood correctly.

---

## [Author Response]

[Editors’ note: the authors resubmitted a revised version of the paper for consideration. What follows is the authors’ response to the first round of review.]

We are sorry to say that, after consultation with the reviewers, we have decided that your work will not be considered further for publication by eLife at this time. The reviewers raised three major issues regarding the clarity of the manuscript, the validity of the model, and the potential of the methodology to be used to generate biological insights in oocytes or additional cell systems. There were several concerns about the validity of the assumptions of the model, most notably the concern that as the coarse-grained model describes a substrate-enzyme binding reaction with a rate that depends only on the substrate concentration, this model would be valid only under conditions in which enzyme concentrations were orders of magnitude higher than substrate concentration. More broadly, the reviewers raised questions as to whether the assumptions in the model are valid in other systems and whether the methodology will be broadly applicable for the generation of novel biological insights. Given these significant concerns, we are returning the manuscript reviews to you so that they may guide you as you seek publication elsewhere. Should you wish to resubmit at a future date, we would be happy to reconsider a revised version that (1) more clearly justifies the validity of the approach and its assumptions and (2) provides an application of the new method to generate new biological insight.

We thank the reviewers for their detailed and constructive comments on our manuscript. We have taken great effort to thoroughly address the issues raised by the reviewers. Our revised manuscript now clearly explains the assumptions of the model, validates the model on an additional cell type and provides two applications of the method to generate new biological insights. We are hereby resubmitting the manuscript for reconsideration for publication at *eLife*.

Regarding the binding reaction issue raised by Reviewer #3, we clarified that we did not assume a linear reaction model, but kept the model entirely general with all nonlinearities (including in the free enzyme concentrations) incorporated into the kinetic rates. The predicted relationship between FLIM measurements and ETC flux generally holds, independent of these nonlinearities. We introduced Figure 2 and Appendix 2 to illustrate this point and built the model systematically from this generalized enzyme kinetics. To demonstrate the general applicability of the model, we validated the model in an additional cell type: human tissue culture cells (Figure 7). This addresses the concerns from Reviewer #1. In response to the suggestions from Reviewer #4, we provided two applications of the method to gain biological insight: we discovered homeostasis (Figure 8) and subcellular heterogeneities (Figure 9) of ETC flux in mouse oocytes. We concluded from these observations that ETC fluxes in mouse oocytes are not controlled by energy demand or supply, but by the intrinsic rates of mitochondrial respiration. Thus, our work provides new insights into the spatiotemporal regulation of metabolic fluxes in cells.

We addressed all reviewers’ comments item-by-item below and made corresponding changes in the manuscript.

With these significant improvements of the manuscript, we hope our work can be reconsidered for publication at *eLife*.

Reviewer #1:In this manuscript, the authors present a model to relate FLIM measurements to mitochondrial metabolic fluxes. Using mouse oocytes, which have little NADPH, the authors develop a coarse-grained model to infer mitochondrial NADH oxidation by exploiting NAD(P)H FLIM. Using this approach, the authors uncover regional variation in mitochondrial fluxes in mouse oocytes. The modeled mitochondrial flux shows a strong negative correlation with mitochondrial membrane potential and no correlation with mitochondrial content. While this is not the first paper to use NAD(P)H FLIM to show subcellular metabolic variability, this manuscript does present a model to connect NAD(P)H FLIM to mitochondrial redox cycles. Therefore, the major utility of the model lies in its ability to provide subcellular information about mitochondrial NAD(P)H oxidation. The authors provide a comprehensive and accessible discussion of the assumptions, caveats, and conclusions enabled by their modeling. At present, however, it is not clear to this reviewer how generalizable this method will prove beyond mouse oocytes. This concern stems from the potential difficulty in establishing key parameters of the model in other cell types in which assumptions safely made in mouse oocytes may not be appropriate.To demonstrate the utility of their model, the authors should test key parameters in at least one additional cell type. In particular, the following issues should be addressed:

We appreciate the reviewers’ constructive comments. We have now validated our model in an additional cell type: human tissue culture cells. The results are summarized in the new section “The NADH redox model enables accurate prediction of ETC flux in human tissue culture cells” and in Figure 7. These results demonstrated the generality of the method. We address the reviewers’ specific comments item by item below.

1. A key requirement of the model is the ability to determine the equilibrium NADH bound ratio. Here, the authors use low oxygen (or rotenone) to establish this parameter. Will this be feasible in other cell types, for example those with active NNT?

We have demonstrated experimentally that we can also use rotenone to obtain the equilibrium NAD(P)H bound ratio in human tissue culture cells, which leads to accurate prediction of the ETC flux under various mitochondrial inhibitors and nutrient perturbations. The results are summarized in the section “NADH redox model enables accurate prediction of ETC flux in tissue culture cells” and in Figure 7 and Figure 7—figure supplement 1.

2. The authors note that the confounding signal from NADPH can be ignored in mouse oocytes, which have 40-fold higher NADH than NADPH. How generalizable is this? Will other mammalian cell types be amenable to this method?

Concentration of NADPH can be of the same order as NADH in other cell types, including mammalian tissue culture cells. Therefore, we have now explicitly accounted for the impact of NADPH signal and other fluorescence background signal on the flux inference procedure. We showed that if the background fluorescence is either constant under perturbations or changes proportionally with NADH, then it does not impact the inference procedure. This derivation is presented in Appendix 5 section “Accounting for NADPH and other background fluorescence”. We further tested this method experimentally in human tissue culture cells, in which NADPH is substantial, and accurately predicted the change of ETC flux under a variety of metabolic perturbations. Our results demonstrated that the presented flux inference method can work on other cell types, despite of the potential enrichment of NADPH.

3. The authors test the assumption that NADH signal originates in the mitochondria by comparing with mitotracker signal under control conditions. This should also be repeated for key conditions (e.g. low oxygen or oxamate treatment).

We repeated the test for oxamate, oligomycin, fccp and rotenone conditions, which yielded a NADH signal originating from mitochondria of 78.6% ± 1.4%, 84.1% ± 1.6%, 83.7% ± 0.5%, 81.7% ± 2.0%, respectively, similar to the control condition. We added these data to the manuscript.

Reviewer #2:In the manuscript "Coarse-grained model of mitochondrial metabolism enables subcellular flux inference from fluorescence lifetime imaging of NADH", the authors use fluorescence imaging to estimate NADH/NAD turnover flux and electron transfer rate in the mitochondria of mouse oocytes. Because of high spatial resolution of microscopy, the authors could also observe significant subcellular spatial gradient of oxidative flux in oocytes.The fluorescence imaging and quantification of flux are generally solid and convincing, but there are issues that need to be addressed.

We appreciate the positive comments of the reviewer. We address the issues below.

– In figure 1c, the author estimated two parameters τl and τs under different oxygen levels. They should be constant in all oxygen levels if this model is valid, but they vary a lot when oxygen level is below 10µM. However, the NADH concentration and bound fraction only vary a lot in this oxygen range. This should be addressed.

The fluorescence lifetimes τl and τs do not have to be constant for the model to be valid. In fact, we have shown that the variation of τl provides an additional method, in a self-consistent way with the bound ratio method, to infer ETC flux. Under this framework, the variations of τl could be attributed to the change of the relative abundance of bound NADH to the oxidases and reductases in the cell (Equation 6). We have now included the derivations and experimental validations in the section “Accurately predicting ETC flux from FLIM of NADH using the NADH redox model” and in Appendix 5.

– The author used a mixture of LDH and NADH to prove the FLIM works in vitro. However, there are lots of different types of enzyme and complex in mitochondria that can bind NADH, and the author's model combines them together to do the calculations. Some justification of this is needed.

FLIM is well established to be able to distinguish bound and free NADH in vivo based on the large change of fluorescence lifetime when NADH binds to enzymes (Bird et al., 2005; Skala et al., 2007; Heikal, 2010; Sharick et al., 2018; Sanchez et al., 2019). Even though how much lifetime changes depends on the specific enzyme NADH binds to, the enzyme-bound NADH always has a much longer fluorescence lifetime than free NADH, so a two exponential fitting of the fluorescence decay curve will yield an average lifetime and fraction of NADH bound to all enzymes in vivo. Therefore, the method to calculate free and bound concentrations of NADH from FLIM measurements is expected to hold in vivo. We have now clarified this point in the manuscript.

– In the spatial model, does variation of thickness of oocyte from center to periphery affect fluorescence levels? If yes, have author corrected this effect and how to correct?

We have shown that the intensity gradient is present for NADH and TMRM, but not for

Mitotracker (Figure 9—figure supplement 2), hence ruling out the impact of thickness variations.

– The nucleus will also cause heterogeneous distribution of mitochondria, which might also need to be considered in modeling spatial distribution.

We are using oocytes arrested at Meiosis II, which do not have a nucleus. The presence of nucleus in other cell types will indeed need to be considered in studying spatial distribution of metabolism.

Reviewer #3:This paper describes an analysis of fluorescence lifetime imaging (FLIM) of NADH in mitochondria in intact mouse oocytes, using a mathematical model to interpret the fluorescence data to infer mitochondrial NADH redox fluxes. The authors measure FLIM data for varying oxygen concentrations and using several other perturbations to mitochondrial respiration, in order to infer consequential changes to key mitochondrial metabolic fluxes. One striking observation is of subcellular spatial gradients in the inferred metabolic flux across the oocytes.The authors tackle an important issue in measurement and understanding mitochondrial function in intact cells. The analysis of the FLIM data is dependent on a mathematical model that the authors develop. The correctness and suitability of this model is not clear to me from the way it is described in the manuscript.The analysis is based on a model, presented in Appendix 2, which considers 'course graining' of a 'detailed' NADH redox model. The latter considers N oxidases and M reductases acting on NADH and NAD+. The aim of the course graining approach is to reduce this model to an equivalent model with one effective oxidase and one effective reductase, and to calculate the effective binding and unbinding coefficients for this reduced model as functions of the binding and unbinding coefficients of the full model. This reduced model is then used to analyse the fluorescence lifetime imaging data, to infer mitochondrial redox fluxes, and to draw some conclusions on spatial gradients of mitochondrial function within the oocytes.As the analysis and interpretation of the fluorescence lifetime imaging data depends closely on this model, it is necessary to (a) be convinced that the full model is an appropriate representation of the underlying system, and that (b) the course graining methodology is valid.In relation to (a), the full model (Figure 2 in Appendix 2, but seemingly never presented as a system of mathematical equations), it is not clear why the authors have chosen this particular kinetic scheme, and there is seemingly no specific justification given for it. A simpler scheme would be (for example, for the ith oxidase)NADH_f_ + Ox_i_ ⇌ Complex ⇌ NAD^+^_f_ + Ox_i_i.e. a standard reversible Michaelis-Menten scheme. It isn't clear why the authors have chosen to represent these reactions with two complexes, when a simpler scheme might suffice.A more serious concern is in relation to (b), the validity of the course graining procedure that is subsequently outlined. The schema that is presented in Appendix 2 has the binding rates to be independent of free enzyme concentration. Thus, in Equation S6, for example, the binding rate for NADH_f_ for the ith oxidase is k_oi_,_b_∙[NADH_f_]. But as per the standard analysis of enzyme reactions (Michaelis Menten, etc), following from basic mass action principles, this should depend on the free enzyme concentration, and should instead be given by k_oi_,_b_∙[NADH_f_]∙[Ox_i_] for the ith enzyme. No justification appears to be given as to why there is no dependence on the free enzyme concentration.Unfortunately, a consequence of this is that the simple factorisation that allows equations S6-S9 to simplify into S10 and S11 is no longer possible. Essentially, by omitting the free enzyme concentration the authors end up considering a linear system which can be factored in the way they have presented, whereas the nonlinear system that is obtained when the free enzyme concentration is included does not allow this simple factorisation. It is this nonlinearity that generates the saturating behaviour in standard enzyme kinetics, for example.Can the authors justify their linear model and omission of the free enzyme concentration? This does not appear to have been justified in the manuscript. One possibility that I can think of may be that the enzymes are all present in very high concentration, such that enzyme concentration is effectively constant. It is not clear that this is an appropriate regime. Alternatively, the authors should justify why the free enzyme concentration has been omitted in these equations. One way to demonstrate the validity of their approach would be through simulation, for example by selecting arbitrary parameters for a model with N and M oxidases and reductases and comparing full simulations of the nonlinear ODE system generated for this model with simulations of the reduced model derived from the course graining approach. As far as I could see there was no demonstration of the validity of the course grained model, however.Otherwise, if the model is indeed incorrect, it is not clear what is the consequence of this will be on the subsequent data analysis (or indeed if the data can be analysed in this manner), given that Equation 1 in the main text and all subsequent analysis appear to follow directly from this assumption.I hope that I am wrong, but if I am then I would strongly encourage the authors to provide further justification as to why their model and their course graining approach is correct and valid.

We thank the reviewer for the detailed and in-depth comments. We agree that we did not do a good job of describing the model in the original manuscript, which caused confusion regarding the notation and assumptions underlying the model. One of the sources of confusion is our usage of a reduced notation which was not adequately explained. We have now introduced Figure 2 together with explicit mathematical descriptions (Appendix 2) to explain the reduced notation.

Because of the use of our reduced notation, it might seem that we have assumed that the binding rate is linear in NADH concentration and does not depend on the free enzyme concentration and other factors. However, this is not the case. In our reduced notation, these dependencies are embedded in the rates themselves. As a result, all of the rates can depend on many variables including the free enzyme concentration (such as [Ox_i_]), NADH concentration and other factors. These additional variables will obey their own dynamical equations. We used this reduced notation because we want to capture a broad class of models, including models where the rates have non-linear dependencies on these variables. Modeling the dynamics of the redox pathways would require specify these dependencies: i.e, writing down the explicit functional form through which the rates depend on these variables and dynamical equations for the temporal variation of these variables. Remarkably, the relationship between FLIM parameters and the predicted ETC flux (Equations 5a-c) does not depend on these modeling details. Thus, using the reduced notation allows us to derive these general results without specifying the detailed enzyme kinetics. For clarity, we now also explicitly show that our general model includes the reversible Michaelis-Menten model as a special case (Appendix 7).

The coarse-graining procedure is mathematically exact and independent of the functional forms of the kinetic rates. In our reduced notation, all the coarse-grained parameters can also have complex dependencies on free enzyme concentrations, NADH concentrations, etc. To completely describe the dynamics of the coarse-grained model would also require specifying these dependencies and the associated dynamical equations. However, they do not affect the predicted relationship between FLIM parameters and ETC flux. As an example, we have also explicitly demonstrated the coarse-graining for a redox cycle in which each of the enzymes obey reversible Michaelis-Menten kinetics (Appendix 7).

The reason our generalized model works is because the predicted flux only depends on two coarse-grained parameters, α and βeq, which are experimentally determined. The detailed kinetics of a specific enzyme model only modifies the functional forms of α and βeq. For example, we have explicitly shown the functional dependence of α and βeq on the parameters of the reversible Michaelis-Menten kinetics (Appendix 7, Equations S73-S74). Since α and βeq can be measured experimentally with FLIM of NADH, they can be used for the flux inference without knowing their functional dependencies.

For our inference method to work, we assumed that the NADH redox loop is at steady state. This assumption is sufficient to predict the relationship between FLIM parameters and the ETC flux (Equations 5a-c). For this relationship to be useful for flux inference, we need to be able to determine α and βeq reliably from experiments. To determine βeq, we need to perform an experiment to inhibit ETC flux without changing the value of βeq. In our work, we chose oxygen drop experiment or rotenone inhibition for this purpose. We also assumed that α is a constant across perturbations, and hence we determined α in a control condition and used the same value across all other perturbations. Thus, the three key assumptions are: (1) that the NADH redox loop can be well approximated as being at steady-state; (2) that βeq does not vary due to the perturbation used to inhibit ETC flux and measure βeq; (3) that α is constant across perturbations. It is an empirical question if these assumptions hold. We tested our model predictions by comparing the predicted ETC flux with direct measurements of oxygen consumption rate (OCR) across a wide range of perturbations for both mouse oocytes (Figure 5, Figure 8) and human tissue culture cells (Figure 7). The quantitative agreement between the predicted flux and OCR robustly validated our model assumptions.

With the improved model presentation and additional experimental validation, we hope we have convinced the reviewer of the correctness and suitability of our method. We thank the reviewer again for giving us the opportunity to significantly improve our manuscript.

Reviewer #4:Not every single cell is the same in terms of its metabolism. To study the causes of such cell-to-cell differences, we need microscopic tools to assess metabolic properties, such as metabolite levels and metabolic fluxes, on the single cell level or even beyond. While sensors exist to visualize certain metabolite levels, we still largely lack methods to assess metabolic fluxes in single cells. The work of Yang and Needleman presents a method that can assess -under certain assumptions- the flux through electron transport chain (ETC) in mitochondria of single mouse oocytes at quasi steady-states with subcellular resolution.For their method, the authors use FLIM (fluorescence lifetime imaging microscopy) to determine the concentration of free and bound NADH in mitochondria, and these measurements are then used in a simple coarse-grained model to infer the flux through the ETC. This coarse-grained steady-state model describes the oxidation of NADH with one oxidase (resembling the ETC) and one NADH reductase (resembling all the 3 TCA cycle NADH dehydrogenases plus pyruvate dehydrogenase, but neglecting the FADH2-dependent succinate dehydrogenase) with only two free model parameters.Strikingly, when fed with the FLIM data, this coarse-grained model could describe the outcomes of a number of perturbations, where the oxygen uptake rate (i.e. a proxy for the flux through the ETC) was independently measured with a different method. Applying the method, the authors also suggest that the ETC flux is higher in mitochondria that are rather located at the outside of the oocyte.While FLIM measurements of bound and unbound NADH have been done before, the main strength of the paper is that it presents a method to infer metabolic activity in an oocyte, where the novelty resides on the development of the simple coarse-grained model and on showing that the model-based analysis of the FLIM data can allow to obtain quasi-steady-state ETC fluxes.The main weakness of the paper is the following: Unfortunately, the work falls short on the application side. One would have wished that for a novel method like this, if it is indeed relevant, it should have been easy for the authors to add exciting application cases that would indeed generate novel biological insight.While the main strength is the paper is the method (i.e. inference of ETC flux of model-based analysis of FLIM data), I feel that the description of the method, its assumptions etc falls short, which made assessment of the method and its potential limitations challenging. I feel that this is due to the fact that the writing of the manuscript is suboptimal. While the biochemistry is described/introduced on a very detailed textbook level, the methods, the measurements, the analyses of the measurement data in the result section and in the method section are described in a very short, condensed, and sometime convoluted, manner. As this is primarily meant to be a method paper, the authors need to do a better job in describing what they have done (i.e. model development, model assumptions, inference procedure, etc) in a clearer manner.I felt that a strong point was that the two different versions of how the experimental data is used in the model, i.e. lifetime (tau) and bound ratio (β), leads to similarly inferred r_ox_. However, due to the above criticized too short explanations, I could not tell whether this would be trivial or not. Also, the whole method boils down to this equation J_ox_ = α * (β – β_eq_) * [NADH_f_], describing the full complexity of mitochondrial metabolism (TCA cycle, the electron transport chain, metabolite exchange between mitochondria and cytoplasm) with a single equation with only two free parameters (α, β_eq_). For this reviewer, also this part still remains somewhat elusive.

We thank the reviewer for the detailed and in-depth review of our manuscript. We appreciate the reviewer’s suggestion to add application cases to demonstrate the usefulness of our method. We now added two application of our flux inference procedure to the revised manuscript. The first case is the discovery of homeostasis of ETC flux in mouse oocytes: perturbations of nutrient supply and energy demand do not change ETC flux despite significantly impacting NADH metabolic state (Figure 8). The second case is the discovery of the intracellular spatial gradient of ETC flux in mouse oocytes. As suggested by the reviewer, we have used metabolic inhibitors to help reveal the cause of this gradient and found that this gradient is primarily a result of a spatially heterogeneous mitochondrial proton leak (Figure 9). We concluded from these observations that ETC flux in mouse oocytes is not controlled by energy demand or supply, but by the intrinsic rates of mitochondrial respiration.

We thank the reviewer for their suggestions to improve the presentation of this work. We have significantly rewritten the paper to clearly describe the model development, model assumptions, data analysis procedures and results. Regarding the comparison of the two inference methods, we presented details of the assumptions and derivations in the Results section and demonstrated that the agreement between these two methods is not trivial, and is a robust self-consistency check of the method. We also now explained the coarse-graining procedure in detail in the main text and in Appendix 2 and 3 to demonstrate how all the model complexities are coarse-grained into only two free parameters α and βeq. In a nutshell, α and βeq would be functions of the kinetic rates of the model, with the kinetic rates depending on the details of mitochondrial metabolism. However, using the model to infer ETC flux does not require knowing the functional forms of α and βeq, because α and βeq can be experimentally measured with FLIM. The only assumptions required are that α remains a constant under perturbations and βeq can be determined from ETC inhibitions. These assumptions are validated experimentally in mouse oocytes and human tissue culture cells from the agreement between predicted ETC flux and direct measurements of OCR.

We also validated our model in an additional cell type of human tissue culture cells, demonstrating the generality of our method (Figure 7).

For publication of a new method in a journal such as eLife, I would expect that the manuscript also shows an application of the new method, which generates new biological insight, thereby demonstrating the power and value of the new method. With the measurements of the ETC flux in the spatially located mitochondria the authors start doing this, but unfortunately the manuscript does not develop this into anything interesting (i.e. unraveling of what could cause this gradient, where for instance one could do a dynamic addition of an inhibitor that would diffuse from the outside of the cell, etc). I think it is important to show a compelling case here.

We have now strengthened the paper with two compelling cases of application: homeostasis and heterogeneity of ETC flux. We have shown that perturbations of nutrient supply and energy demand impact NADH metabolic state but do not impact ETC flux in mouse oocytes (Figure 8). We have also unraveled the cause of the ETC flux gradient in the oocyte using oligomycin and discovered that the flux gradient is caused by a higher rate of proton leak of mitochondria closer to the cell periphery (Figure 9). We concluded from these observations that ETC flux in mouse oocytes is not controlled by energy demand or supply, but by the intrinsic rates of mitochondrial respiration. Our work provided new insights into the spatiotemporal regulations of metabolic fluxes in cells.

[Editors’ note: what follows is the authors’ response to the second round of review.]

Please see the suggestions made by Reviewer #3 for clarifying points in the text. No additional experiments - only text revisions - are required.Reviewer #3:I was not one of the original reviewers of this manuscript. I have carefully read the manuscript, previous comments by 4 reviewers and authors response. I feel that authors addressed all of the comments raised during original review.Below I provide comments based on my review of this manuscript, but I want to highlight that I feel that the manuscript is strong in current form and that model is interesting and well validated and that this model will provide interesting insight into activity of ETC in single cells or even subcellular compartment that is difficult to measure currently.

We thank the reviewer for the positive comments and detailed suggestions on our manuscript.

1) Interpretation of Figure 8 data/ homeostasis of ETC flux in MII oocytes. I agree with the authors that Figure 8 clearly demonstrates that various perturbations can lead to changes in redox status of cells without affect Jox/OCR and it is impressive validation of the model that not of these changes affect Jox validated by no effect on OCR. It feels to this reviewer that it is still entirely possible that textbook view that Jox/OCR is regulated by ATP demand is still correct here and it just happened that none of the perturbation that authors used were enough to significantly change ATP consumption in this system and instead all perturbations somehow affected substrate supply without changing demand. Since no measurements of ATP, ADP, AMP have been reported by authors, it is difficult to be sure that any of the perturbation actually affected ATP demand in a significant way although I agree that would have been my expectation. Perhaps, authors should consider adding this caveat to the text that it is possible that none of the perturbations changes ATP demand or if authors strongly believe this hypothesis is correct then showing that large changes in ATP, ADP, AMP are observed without change in Jox/OCR would strongly support this hypothesis.

We thank the reviewer for the insightful comments. We agree that while NADH metabolic state significantly changed in response to perturbing energy demand and supply, indicating cell metabolism was indeed impacted, it is unclear if these perturbations also influenced ATP, ADP or AMP levels. Future work, including direct measurements of ATP, ADP and AMP levels, will be required to uncover the mechanism of flux homeostasis. We added texts in the discussion (pg. 26) to clarify this point.

2) Why does lifetime of free NADH change with low oxygen and with other treatments in other figures)? I can imagine how protein bound lifetime might change as the fractional contributions of proteins to which NADH binds might change with changes in [NADH] but I would have expected free NADH lifetime to stay the same. Perhaps authors should comment on this in the text to clarify this for readers and maybe provide examples of factors that can change free NADH lifetime.

We have clarified in the text (pg. 4) that in addition to enzyme binding (Sharick et al., 2018, Ghukasyan and Heikal, 2015), NADH fluorescence lifetimes are also impacted by other factors including viscosity and pH (Ghukasyan and Heikal, 2015). We expect that these latter effects might be responsible for the observed changes in the lifetime of free NADH.

3) Figure 1c. What are the confidence intervals for fitting-based estimates of lifetime and fraction bound of each sample? I could be wrong, but it seems to me that fitting of double exponential equation to decay data might produce redundant values for estimates of lifetimes and fraction bound. I think it would be useful if authors could add confidence intervals (e.g., using bootstrapping with randomly drawn points with substitution from each lifetime curve) that estimate the uncertainty of the fitting-based estimates to show that the fitting procedure used by authors produces non-redundant values of lifetimes and fraction bound. These estimates of uncertainty could also be propagated to Jox to provide a better estimate of Jox uncertainty compared to using one set of values from each FLIM trace that authors use currently if I understood correctly.

We thank the reviewer for the suggestions. In this manuscript, we independently fit FLIM curves for each individual oocyte. The reported error bars in this manuscript are standard errors of the mean (SEMs) across these measurements, which depends on the level of variation (the standard deviation) between the oocytes. Two sources of variation in FLIM measurements across the oocytes are: (1) true biological variations between oocytes and (2) fitting errors in the FLIM analysis. To estimate the error of fitting, we performed bootstrapping with randomly drawn points with substitution from each fluorescence decay curve for 53 oocytes. There are ~66000 photons per oocyte, from which we generated 10 bootstrapped decay curves per oocyte to estimate the fitting error. The fitting error is computed as the variance and covariance of the fitted parameters across bootstrapped decay curves and averaged over 53 oocytes.

For oocytes at high oxygen levels in AKSOM, the bootstrapping yields a variance of 2.2×10^-4^, 4.6×10^-3^ ns^2^, 6.0×10^-4^ ns^2^ for bound fraction, long lifetime and short lifetime, respectively. The cell-to-cell variances obtained from a single fit per oocyte are 4.4×10^-4^, 9.5×10^-3^ ns^2^, 1.6×10^-3^ ns^2^ for bound fraction, long lifetime, short lifetime, respectively. Hence fitting errors account for 50%, 49% and 40% of the cell-to-cell variance in bound fraction, long lifetime and short lifetime, respectively.

The inferred mean flux for oocytes at high oxygen levels in AKSOM is ⟨Jox⟩=56.6 μM⋅s−1. Propagating the error of fitting in all parameters from the bootstrapping analysis to the inferred flux gives a standard error of the mean in Jox of 1.1 µM·s^-1^. The standard error of the mean in Jox obtained from a single fit per oocyte was 2.0 µM·s^-1^. Thus, fitting errors account for ∼50% of the standard error of the mean in Jox.

Reviewer 3 is correct that fitting errors can induce covariations between long lifetime and fraction bound: our bootstrapping analysis indicates that with ~66000 photons per oocyte this covariance is of -1.0×10^-3^ ns. We believe that such fitting errors do not significantly impact our results for two reasons: (1) The level of covariation between these two parameters in this study is much greater than can be accounted for by such errors. For example, the covariance between long lifetime and bound fraction during oxygen drop experiments is -4.6×10^-3^ ns, so fitting errors can only account for ~20% of this covariation; (2) Under other conditions, such as during mouse preimplantation embryo development, we observe that long lifetime and bound fraction are *positively* correlated during certain developmental stages (Sanchez et al. 2019).

We have added a section of error analysis in the methods section of the manuscript (pg. 30).